# Incremental Gradient Descent with Small Epoch Counts is Surprisingly Slow on Ill-Conditioned Problems

Yujun Kim [* 1]   Jaeyoung Cha [* 1]   Chulhee Yun [1]

## Abstract

Recent theoretical results demonstrate that the convergence rates of permutation-based SGD (e.g., random reshuffling SGD) are faster than uniform-sampling SGD; however, these studies focus mainly on the *large epoch regime*, where the number of epochs $K$ exceeds the condition number $\kappa$. In contrast, little is known when $K$ is smaller than $\kappa$, and it is still a challenging open question whether permutation-based SGD can converge faster in this *small epoch regime* (Safran & Shamir, 2021). As a step toward understanding this gap, we study the naive deterministic variant, Incremental Gradient Descent (IGD), on smooth and strongly convex functions. Our lower bounds reveal that for the small epoch regime, IGD can exhibit surprisingly slow convergence even when all component functions are strongly convex. Furthermore, when some component functions are allowed to be nonconvex, we prove that the optimality gap of IGD can be significantly worse throughout the small epoch regime. Our analyses reveal that the convergence properties of permutation-based SGD in the small epoch regime may vary drastically depending on the assumptions on component functions. Lastly, we supplement the paper with tight upper and lower bounds for IGD in the large epoch regime.

## 1. Introduction

Many machine learning and deep learning tasks can be formulated as finite-sum minimization problems:

$$\min_{\boldsymbol{x} \in \mathbb{R}^d} F(\boldsymbol{x}) := \frac{1}{n} \sum_{i=1}^{n} f_i(\boldsymbol{x}),$$

where the objective $F(\boldsymbol{x})$ is the average of a finite number of component functions $f_i(\boldsymbol{x})$. In modern deep learning applications, the number of components $n$ is often extremely large, making full gradient optimization methods computationally expensive. To address this, stochastic gradient descent (SGD) and its variants have gained attention for their computational efficiency and scalability (Lan, 2020).

SGD methods can be categorized based on the strategy used to select the component index $i(t)$ at iteration $t$: (1) *with-replacement* SGD, and (2) *permutation-based* SGD. In with-replacement SGD, also known as *uniform-sampling* SGD, each index is drawn independently from a uniform distribution over $\{1, 2, \ldots, n\}$. This approach has been the primary focus of theoretical studies, as it guarantees the stochastic gradient at each step to be an unbiased estimator of the gradient of the overall objective $F$ (Bubeck et al., 2015).

In contrast, permutation-based SGD—where indices are sampled in a shuffled order, also referred to as *without-replacement* SGD or *shuffling gradient methods*—is more commonly used in practice. Its popularity arises from strong empirical performance and simplicity of implementation, making it the standard choice for real-world machine learning applications. However, despite its widespread use, the theoretical understanding of permutation-based SGD had remained underdeveloped until recently, due to challenges arising from the lack of independence between iterates.

Nevertheless, recent advances have successfully addressed the theoretical challenges of permutation-based SGD (Haochen & Sra, 2019; Nagaraj et al., 2019). For example, Random Reshuffling (RR), one of the most common permutation-based methods, randomly shuffles the indices at the start of each epoch. It has been theoretically shown that RR achieves a convergence rate of $\mathcal{O}(1/nK^2)$ for smooth and strongly convex objectives, which is faster than the rate $\mathcal{O}(1/nK)$ of with-replacement SGD, where $K$ is the number of epochs (Ahn et al., 2020; Mishchenko et al., 2020).

While these results suggest that RR is theoretically superior to with-replacement methods, the story is far from complete. Existing analyses of permutation-based SGD are mostly restricted to the large epoch regime, where $K$ is sufficiently large relative to the problem's condition number $\kappa$ (defined

[*]Equal contribution [1]Kim Jaechul Graduate School of AI, Korea Advanced Institute of Science and Technology. Correspondence to: Chulhee Yun <chulhee.yun@kaist.ac.kr>.

*Proceedings of the 42nd International Conference on Machine Learning*, Vancouver, Canada. PMLR 267, 2025. Copyright 2025 by the author(s).

in Section 2.1). However, this regime is often unrealistic in practical machine learning scenarios, especially when training large language models. Neural network training typically involves highly ill-conditioned optimization landscapes (Li et al., 2018; Ghorbani et al., 2019), where $\kappa$ is large, and $K$ is comparatively small due to computational constraints. In such cases, the small epoch regime, where $K$ is smaller than $\kappa$, becomes significantly more relevant, yet its convergence behavior remains poorly understood.

In fact, Safran & Shamir (2021) establish a lower bound in strongly convex objectives for RR, revealing that RR cannot outperform with-replacement SGD in the small epoch regime. This highlights the need to further investigate permutation-based methods under small epoch constraints and explore whether they can outperform with-replacement SGD in such settings. However, analyzing permutation-based SGD in the small epoch regime poses significant theoretical challenges (as explained in Section 2.4). Even for Incremental Gradient Descent (IGD) (Bertsekas, 2011; Gurbuzbalaban et al., 2019), the simplest permutation-based SGD method where components are processed sequentially and deterministically from indices 1 to $n$ in each epoch, its convergence behavior in this regime is not well-understood.

In this study, as an initial step toward understanding the convergence behavior of permutation-based SGD in the small epoch regime, we focus on the convergence analysis of IGD. Our study presents convergence rates for both the small epoch regime and the large epoch regime, offering a result that highlights the distinct behavior of permutation-based SGD in the small epoch regime.

### 1.1. Summary of Our Contributions

Our analysis focuses on the setting where the objective $F$ is smooth and strongly convex, and the step size is kept constant throughout the optimization process. We summarize our contributions as follows, where the convergence rates reflect the function optimality gap at the final iterate. For a clear overview, we refer readers to Table 1 and Figure 1.

- In Section 3, we provide convergence analyses of IGD in the small epoch regime. We establish lower bound convergence rates under three scenarios (Theorems 3.1, 3.3 and 3.5): (i) strongly convex components sharing the same Hessian, (ii) strongly convex components, and (iii) allowing nonconvex components. Additionally, we provide the upper bound convergence rates for the first two cases (Theorem 3.2, Proposition 3.4). Our results indicate that even with stronger assumptions, IGD remains slower than the known upper bound of with-replacement SGD. Furthermore, IGD exhibits surprisingly slow convergence even when all components are strongly convex, and the inclusion of nonconvex components further amplifies this slowdown.

- In Section 3.2, we study whether a suitable permutation strategy can accelerate permutation-based SGD in the small epoch regime. We prove that there exists a permutation such that repeatedly using it in permutation-based SGD can outperform with-replacement SGD (Theorem 3.7). To our knowledge, this is the first result showing the existence of a permutation-based SGD method that converges faster than with-replacement SGD in this regime.

- In Section 4, we establish tight convergence rates for IGD in the large epoch regime. We derive matching lower and upper bound rates, up to polylogarithmic factors, for scenarios where all components are convex or some are nonconvex (Theorems 4.1, 4.3 and 4.4). Unlike in the small epoch regime where nonconvex components significantly slow convergence, the rate gap between these two scenarios is only a factor of $\kappa$, revealing the clear distinction in the behavior of IGD in the small and large epoch regimes.

## 2. Preliminaries

We start by introducing the basic notation used throughout this paper. We use $n$ to denote the number of component functions and $K$ to denote the total number of epochs. The Euclidean norm is denoted by $\|\cdot\|$. For a positive integer $N \in \mathbb{N}$, we use $[N]$ to represent the set $\{1, 2, \ldots, N\}$. The symbol $q = \mathrm{poly}(p_1, \ldots, p_s)$ means that $q$ can be expressed as a finite sum of monomials of the form $p_1^{c_1} p_2^{c_2} \cdots p_s^{c_s}$, where each $c_i$ is a bounded real number (which may be negative or non-integer). Similarly, $q = \mathrm{polylog}(p_1, \ldots, p_s)$ denotes a function expressible as $q = \sum_{(c, c_1, \ldots, c_s)} \log^c \prod_{i \in [s]} p_i^{c_i}$ for bounded real $c$ and $c_i$.

Importantly, while existing works use $\mathcal{O}$ and $\Omega$ (or $\tilde{\mathcal{O}}$ and $\tilde{\Omega}$ to hide polylogarithmic factors) to express the growth rates of the convergence rates, we adopt the symbols $\lesssim$ and $\gtrsim$ in this paper to describe our results in better detail. Formally, $x \lesssim y$ means that there exists a universal constant $c > 0$ such that $x \leq c \cdot y \cdot \mathrm{polylog}(n, K, \mu, L, \ldots)$ holds for the specified $n$, $K$, and other parameters; vice-versa, $x \gtrsim y$ means $x \geq c \cdot y \cdot \mathrm{polylog}(n, K, \mu, L, \ldots)$. Unlike $\mathcal{O}$ and $\Omega$, which are often used to express the asymptotic behavior of the rate as $K \to \infty$, $\lesssim$ and $\gtrsim$ here apply to all valid values of $K$. The reason for using these symbols is that many of the upper and lower bounds in this paper are established in the small epoch regime, where the total number of epochs $K$ is explicitly bounded above by the condition number $\kappa$.

### 2.1. Definitions and Assumptions

We list definitions and assumptions that will be used to describe the function class.

**Definition 2.1** (Smoothness). A differentiable function $F : \mathbb{R}^d \to \mathbb{R}$ is *L-smooth*, for some $L > 0$, if

$$\|\nabla F(\boldsymbol{x}) - \nabla F(\boldsymbol{y})\| \leq L\|\boldsymbol{x} - \boldsymbol{y}\|, \ \forall \boldsymbol{x}, \boldsymbol{y} \in \mathbb{R}^d.$$

*Table 1.* Summary of our results. All upper bounds, except for Theorem 3.7, apply to arbitrary permutation-based SGD. Theorem 3.7 specifically applies to a permutation-based SGD method proposed in its theorem. All lower bound results apply to IGD.

| Epoch | Component Assumption | Convergence Rate | Gradient Assumption |
|---|---|---|---|
| Small $K \lesssim \kappa$ | Strongly Convex Identical Hessian | $\mathcal{O}\left(\frac{G_*^2}{\mu K}\right)$, Theorem 3.2 | $\|\nabla f_i(\boldsymbol{x}^*)\| \leq G_*$ |
| | | $\Omega\left(\frac{G^2}{\mu K}\right)$, Theorem 3.1 | $\|\nabla f_i(\boldsymbol{x}) - \nabla F(\boldsymbol{x})\| \leq G, \forall \boldsymbol{x}$ |
| | Strongly Convex | $\mathcal{O}\left(\frac{L^2 G_*^2}{\mu^3 K^2}\right)$, Mishchenko et al. (2020) | $\|\nabla f_i(\boldsymbol{x}^*)\| \leq G_*$ |
| | | $\Omega\left(\frac{LG^2}{\mu^2} \min\{1, \frac{L^2}{\mu^2 K^4}\}\right)$, Theorem 3.3 | $\|\nabla f_i(\boldsymbol{x}) - \nabla F(\boldsymbol{x})\| \leq G + \|\nabla F(\boldsymbol{x})\|, \forall \boldsymbol{x}$ |
| | Potentially Nonconvex | $\Omega\left(\frac{G^2}{L}\left(1 + \frac{L}{2\mu n K}\right)^n\right)$, Theorem 3.5 | $\|\nabla f_i(\boldsymbol{x}) - \nabla F(\boldsymbol{x})\| \leq G + 3\|\nabla F(\boldsymbol{x})\|, \forall \boldsymbol{x}$ |
| | Strongly Convex | $\mathcal{O}\left(\frac{H^2 L^2 G_*^2}{\mu^3 n^2 K^2}\right)$, Theorem 3.7[1] | $\|\nabla f_i(\boldsymbol{x}^*)\| \leq G_*$ |
| Large $K \gtrsim \kappa$ | Convex | $\mathcal{O}\left(\frac{LG_*^2}{\mu^2 K^2}\right)$, Liu & Zhou (2024a) | $\frac{1}{n}\sum_{i=1}^n \|\nabla f_i(\boldsymbol{x}^*)\| \leq G_*$ |
| | | $\Omega\left(\frac{LG^2}{\mu^2 K^2}\right)$, Theorem 4.1 | $\|\nabla f_i(\boldsymbol{x}) - \nabla F(\boldsymbol{x})\| \leq G, \forall \boldsymbol{x}$ |
| | Potentially Nonconvex | $\mathcal{O}\left(\frac{L^2 G^2}{\mu^3 K^2}\right)$, Theorem 4.4[2] | $\|\nabla f_i(\boldsymbol{x}) - \nabla F(\boldsymbol{x})\| \leq G + P\|\nabla F(\boldsymbol{x})\|, \forall \boldsymbol{x}$ |
| | | $\Omega\left(\frac{L^2 G^2}{\mu^3 K^2}\right)$, Theorem 4.3[3] | $\|\nabla f_i(\boldsymbol{x}) - \nabla F(\boldsymbol{x})\| \leq G + \kappa\|\nabla F(\boldsymbol{x})\|, \forall \boldsymbol{x}$ |

[1] Only shows the existence of a permutation that guarantees this convergence, and $H = \tilde{\mathcal{O}}(\sqrt{d})$.
[2] Requires $K \gtrsim (1+P)\kappa$.   [3] Requires $K \geq \max\{\kappa^3/n^2, \kappa^{3/2}\}$ and $\kappa \geq n$.

**Definition 2.2** (Strong Convexity). A differentiable function $F : \mathbb{R}^d \to \mathbb{R}$ is $\mu$-*strongly convex*, for some $\mu > 0$, if

$$F(\boldsymbol{y}) \geq F(\boldsymbol{x}) + \langle \nabla F(\boldsymbol{x}), \boldsymbol{y} - \boldsymbol{x} \rangle + \frac{\mu}{2}\|\boldsymbol{y} - \boldsymbol{x}\|^2$$

for all $\boldsymbol{x}, \boldsymbol{y} \in \mathbb{R}^d$. If this inequality holds with $\mu = 0$, we say that $F$ is *convex*.

Now, we define a common assumption on the objective function used in our analyses.

**Assumption 2.3** (Common Assumption). The overall function $F : \mathbb{R}^d \to \mathbb{R}$ is $\mu$-strongly convex and each component function $f_i$ is $L$-smooth.

Additionally, we define the *condition number* of $F$ as $\kappa := \frac{L}{\mu}$, which is closely related to the problem geometry. We note that component smoothness is commonly utilized in the literature studying permutation-based SGD (Ahn et al., 2020; Mishchenko et al., 2020; Lu et al., 2022a; Liu & Zhou, 2024a).

Lastly, we introduce assumptions on the gradients.

**Assumption 2.4** (Bounded Gradient Errors). There exists constants $G \geq 0$ and $P \geq 0$ such that, for all $\boldsymbol{x} \in \mathbb{R}^d$ and $i \in [n]$,

$$\|\nabla f_i(\boldsymbol{x}) - \nabla F(\boldsymbol{x})\| \leq G + P\|\nabla F(\boldsymbol{x})\|.$$

**Assumption 2.5** (Bounded Gradients at the Optimum). There exists a constant $G_* \geq 0$ such that, for all $i \in [n]$, the gradient norm of each component function satisfies

$$\|\nabla f_i(\boldsymbol{x}^*)\| \leq G_*.$$

Our results require either Assumption 2.4 or Assumption 2.5. Notably, whenever Assumption 2.4 holds, Assumption 2.5 also holds with $G_* = G$ because $\nabla F(\boldsymbol{x}^*) = 0$.

## 2.2. Algorithms

---
**Algorithm 1** Permutation-Based SGD

---
**Input:** Initial point $\boldsymbol{x}_0$, Step size $\eta$, Number of epochs $K$
Initialize $\boldsymbol{x}_0^1 = \boldsymbol{x}_0$
**for** $k = 1$ to $K$ **do**
    Generate a permutation $\sigma_k : [n] \to [n]$
    **for** $i = 1$ **to** $n$ **do**
        $\boldsymbol{x}_i^k = \boldsymbol{x}_{i-1}^k - \eta \nabla f_{\sigma_k(i)}(\boldsymbol{x}_{i-1}^k)$
    **end for**
    $\boldsymbol{x}_0^{k+1} = \boldsymbol{x}_n^k$
**end for**
**Output:** $\boldsymbol{x}_n^K$

---

We present the basic pseudocode for permutation-based SGD methods in Algorithm 1. At the start of $k$-th epoch, a permutation $\sigma_k : [n] \to [n]$ is generated. The algorithm then updates the iterate according to the component functions in the order $f_{\sigma_k(1)}, f_{\sigma_k(2)}, \ldots, f_{\sigma_k(n)}$. The method by which the permutation $\sigma_k$ is generated determines the specific variant of permutation-based SGD. Here, we describe some popular methods studied in the literature:

- **Incremental Gradient Descent** (IGD, Algorithm 2): Each $\sigma_k$ is the identity permutation.

- **Single Shuffling** (SS): The first permutation $\sigma_1$ is drawn uniformly at random and reused for all epochs.

- **Random Reshuffling** (RR): Each $\sigma_k$ is independently drawn uniformly at random in every epoch.

- **Gradient Balancing** (GraB (Lu et al., 2022a)): Each $\sigma_k$ is selected based on observations at the previous epoch.

It has been widely studied that the performance guaran-

tees vary drastically with the choice of permutation strategy (Mohtashami et al., 2022; Lu et al., 2022b). In general, one might expect IGD to converge slowly, as the identity mapping could represent the *worst-case* scenario for convergence. In contrast, GraB can be faster than IGD, SS, or RR as it adaptively selects effective permutations over time.

In this paper, we derive upper bound results for *arbitrary* permutation-based SGD, and the lower bound results for IGD. These results allow us to characterize how much the convergence of permutation-based SGD deteriorates when permutations are chosen in the worst-case manner. For further clarification of the relationship between the upper and the lower bound, we point readers to Appendix A.3.

### 2.3. What is known so far?

For simplicity, in this section, we use the conventional symbols $\mathcal{O}(\cdot)$ and $\Omega(\cdot)$ (even for the small epoch regime) to denote upper and lower bounds, respectively. The symbol $\tilde{\mathcal{O}}(\cdot)$ hides the dependency on polylogarithmic factors. Convergence rates are expressed in terms of $n$, $K$, $\mu$, and $L$ to represent the optimality gap of the function.

**Permutation-Based SGD.** Numerous studies have explored the convergence of permutation-based SGD (Bertsekas, 2011; Recht & Ré, 2012; Haochen & Sra, 2019; Nagaraj et al., 2019; Gurbuzbalaban et al., 2019; Safran & Shamir, 2020; 2021; Ahn et al., 2020; Mishchenko et al., 2020; Rajput et al., 2020; 2022; Nguyen et al., 2021; Lu et al., 2022a; Cha et al., 2023; Liu & Zhou, 2024a; Cai & Diakonikolas, 2024). Here, we summarize recent advances in the convergence analysis of the last iterate for strongly convex objectives, and we refer readers to these works for a more comprehensive understanding.

For both RR and SS, under the assumption of component convexity, Mishchenko et al. (2020) derive a convergence rate of $\tilde{\mathcal{O}}(\frac{L^2}{\mu^3 n K^2})$. Later, Liu & Zhou (2024a) improve this result by a factor of $\kappa$, showing a rate of $\tilde{\mathcal{O}}(\frac{L}{\mu^2 n K^2})$. The corresponding lower bounds, $\Omega(\frac{L}{\mu^2 n K^2})$, are established by Cha et al. (2023) for RR and Safran & Shamir (2021) for SS, thereby fully closing the gap between the upper and lower bounds only up to polylogarithmic factors.

There are also several works that derive upper bounds applicable to arbitrary permutation-based SGD, which naturally encompass the convergence of IGD. Under the assumption of component convexity, Liu & Zhou (2024a) establish a rate of $\tilde{\mathcal{O}}(\frac{L}{\mu^2 K^2})$, which is slower than the rate for RR by a factor of $n$. For the matching lower bound, Safran & Shamir (2020) derive a rate of $\Omega(\frac{1}{\mu K^2})$ for IGD, revealing a gap of $\kappa$ between the upper and lower bounds.

Recent research (Rajput et al., 2022; Lu et al., 2022b; Mohtashami et al., 2022) has shifted toward exploring

permutation-based SGD methods that go beyond RR, focusing on manually selecting permutations that induce faster convergence rather than relying on random permutations. A notable work by Lu et al. (2022a) proposes a practical permutation-based SGD algorithm called GraB and provides a theoretical guarantee of convergence at the rate of $\tilde{\mathcal{O}}(\frac{L^2}{\mu^3 n^2 K^2})$—a strictly faster rate than RR. Later, Cha et al. (2023) establishes a matching lower bound, confirming that GraB is optimal (for low-dimensional functions).

We note that most of these works require a condition on $K$ of the form $K \geq \kappa^\alpha \log(nK)$ with $\alpha \geq 1$.

**Small Epoch Analysis.** The convergence behavior of permutation-based in the small epoch regime was first explicitly investigated by Safran & Shamir (2021). They provide both upper and lower bounds for RR and SS in both the small and large epoch regimes, with the rates matching exactly up to polylogarithmic factors for quadratic objectives with commuting component Hessians. Interestingly, in the small epoch regime, both RR and SS achieve a convergence rate of $\Theta(\frac{1}{\mu n K})$, equivalent to the known rate of $\tilde{\mathcal{O}}(\frac{1}{\mu T})$ for with-replacement SGD, where the total number of iterations $T$ can be expressed as $nK$ in the without-replacement setting (Shamir & Zhang, 2013; Liu & Zhou, 2024b).

To the best of our knowledge, no meaningful upper bound result with a rate of $\tilde{\mathcal{O}}(\frac{1}{\mu n K})$ has been established for permutation-based SGD in the small epoch regime. This rate is of significant importance, as it corresponds to the rate for with-replacement SGD and matches the best-known lower bound for RR in this regime (Safran & Shamir, 2021; Cha et al., 2023). The upper bounds provided by Safran & Shamir (2021) for RR and SS are restricted to quadratic objectives with additional assumptions, and therefore, do not differ significantly from the scenario of a 1-dimensional quadratic objective.

Some knowledgeable readers may point to the results of Mishchenko et al. (2020), which present the convergence rates for RR without imposing any constraint on $K$. Specifically, under component convexity, Theorem 2 of Mishchenko et al. (2020) states

$$\mathbb{E}\left[\left\|\boldsymbol{x}_n^K - \boldsymbol{x}^*\right\|^2\right] = \tilde{\mathcal{O}}\left(\exp\left(-\frac{\mu K}{\sqrt{2}L}\right)D^2 + \frac{L}{\mu^3 n K^2}\right),$$

where $D := \|\boldsymbol{x}_0 - \boldsymbol{x}^*\|$. However, we believe that Theorem 2 does not provide a tight bound in the small epoch regime for two reasons. First, the polynomial term induces the function optimality gap of $\tilde{\mathcal{O}}(\frac{L^2}{\mu^3 n K^2})$, which is slower than the lower bound rate for RR by a factor of $\frac{\kappa^2}{K}$. Second, as $K$ decreases below $\kappa$, the exponential term grows rapidly and dominates, deviating substantially from the rate of $\tilde{\mathcal{O}}(\frac{1}{\mu n K})$. While their Theorem 1 improves the exponential term by assuming a strong convexity of components, it leaves the polynomial term unchanged. Also, a more recent

result by Liu & Zhou (2024a) (Theorem 4.6) refines the polynomial term and also improves the exponential term to $\exp(-K/\kappa)\frac{LD^2}{K}$. However, since the term inside the exponential remains unchanged, this still fails to reveal a tight bound when $K$ is small.

While Koloskova et al. (2024) derive an upper bound convergence rate for permutation-based SGD that does not rely on large $K$ for nonconvex objectives, we were unable to extend their proof techniques to the strongly convex setting to yield a rate of $\tilde{\mathcal{O}}(\frac{1}{\mu n K})$.

## 2.4. Why Do Existing Bounds Require Large Epochs?

To understand the challenges in establishing upper bounds for permutation-based SGD, it is important to observe that, unlike with-replacement SGD, permutation-based SGD uses each component function exactly once per epoch. Therefore, $n$ steps of update in permutation-based SGD can be expressed as an approximation of gradient descent on the overall objective, combined with a cumulative error term. Much of the prior literature on establishing the upper bounds for permutation-based SGD focuses on capturing the "cumulative error" effect within a single epoch.

Technically, to show that the cumulative error within an epoch is small, the step size must be sufficiently small to ensure that the iterate does not move too far during a single epoch. Specifically, the step size must be less than $\mathcal{O}(1/nL)$, where $L$ represents the smoothness parameter (Mishchenko et al., 2020; Liu & Zhou, 2024a). However, when the number of epochs $K$ is small, the step size should be larger in order to bring the iterate close to the optimal point. In fact, the step size should be at least as large as a value proportional to $1/K$.

These two requirements—the need for a small step size to control error within a single epoch and the need for a larger step size to achieve fast convergence when $K$ is small—lead to a conflict. Consequently, existing analyses generally hold only when $K$ is sufficiently large. While some analyses are valid even when $K$ is small, their bounds are not tight as discussed in the previous subsection.

## 3. IGD in Small Epoch Regime

We have highlighted that studying permutation-based SGD in the small epoch regime, where the total number of epochs $K$ satisfies $K \lesssim \kappa$, is both underexplored and highly challenging, despite its practical relevance. As an initial step toward understanding its convergence behavior in this regime, we investigate IGD, the simplest and deterministic permutation-based SGD method. We explore this regime under three distinct scenarios: (i) each component is strongly convex with a common Hessian, (ii) each component is strongly convex, and (iii) some components may be noncon-

vex. For each scenario, we establish a convergence lower bound and demonstrate degradation in convergence.

### 3.1. Convergence Analysis of IGD

We introduce our first lower bound result of IGD in the small epoch regime.

**Theorem 3.1.** *For any $n \geq 2$, $\kappa \geq 2$, and $K \leq \frac{1}{2}\kappa$, there exists a $3$-dimensional function $F$ satisfying Assumptions 2.3 and 2.4 with $P = 0$, where each component function shares the same Hessian, i.e., $\nabla^2 f_i(\boldsymbol{x}) = \nabla^2 F(\boldsymbol{x})$ for all $i \in [n]$ and $\boldsymbol{x} \in \mathbb{R}^3$, along with an initialization point $\boldsymbol{x}_0$, such that for any constant step size $\eta$, the final iterate $\boldsymbol{x}_n^K$ obtained by Algorithm 2 satisfies*

$$F(\boldsymbol{x}_n^K) - F(\boldsymbol{x}^*) \gtrsim \frac{G^2}{\mu K}.$$

The proof of Theorem 3.1 is presented in Appendix B.1. Note that if all component functions share the same Hessian, they are also $\mu$-strongly convex. To the best of our knowledge, the previous best lower bound rate for IGD in this setting was $\frac{G^2}{\mu K^2}$ (Safran & Shamir (2020)), and Theorem 3.1 improves it by a factor of $K$. Additionally, RR has a lower bound of $\frac{G^2}{\mu n K}$ (Safran & Shamir (2021)), and the optimal permutation-based SGD method has a lower bound of $\frac{LG^2}{\mu^2 n^2 K^2}$ (Cha et al. (2023)) in the same setting.

For with-replacement SGD, the known upper bound on the function optimality gap is $\frac{G^2}{\mu n K}$ (Liu & Zhou, 2024b), which is faster than the rate $\frac{G^2}{\mu K}$ in Theorem 3.1 by a factor of $n$. We emphasize that this comparison is made under conditions advantageous to IGD, as the lower bound from Theorem 3.1 assumes all component functions share the same Hessian, while the upper bound for with-replacement SGD does not require such a condition. However, this comparison has some subtleties: the upper bound rate is derived under a varying step size scheme, leaving open the possibility that IGD can converge faster under such a scheme. For a more complete comparison, it would be important to extend Theorem 3.1 to the varying step size setting, which we leave for future work.

Next, we present the upper bound for arbitrary permutation-based SGD methods when each component is 1-dimensional and shares the same Hessian.

**Theorem 3.2.** *Let $n \geq 1$, $\frac{\kappa}{n} \lesssim K \leq \kappa$, and an initialization point $x_0$. Suppose $F$ is a $1$-dimensional function satisfying Assumptions 2.3 and 2.5. Assume that each component function $f_i$ shares the same Hessian for all $i \in [n]$ and $x \in \mathbb{R}$. Then, for any choice of permutation $\sigma_k$ in each epoch, the final iterate $x_n^K$ obtained by Algorithm 1 with the step size $\eta = \frac{1}{\mu n K} \max\left\{\log\left(\frac{L|x_0 - x^*|}{G_*}\right), 1\right\}$ satisfies*

$$F(x_n^K) - F(x^*) \lesssim \frac{G_*^2}{\mu K}.$$

The proof of Theorem 3.2 is in Appendix C.1. We note that the minimum epoch requirement $K \gtrsim \frac{\kappa}{n}$ is necessary for valid analysis, as mentioned in Safran & Shamir (2021) (Remark 2). While Theorem 3.2 additionally requires the objective to be 1-dimensional due to technical challenges, the bound can be directly applied to objectives with diagonal Hessians, which aligns with the construction in the proof of Theorem 3.1.

Indeed, the function class Theorems 3.1 and 3.2 apply to is restrictive. However, our next theorem indicates that the convergence of IGD deteriorates immediately when the identical Hessian assumption is removed, even when each component function remains strongly convex.

**Theorem 3.3.** *For any $n \geq 3$, $\kappa \geq 2$, and $K \leq \frac{1}{16\pi}\kappa$, there exists a 4-dimensional function $F$ satisfying Assumptions 2.3 and 2.4 with $P = 1$, where each component function is $\mu$-strongly convex, along with an initialization point $\boldsymbol{x}_0$, such that for any constant step size $\eta$, the final iterate $\boldsymbol{x}_n^K$ obtained by running Algorithm 2 satisfies*

$$F(\boldsymbol{x}_n^K) - F(\boldsymbol{x}^*) \gtrsim \frac{LG^2}{\mu^2} \min\left\{1, \frac{\kappa^2}{K^4}\right\}.$$

Theorem 3.3 is a technically complex result, and we briefly outline the key strategy here. We construct each component function by applying a rotation, positioning each minimizer to form a regular $n$-polygon. The key idea is that, with a carefully chosen initialization, the iterates preserve rotational symmetry and also form a regular $n$-polygon, maintaining a constant distance from the global minimizer throughout the optimization process. The proof of Theorem 3.3 is presented in Appendix B.2.

Compared to Theorem 3.1, Theorem 3.3 provides a consistently larger lower bound. Specifically, depending on the relationship between $K$ and $\sqrt{\kappa}$, the bound in Theorem 3.3 is larger by a factor of either $\kappa K$ or $\kappa^3/K^3$, both exceeding 1 in the small epoch regime. When $K = \Theta(\kappa)$, both bounds in Theorems 3.1 and 3.3 become $\frac{G^2}{L}$.

Theorem 5 of Mishchenko et al. (2020) provides an upper bound for IGD when all component functions are strongly convex. We restate this result in Proposition 3.4, with a slight modification to extend its applicability to arbitrary permutation-based SGD methods.

**Proposition 3.4** (Mishchenko et al. (2020), Theorem 5)**.** *Let $n \geq 1$, $K \gtrsim \frac{\kappa}{n}$, and $\boldsymbol{x}_0$ be the initialization point. Suppose $F$ is a function satisfying Assumptions 2.3 and 2.5 where each component function is $\mu$-strongly convex. Then, for any choice of permutation $\sigma_k$ in each epoch, the final iterate $\boldsymbol{x}_n^K$ obtained by running Algorithm 1 with a step size $\eta = \frac{2}{\mu n K} \max\left\{\log\left(\frac{\|\boldsymbol{x}_0 - \boldsymbol{x}^*\|\mu K}{\sqrt{\kappa}G_*}\right), 1\right\}$, satisfies*

$$\|\boldsymbol{x}_n^K - \boldsymbol{x}^*\|^2 \lesssim \frac{LG_*^2}{\mu^3 K^2}.$$

The proof of Proposition 3.4 is presented in Appendix C.2. The squared distance bound in Proposition 3.4 naturally translates to a function optimality gap of $\frac{L^2 G_*^2}{\mu^3 K^2}$. Although Theorem 3.3 and Proposition 3.4 do not match in general, they do align when $K = \Theta(\sqrt{\kappa})$: in this case, both bounds become the rates $\frac{LG^2}{\mu^2}$ and $\frac{LG_*^2}{\mu^2}$, achieving a tight match up to polylogarithmic factors.

Now, we present the result for the case where nonconvex components exist. While some slowdown in convergence is expected, Theorem 3.5 reveals that it is far more drastic.

**Theorem 3.5.** *For any $n \geq 4$, $\kappa \geq 4$, and $K \leq \frac{\kappa}{4}$, there exists a 2-dimensional function $F$ satisfying Assumptions 2.3 and 2.4 with $P = 3$ such that for any constant step size $\eta$, the final iterate $\boldsymbol{x}_n^K$ obtained by running Algorithm 2 starting from the initialization point $\boldsymbol{x}_0 = (D, 0)$ satisfies*

$$F(\boldsymbol{x}_n^K) - F(\boldsymbol{x}^*) \gtrsim \min\left\{\mu D^2, \frac{G^2}{L}\left(1 + \frac{L}{2\mu n K}\right)^n\right\}.$$

Our construction involves component functions that are concave in particular directions. The proof of Theorem 3.5 is presented in Appendix B.3. One distinction of this statement is the explicit inclusion of the initial distance $D$. This dependence cannot be removed unless the initial point is placed exponentially far from the global minimum, which would lead to an unfair comparison with upper bound theorems, as they typically include a $\log D$ term in their bounds.

Roughly, an expression of the form $(1 + a)^b$ can be approximated by $\exp(ab)$. Applying this to $(1 + \frac{\kappa}{2nK})^n$, we obtain the approximation $\exp(\frac{\kappa}{2K})$. Thus, when $K = \Theta(\kappa)$, the second term scales as $\frac{G^2}{L}$, and as $K$ decreases, it grows at a rate exponential in $\frac{\kappa}{K}$. This contrasts with other bounds, which typically exhibit polynomial dependence on $\frac{\kappa}{K}$.

To validate our findings, we conduct experiments on our lower bound constructions in Appendix G. For Theorem 3.3, we confirm that the iterates follow a circular trajectory, as intended by the original design. For Theorem 3.5, we observe that the function optimality gap for IGD skyrockets whereas other permutation-based SGD methods remain robust in the small epoch regime. To our knowledge, no upper bound exists for RR in this setting with nonconvex components. Based on experimental results for Theorem 3.5, we conjecture that RR will theoretically exhibit robust convergence in this nonconvex component setting, unlike IGD.

While our lower bound results are stated in terms of the function optimality gap to align with the form of upper bounds, our proof can be directly extended to derive lower bounds in terms of the distance to the optimal solution. Specifically, the lower bounds on the distance metric $\|\boldsymbol{x}_n^K - \boldsymbol{x}^*\|$ are:

$\frac{G}{\mu K}$, $\frac{G}{\mu} \cdot \min\left\{1, \frac{\kappa}{K^2}\right\}$, and $\min\left\{D, \frac{G}{L}\left(1 + \frac{L}{2\mu n K}\right)^{n/2}\right\}$

for Theorems 3.1, 3.3 and 3.5, respectively.

## 3.2. Breaking the Barrier of With-Replacement SGD

Up to this point, we have analyzed the worst-case convergence behavior of permutation-based SGD with respect to the permutation choice in the small epoch regime. A natural question that follows is: what happens in the best case?

Unlike the large epoch regime where the question has been sufficiently explored (Lu et al., 2022a; Cha et al., 2023), the small epoch regime remains less understood. In this section, we slightly deviate from the main topic and demonstrate that a well-designed permutation can enable permutation-based SGD to achieve a faster convergence rate than with-replacement SGD in the small epoch regime—which RR has been proven not to do so (Safran & Shamir, 2021).

Before presenting our finding, we introduce an additional lemma that is used in deriving our result.

**Lemma 3.6** (Herding Algorithm (Bansal & Garg, 2017)). *Let $z_1, \cdots, z_n \in \mathbb{R}^d$ satisfy $\|z_i\| \leq 1$ for all $i \in [n]$ and $\sum_{i=1}^{n} z_i = 0$. Then, there exists an algorithm,* Herding, *that outputs a permutation $\sigma : [n] \to [n]$ such that*

$$\max_{i \in [n]} \left\| \sum_{j=1}^{i} z_{\sigma(j)} \right\| \leq H, \text{ where } H = \tilde{\mathcal{O}}(\sqrt{d}).$$

The Herding algorithm was used in Lu et al. (2022a) for designing GraB. Our next theorem leverages Herding in a different way to show the existence of a permutation-based SGD method (but impractical) that achieves acceleration even in the small epoch regime.

**Theorem 3.7** (Herding at Optimum). *Let $n \geq 1$, $K \gtrsim \frac{\kappa}{n}$, and $x_0$ be the initialization point. Suppose $F$ is a function satisfying Assumptions 2.3 and 2.5 where each component function is $\mu$-strongly convex. Then, there exists a permutation $\sigma$ such that the final iterate $x_n^K$ obtained by running Algorithm 1 with $K$ epochs of $\sigma$ and a step size $\eta = \frac{2}{\mu n K} \max \left\{ \log \left( \frac{\|x_0 - x^*\| \mu n K}{\sqrt{\kappa} H G_*} \right), 1 \right\}$, satisfies*

$$\left\| x_n^K - x^* \right\|^2 \lesssim \frac{H^2 L G_*^2}{\mu^3 n^2 K^2}.$$

Unlike GraB which dynamically adapts the permutation at each epoch based on the gradient observations, Theorem 3.7 applies a fixed $\sigma$ consistently throughout entire epochs. The permutation $\sigma$ is obtained by running Herding for the scaled component gradients at the global optimum $x^*$, ensuring $\max_{i \in [n]} \| \sum_{j=1}^{i} \nabla f_{\sigma(j)}(x^*) \| \leq H G_*$. The proof of Theorem 3.7 is presented in Appendix C.3.

By $L$-smoothness, it immediately follows that the function optimality gap is bounded as $F(x_n^K) - F(x^*) \lesssim \frac{H^2 L^2 G_*^2}{\mu^3 n^2 K^2}$. We make two key observations regarding this result. First, Cha et al. (2023) prove the lower bound rate of $\frac{L G^2}{\mu^2 n^2 K^2}$ applicable to arbitrary permutation-based SGD without any constraint on $K$. This confirms that Theorem 3.7 achieves opti-

mal performance in terms of $n$ and $K$ among permutation-based SGD methods. Second, this rate outperforms the rate of $\frac{G^2}{\mu n K}$ for with-replacement SGD (Liu & Zhou, 2024b) whenever $n \geq H^2 \kappa^2 / K$. In particular, even when $K \lesssim \kappa$, problems involving a large number of component functions with small input dimensions can still satisfy this condition.

To our knowledge, this is the first result showing that a permutation-based SGD method may outperform with-replacement SGD in the small epoch regime. However, we identify two key limitations. First, Theorem 3.7 is not an implementable algorithm, as it requires prior knowledge of component gradients at $x^*$. Second, the upper bound in Theorem 3.7 and the lower bound established by Cha et al. (2023) still differ by a factor of $H^2 \kappa$. An interesting future direction would be to design a practical permutation-based SGD method that tightly matches this lower bound.

We conclude by suggesting a setting where we can efficiently obtain this permutation. Suppose all component functions have the same Hessian so that $\nabla^2 f_i - \nabla^2 F \equiv 0$. Then, the gradient difference $\nabla f_i - \nabla F$ remains constant across the domain, leading to the following equation:

$$\nabla f_i(x^*) = \nabla f_i(x^*) - \nabla F(x^*) = \nabla f_i(x_0) - \nabla F(x_0).$$

In this scenario, we can use scaled gradient errors at the initialization $(\nabla f_i(x_0) - \nabla F(x_0))/G_*$, which can be efficiently obtained, as inputs to Herding to attain the desired permutation $\sigma$. Furthermore, the lower bound construction of Cha et al. (2023), which achieves a rate of $\frac{L G^2}{\mu^2 n^2 K^2}$, also satisfies the identical Hessian assumption. This confirms the algorithmically optimal convergence for this specific function class, up to a factor of $H^2 \kappa$ gap.

# 4. IGD in Large Epoch Regime

We now shift focus to the *large epoch regime*, where $K \gtrsim \kappa$. We examine convergence under two distinct scenarios: (i) each component is convex, and (ii) some components may be nonconvex. While the presence of nonconvex components significantly deteriorates convergence in the small epoch regime, we observe that this effect diminishes in the large epoch regime.

## 4.1. Convergence with Component Convexity

We first focus on the case where all components are convex.

**Theorem 4.1.** *For any $n \geq 2$, $\kappa \geq 2$, and $K \geq \kappa$, there exists a 3-dimensional function $F$ satisfying Assumptions 2.3 and 2.4 with $P = 0$, where each component function shares the same Hessian, along with an initialization point $x_0$, such that for any constant step size $\eta$, the final iterate obtained by running Algorithm 2 satisfies*

$$F(x_n^K) - F(x^*) \gtrsim \frac{L G^2}{\mu^2 K^2}.$$

The detailed proof of the theorem is provided in Appendix D.1. As previously discussed in Theorem 3.1, since the overall function is strongly convex and each component function shares the same Hessian, it follows that each component function is also $\mu$-strongly convex. The previous best lower bound for IGD in this setting was $\frac{G^2}{\mu K^2}$ (Safran & Shamir, 2020), and our result improves upon this by a factor of $\kappa$. Also, when $K = \Theta(\kappa)$, this bound simplifies to $\frac{G^2}{L}$, thereby continuously interpolating the lower bound results in the small epoch regime (Theorems 3.1, 3.3 and 3.5).

Next, we present a complementary upper bound result, originally established in Theorem 4.6 of Liu & Zhou (2024a). For consistency with the assumptions used throughout this paper, we restate it under slightly stronger assumptions.

**Proposition 4.2** (Liu & Zhou (2024a), Theorem 4.6). *Let $n \geq 1$, $K \gtrsim \kappa$, and $\boldsymbol{x}_0$ be the initialization point. Suppose $F$ is a function satisfying Assumptions 2.3 and 2.5 where each component function is convex. Then, for any choice of permutation $\sigma_k$ in each epoch, the final iterate obtained by Algorithm 1 with the step size $\eta = \frac{1}{\mu n K} \max\left\{\log\left(\frac{\|\boldsymbol{x}_0 - \boldsymbol{x}^*\|^2 \mu^3 K^2}{LG^2(1 + \log K)}\right), 1\right\}$ satisfies*

$$F(\boldsymbol{x}_n^K) - F(\boldsymbol{x}^*) \lesssim \frac{LG_*^2}{\mu^2 K^2}.$$

We observe that the lower bound in Theorem 4.1 and the upper bound in Proposition 4.2 match exactly, up to polylogarithmic factors. The component functions for the lower bound satisfy strictly stronger assumptions than those required for the upper bound. Unlike upper bounds where stronger assumptions may improve the convergence rate, fulfilling stronger assumptions in lower bound analyses rather strengthens the result of the bound. Thus, Theorem 4.1 remains a valid lower bound matching Proposition 4.2.

### 4.2. Convergence without Component Convexity

In this section, we investigate the case where the assumption of component convexity is removed. Our next theorem, Theorem 4.3, establishes a lower bound for IGD, quantifying the degradation in convergence rate when nonconvex components are included in the large epoch setting.

**Theorem 4.3.** *For any $n \geq 4$, $\kappa \geq n$, and $K \geq \max\left\{\kappa^3/n^2, \kappa^{3/2}\right\}$, there exists a 4-dimensional function $F$ satisfying Assumptions 2.3 and 2.4 with $P = \kappa$, along with an initialization point $\boldsymbol{x}_0$, such that for any constant step size $\eta$, the final iterate obtained by running Algorithm 2 satisfies*

$$F(\boldsymbol{x}_n^K) - F(\boldsymbol{x}^*) \gtrsim \frac{L^2 G^2}{\mu^3 K^2}.$$

The proof of Theorem 4.3 is presented in Appendix D.2. Additional assumptions on $n$ and $K$ are introduced for technical reasons. Since the construction in Theorem 4.3 involves

nonconvex components, Proposition 4.2 is no longer applicable for direct comparison. Theorem 4.4 addresses this by providing an upper bound allowing nonconvex component functions for arbitrary permutation-based SGD.

**Theorem 4.4.** *Let $n \geq 1$, $K \gtrsim (1 + P)\kappa$, and $\boldsymbol{x}_0$ be the initialization point. Suppose $F$ is a function satisfying Assumptions 2.3 and 2.4. Then, for any choice of permutation $\sigma_k$ in each epoch, the final iterate $\boldsymbol{x}_n^K$ obtained by Algorithm 1 with a step size $\eta = \frac{2}{\mu n K} \max\left\{\log\left(\frac{(F(\boldsymbol{x}_0) - F(\boldsymbol{x}^*))\mu^3 K^2}{L^2 G^2}\right), 1\right\}$ satisfies*

$$F(\boldsymbol{x}_n^K) - F(\boldsymbol{x}^*) \lesssim \frac{L^2 G^2}{\mu^3 K^2}.$$

The proof of Theorem 4.4 is in Appendix E.1. This upper bound aligns with the lower bound in Theorem 4.3, differing only by polylogarithmic factors, when the objective is sufficiently ill-conditioned and the number of epochs $K$ is sufficiently large, specifically $K \gtrsim \max\left\{\kappa^3/n^2, \kappa^2\right\}$.

Importantly, the convergence rate in this setting degrades by only a factor of $\kappa$ compared to the convex components case. These results highlight an *intriguing* behavior of IGD: allowing nonconvex components significantly degrades convergence in the small epoch regime; however, this slowdown is much less severe in the large epoch regime.

Similar to the small epoch case, the lower bounds in the large epoch regime can also be expressed in terms of the distance to the optimum. Specifically, the lower bounds on $\|\boldsymbol{x}_n^K - \boldsymbol{x}^*\|$ are $\frac{G}{\mu K}$ and $\frac{LG}{\mu^2 K}$ (for $K \geq \max\{\kappa^3/n^2, \kappa^2\}$) for Theorems 4.1 and 4.3, respectively.

### 4.3. Comparison with Other Methods

Here, we provide a detailed comparison of the convergence rates across different permutation-based SGD methods.

**Random Reshuffling.** In the large epoch regime, Liu & Zhou (2024a) show that RR achieves an upper bound of $\frac{LG_*^2}{\mu^2 n K^2}$, while Cha et al. (2023) establish a tight matching lower bound under the same setting. Both results assume that the component functions are convex. This implies that in settings where all component functions are convex, RR outperforms IGD by a factor of $n$ in terms of convergence rate in the large epoch regime.

**Optimal Permutation-based SGD.** Lu et al. (2022a) demonstrate that GraB achieves an upper bound of $\frac{H^2 L^2 G^2}{\mu^3 n^2 K^2}$, where $H$ is a constant that scales as $\sqrt{d}$ (Lemma 3.6). Similarly, Cha et al. (2023) establish a lower bound of $\frac{L^2 G^2}{\mu^3 n^2 K^2}$, for any permutation strategy over $K$ epochs, assuming sufficiently ill-conditioned problems and a large number of epochs. Both results are derived without assuming component convexity. Together, these results indicate that, when nonconvex components exist and $d$ is fixed, the optimal con-

vergence rate for permutation-based SGD in the large epoch regime is $\frac{L^2 G^2}{\mu^3 n^2 K^2}$. This implies that in settings where some components are nonconvex, IGD converges at a rate slower than optimal permutation-based SGD by a factor of $n^2$.

## 5. Conclusion

We provide a detailed analysis of IGD across both small and large epoch regimes, considering various assumptions on the component functions. Our results show that, unlike in the large epoch regime, even when the component functions are strongly convex, the convergence can be significantly slow. Furthermore, the presence of nonconvex components exacerbates this slowdown exponentially. We also demonstrate the existence of a permutation-based SGD method that allows faster convergence in the small epoch regime.

Finally, we highlight two promising directions for future work. The first is to establish a tight convergence bound for RR in the small epoch regime, similar to our analysis for IGD in Section 3. We discuss the current state of research and the key challenges in this direction in Appendix A.4. The second is to develop an efficient and practical permutation-based SGD method that enjoys provable fast convergence in this regime.

## Acknowledgements

This work was partly supported by a National Research Foundation of Korea (NRF) grant funded by the Korean government (MSIT) (No. RS-2023-00211352) and an Institute for Information & communications Technology Planning & Evaluation (IITP) grant funded by the Korean government (MSIT) (No. RS-2019-II190075, Artificial Intelligence Graduate School Program (KAIST)). CY acknowledges support from a grant funded by Samsung Electronics Co., Ltd.

## Impact Statement

This paper aims to advance the theoretical understanding of convex optimization in machine learning. While optimization methods have broad applications, we do not foresee any specific ethical concerns or societal implications arising directly from this work.

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

# A. Supplementary Details

In this section, we provide additional details omitted from the main text.

## A.1. Visualization of Upper and Lower Bounds

We begin by presenting a plot that summarizes our theoretical findings. Theorems 3.1 to 3.5 (and the result from Mishchenko et al. (2020)) apply to the small epoch regime ($K \lesssim \kappa$), and Theorems 4.1 to 4.4 (and the result from Liu & Zhou (2024a)) apply to the large epoch regime ($K \gtrsim \kappa$). In the figure, solid lines indicate upper bounds and dash-dot lines represent lower bounds. Each color represents a pair of upper and lower bounds derived under similar assumptions—what we refer to as *matching bounds*. The vertical line at $K = \kappa$ marks the transition between the small and large epoch regimes. Both axes are log-scaled for better visualization of rate differences.

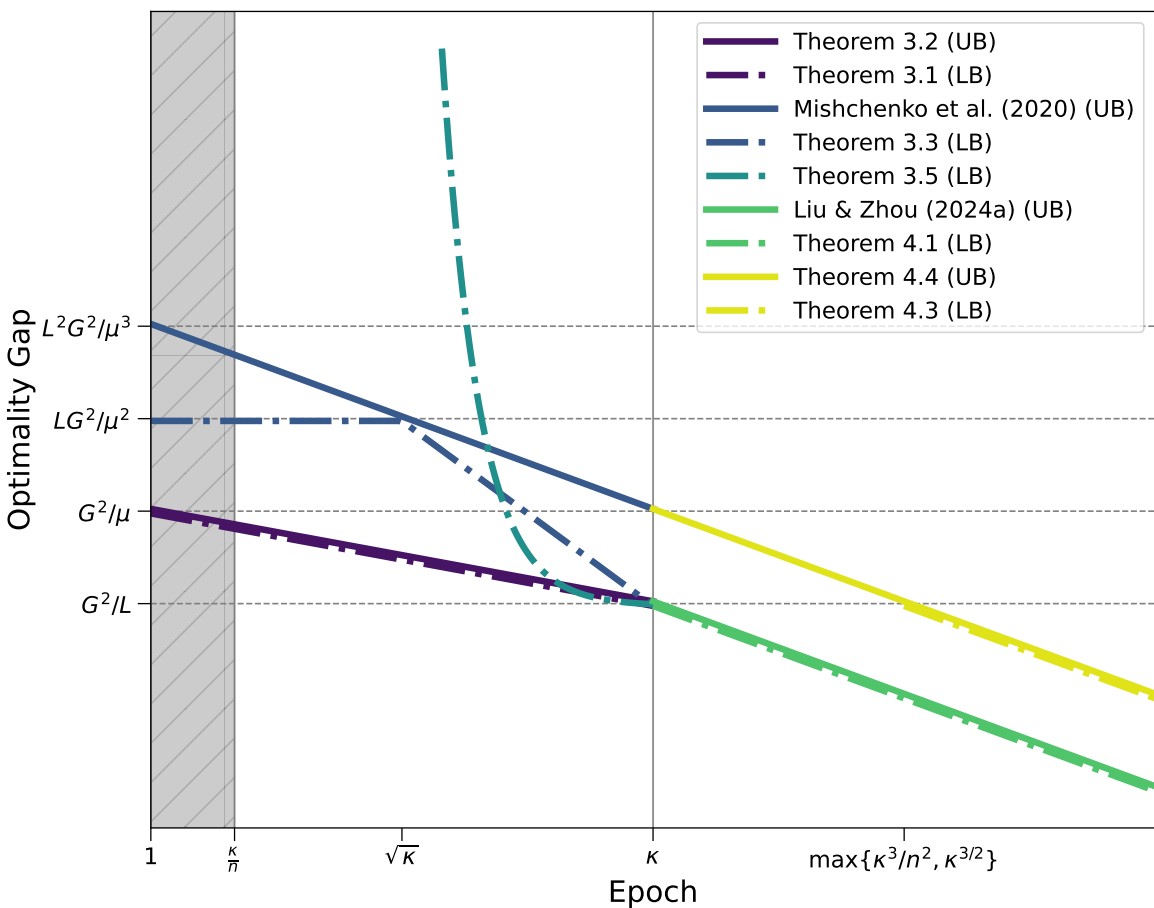

*Figure 1.* Visualization of the bounds in Table 1. Both axes are log-scaled. Upper bounds (UB) are represented using a solid line, and lower bounds (LB) are depicted with a dash-dot line. The small and large epoch results are combined into a single figure with a separation by the vertical line $K = \kappa$. Upper bound results for the small epoch regime only hold under $K \gtrsim \kappa/n$, while lower bound results hold for $K$ greater than some constant.

## A.2. Pseudeocode of IGD

Next, we provide the pseudocode for IGD as Algorithm 2.

---
**Algorithm 2** Incremental Gradient Descent
---

**Input:** Initial point $\boldsymbol{x}_0$, Step size $\eta$, Number of epochs $K$
Initialize $\boldsymbol{x}_0^1 = \boldsymbol{x}_0$
**for** $k = 1$ to $K$ **do**
    **for** $i = 1$ **to** $n$ **do**
        $\boldsymbol{x}_i^k = \boldsymbol{x}_{i-1}^k - \eta \nabla f_i(\boldsymbol{x}_{i-1}^k)$
    **end for**
    $\boldsymbol{x}_0^{k+1} = \boldsymbol{x}_n^k$
**end for**
**Output:** $\boldsymbol{x}_n^K$

---

### A.3. Connecting IGD Lower Bounds with Upper Bounds of General Permutation-Based SGD

In this paper, we derive upper bound results for *arbitrary* permutation-based SGD, and the lower bound results for IGD. To clarify the connection between these results, we explain why the lower bounds derived for IGD are relevant to the upper bounds for arbitrary permutation-based SGD. Intuitively, deriving the upper bound for arbitrary permutation-based SGD can be viewed as bounding the following inf-sup problem from above:

$$\inf_{\substack{\text{step size } \eta}} \sup_{\substack{\text{function } F(\boldsymbol{x}) \\ \text{permutation } \{\sigma_k\}_{k=1}^K}} F(\boldsymbol{x}_n^K) - F(\boldsymbol{x}^*). \tag{1}$$

On the other hand, the corresponding lower bound is one that bounds the following sup-inf problem from below:

$$\sup_{\substack{\text{function } F(\boldsymbol{x}) \\ \text{permutation } \{\sigma_k\}_{k=1}^K}} \inf_{\substack{\text{step size } \eta}} F(\boldsymbol{x}_n^K) - F(\boldsymbol{x}^*). \tag{2}$$

Notably, in the lower bound formulation, the permutations $\{\sigma_k\}_{k=1}^K$ appear in the supremum term. This implies that the lower bound for IGD, which can be formulated as:

$$\sup_{\substack{\text{function } F(\boldsymbol{x})}} \inf_{\substack{\text{step size } \eta}} F(\boldsymbol{x}_n^K) - F(\boldsymbol{x}^*), \tag{3}$$

where every $\sigma_k$ is an identity mapping, is at most equation (2). Therefore, our lower bound results, derived specifically for IGD, also provide valid lower bounds for the upper bound results established for any permutation-based SGD.

To further clarify, we compare it with the work of Lu et al. (2022a). In Lu et al. (2022a), the authors introduce a permutation-based SGD algorithm called GraB that provably converges faster by carefully selecting permutations at each epoch. This problem can be formulated as bounding the following inf-sup problem:

$$\inf_{\substack{\text{step size } \eta \\ \text{permutation } \{\sigma_k\}_{k=1}^K}} \sup_{\substack{\text{function } F(\boldsymbol{x})}} F(\boldsymbol{x}_n^K) - F(\boldsymbol{x}^*). \tag{4}$$

In addition, Cha et al. (2023) proves that GraB is an optimal permutation-based SGD by providing a lower bound that holds for every possible combination of permutations over $K$ epochs:

$$\sup_{\substack{\text{function } F(\boldsymbol{x})}} \inf_{\substack{\text{step size } \eta \\ \text{permutation } \{\sigma_k\}_{k=1}^K}} F(\boldsymbol{x}_n^K) - F(\boldsymbol{x}^*). \tag{5}$$

Clearly, equation (4) and equation (5) are smaller than equation (1) and equation (2), respectively.

### A.4. Status and Open Challenges in Establishing Tight Bounds for RR in the Small Epoch Regime

We begin by summarizing the current state of research on RR in the small epoch regime. To the best of our knowledge, there are two noteworthy results (under the assumption that the overall function is strongly convex and each component is smooth):

1. (Mishchenko et al., 2020): When all component functions are also strongly convex, an upper bound of $\tilde{\mathcal{O}}(\frac{L^2}{\mu^3 n K^2})$ is provided.

2. (Safran & Shamir, 2021): When all component functions are quadratic and their Hessians commute, a tight convergence rate of $\Theta(\frac{1}{\mu n K})$ is established.

Unlike scenario (2) where the authors provide matching UB and LB (up to polylogarithmic factor), the lower bound in scenario (1) is unknown, and it remains open whether the rate $\tilde{\mathcal{O}}(\frac{L^2}{\mu^3 n K^2})$ can be improved or not.

Given this context, there are two clear directions for future exploration in small epoch RR literature:

- **Upper Bound Direction.** Improve the existing bound of $\tilde{\mathcal{O}}(\frac{L^2}{\mu^3 n K^2})$ under the strongly convex component assumption, or derive new bounds under weaker assumptions (e.g., convexity, or even without convexity).

- **Lower Bound Direction.** Develop a matching lower bound (under the strongly convex component case) to close the gap with the existing upper bound $\tilde{\mathcal{O}}(\frac{L^2}{\mu^3 n K^2})$.

The primary challenge on the upper bound side is that deriving new upper bounds in the small epoch regime appears to require sophisticated analytical techniques (due to challenges discussed in Section 2.4). As can be found in Safran & Shamir (2021), even the proof for 1D quadratic is highly technical. One promising technique we explored is from Koloskova et al. (2024). In contrast to traditional analyses that group updates within a single epoch (i.e., chunks of size $n$), this method groups updates into chunks of size $\tau := 1/\eta L$. While this chunk-based approach can be successfully applied to derive upper bounds for IGD, it becomes problematic for RR. Specifically, when the chunk size $\tau$ does not align neatly within epochs, handling the dependencies between iterates becomes extremely difficult.

Regarding the lower bound direction, we believe any progress beyond current results will likely require more complicated constructions that go beyond simple quadratic functions. This is because for simple quadratic functions where the Hessians commute with each other (e.g., $f_i(x_1, x_2) = \frac{L}{2} x_1^2 + a_i x_1 + \frac{\mu}{2} x_2^2 + b_i x_2$), the tight rate of $\Theta(\frac{1}{\mu n K})$ is already established by Safran & Shamir (2021). Therefore, to surpass the existing LB barrier $\Omega(\frac{1}{\mu n K})$, future constructions must involve quadratic functions with non-commuting Hessians or even non-quadratic functions, necessitating more advanced analytical techniques. While our own lower bound construction in Theorem 3.3 is based on quadratic functions with non-commuting Hessians, it is tailored to IGD, and we do not see a clear way to extend this idea to RR.

# B. Proofs for Small Epoch Lower Bounds

In this section, we present the detailed proofs for Theorems 3.1, 3.3 and 3.5 which are the lower bound results in the small epoch regime. To establish these results, we construct a specific function $F$ that achieves the stated lower bound for each theorem. We note that constructing a lower-bound function for SGD presents a significant challenge, as it must exhibit poor convergence for any choice of step size $\eta$. The difficulty lies in the fact that the convergence behavior of SGD is highly sensitive to $\eta$: a small step size leads to slow updates, whereas a large step size can cause divergence.

To overcome this challenge, we partition the positive real line of possible step sizes into three regimes: small, moderate, and large. For each regime, we design a distinct lower-bound function tailored to follow the stated convergence behavior within that range. Finally, we combine these functions across dimensions, ensuring that the resulting function satisfies the stated lower bound for any choice of $\eta$. This "dimension-aggregating" technique has been developed in the recent literature (e.g., Safran & Shamir (2021); Yun et al. (2022); Cha et al. (2023)).

## B.1. Proof of Theorem 3.1

**Theorem 3.1.** *For any $n \geq 2$, $\kappa \geq 2$, and $K \leq \frac{1}{2}\kappa$, there exists a 3-dimensional function $F$ satisfying Assumptions 2.3 and 2.4 with $P = 0$, where each component function shares the same Hessian, i.e., $\nabla^2 f_i(\boldsymbol{x}) = \nabla^2 F(\boldsymbol{x})$ for all $i \in [n]$ and $\boldsymbol{x} \in \mathbb{R}^3$, along with an initialization point $\boldsymbol{x}_0$, such that for any constant step size $\eta$, the final iterate $\boldsymbol{x}_n^K$ obtained by Algorithm 2 satisfies*

$$F(\boldsymbol{x}_n^K) - F(\boldsymbol{x}^*) \gtrsim \frac{G^2}{\mu K}.$$

*Proof.* We divide the range of step sizes $\eta > 0$ into three regimes that will be specified subsequently. For each regime, we construct the overall functions $F_1$, $F_2$, and $F_3$ respectively, along with their respective component functions and an initial point. Each function is 1-dimensional and satisfies Assumption 2.3. $F_1$ and $F_3$ satisfy Assumption 2.4 with $G = P = 0$, and $F_2$ satisfies with $P = 0$. Also, the component functions within each overall function share the same Hessian. Importantly, each function is designed to satisfy the following properties:

- (Small step size regime) There exists an initialization point $x_0 = \text{poly}(\mu, L, n, K, G)$ such that for any choice of $\eta \in \left(0, \frac{1}{\mu n K}\right)$, the final iterate $x_n^K$ obtained by running Algorithm 2 satisfies $F_1(x_n^K) - F_1(x^*) \gtrsim \frac{G^2}{\mu K}$.

- (Moderate step size regime) There exists an initialization point $y_0 = \text{poly}(\mu, L, n, K, G)$ such that for any choice of $\eta \in \left[\frac{1}{\mu n K}, \frac{2}{L}\right)$, the final iterate $y_n^K$ obtained by running Algorithm 2 satisfies $F_2(y_n^K) - F_2(y^*) \gtrsim \frac{G^2}{\mu K}$.

- (Large step size regime) There exists an initialization point $z_0 = \text{poly}(\mu, L, n, K, G)$ such that for any choice of $\eta \in \left[\frac{2}{L}, \infty\right)$, the final iterate $z_n^K$ obtained by running Algorithm 2 satisfies $F_3(z_n^K) - F_3(z^*) \gtrsim \frac{G^2}{\mu K}$.

Here, $x^*$, $y^*$, $z^*$ denote the minimizers of $F_1$, $F_2$, and $F_3$, respectively. Detailed constructions of $F_1$, $F_2$, and $F_3$, as well as the verification of the assumptions and the stated properties are presented in Appendices B.1.1 to B.1.3.

We now aggregate these functions across dimensions: $F(\boldsymbol{x}) := F(x, y, z) = F_1(x) + F_2(y) + F_3(z)$ and $f_i(\boldsymbol{x}) = f_{1i}(x) + f_{2i}(y) + f_{3i}(z)$ for all $i \in [n]$. Here, $f_i, f_{1i}, f_{1i}, f_{3i}$ denote the $i$-th component function of $F$, $F_1$, $F_2$, and $F_3$, respectively. Since each dimension is independent, it is obvious that $\boldsymbol{x}^* = (x^*, y^*, z^*)$ minimizes $F$.

Finally, by choosing the initialization point as $\boldsymbol{x}_0 = (x_0, y_0, z_0)$, the final iterate $\boldsymbol{x}_n^K = (x_n^K, y_n^K, z_n^K)$ obtained by running Algorithm 2 on $F$ satisfies

$$F(\boldsymbol{x}_n^K) - F(\boldsymbol{x}^*) \gtrsim \frac{G^2}{\mu K},$$

regardless of the choice of $\eta > 0$.

Note that $F$ satisfies the stated assumptions as

$$\mu\boldsymbol{I} \preceq \min\{\nabla^2 F_1(x), \nabla^2 F_2(y), \nabla^2 F_3(z)\} \preceq \nabla^2 F(\boldsymbol{x}) \preceq \max\{\nabla^2 F_1(x), \nabla^2 F_2(y), \nabla^2 F_3(z)\} \preceq L\boldsymbol{I},$$

and

$$\|\nabla f_i(\boldsymbol{x}) - \nabla F(\boldsymbol{x})\| \leq \|\nabla f_{1i}(x) - \nabla F_1(x)\| + \|\nabla f_{2i}(y) - \nabla F_2(y)\| + \|\nabla f_{3i}(z) - \nabla F_3(z)\| \leq 0 + G + 0 = G.$$

Moreover, $\nabla^2 f_i(\boldsymbol{x}) = \mathrm{diag}(\nabla^2 f_{1i}(x), \nabla^2 f_{2i}(y), \nabla^2 f_{3i}(z)) = \mathrm{diag}(\nabla^2 f_i(x), \nabla^2 f_i(y), \nabla^2 f_i(z)) = \nabla^2 F(\boldsymbol{x})$ holds, since the component functions within each overall function share the same diagonal Hessian. This concludes the proof of Theorem 3.1. $\qquad \square$

In the following subsections, we present the specific construction of $F_1$, $F_2$, and $F_3$, and demonstrate that each satisfies the stated lower bound within its corresponding step size regime. For simplicity of notation, we omit the index of the overall function when referring to its component functions, e.g., we write $f_i(x)$ instead of $f_{1i}(x)$. Moreover, we use the common variable notation $x$ while constructing functions for each dimension, though we use different variables in the "dimension-aggregation" step.

### B.1.1. CONSTRUCTION OF $F_1$

Let $F_1(x) = \frac{\mu}{2}x^2$ with component functions $f_i(x) = F_1(x)$ for all $i \in [n]$. It is clear that $F_1$ satisfies Assumption 2.3, Assumption 2.4 with $P = 0$, and its component functions share an identical Hessian. Also, we note that $x^* = 0$ and $F_1(x^*) = 0$.

Let the initialization be $x_0 = \frac{G}{\mu\sqrt{K}}$. For all $\eta \in \left(0, \frac{1}{\mu n K}\right)$, the final iterate is given by

$$x_n^K = (1 - \eta\mu)^{nK} x_0 \geq \left(1 - \frac{1}{nK}\right)^{nK} x_0 \geq \frac{G}{4\mu\sqrt{K}},$$

where the last inequality uses the fact that $(1 - \frac{1}{m})^m \geq \frac{1}{4}$ for all $m \geq 2$.

Thus, we have

$$F_1(x_n^K) - F_1(x^*) = \frac{\mu}{2}(x_n^K)^2 \gtrsim \frac{G^2}{\mu K}.$$

### B.1.2. CONSTRUCTION OF $F_2$

We construct the function by dividing the cases by the parity of $n$. We first consider the case where $n$ is even, and address the case where $n$ is odd later in this subsection. Let $F_2(x) = \frac{\mu K}{2}x^2$ with component functions

$$f_i(x) = \begin{cases} \frac{\mu K}{2}x^2 + Gx & \text{if } i \leq n/2, \\ \frac{\mu K}{2}x^2 - Gx & \text{otherwise.} \end{cases}$$

It is clear that $f_i$ satisfies Assumption 2.4 with $P = 0$ and shares the same Hessian. From the assumption $K \leq \frac{1}{2}\kappa$, we have $\mu \leq \mu K \leq \frac{L}{2}$. Hence, each $f_i$ is $L$-smooth and $\mu$-strongly convex. Also, we note that $x^* = 0$ and $F_2(x^*) = 0$.

By Lemma F.1, the final iterate obtained by running Algorithm 2 is given by

$$x_n^K = (1 - \eta\mu K)^{nK} x_0 + \frac{G}{\mu K} \cdot \frac{1 - (1 - \eta\mu K)^{\frac{n}{2}}}{1 + (1 - \eta\mu K)^{\frac{n}{2}}} \left(1 - (1 - \eta\mu K)^{nK}\right). \tag{6}$$

For any $\eta \in \left[\frac{1}{\mu n K}, \frac{2}{L}\right)$, it follows that $\frac{1}{n} \leq \eta\mu K < \frac{2\mu K}{L} = \frac{2K}{\kappa} \leq 1$. Then, we have $(1 - \eta\mu K)^{nK} \leq \left(1 - \frac{1}{n}\right)^{nK} \leq e^{-K} \leq e^{-1}$ which implies $1 - (1 - \eta\mu K)^{nK} \geq 1 - e^{-1}$. Moreover, we have $(1 - \eta\mu K)^{\frac{n}{2}} \leq (1 - \frac{1}{n})^{\frac{n}{2}} \leq e^{-\frac{1}{2}}$ and thus,

$$\frac{1 - (1 - \eta\mu K)^{\frac{n}{2}}}{1 + (1 - \eta\mu K)^{\frac{n}{2}}} \geq \frac{1 - e^{-\frac{1}{2}}}{2}.$$

Substituting these inequalities into equation (6) and setting $x_0 = 0$, we obtain

$$x_n^K \geq \frac{\left(1 - e^{-1}\right)\left(1 - e^{-\frac{1}{2}}\right)G}{2\mu K},$$

and

$$F_2(x_n^K) - F_2(x^*) = \frac{\mu K}{2}(x_n^K)^2 \gtrsim \frac{G^2}{\mu K}.$$

We now consider the case where $n$ is odd. Let $F_2(x) = \frac{\mu K}{2}x^2$ with component functions

$$f_i(x) = \begin{cases} \frac{\mu K}{2}x^2 & \text{if } i = 1, \\ \frac{\mu K}{2}x^2 + Gx & \text{if } 2 \le i \le (n+1)/2, \\ \frac{\mu K}{2}x^2 - Gx & \text{if } (n+3)/2 \le i \le n. \end{cases}$$

Compared to the case of even $n$, $f_1(x) = \frac{\mu K}{2}x^2$ is introduced newly. It is clear that $f_i$ satisfies Assumption 2.4 with $P = 0$ and shares the same Hessian. From the assumption $K \le \frac{1}{2}\kappa$, we have $\mu \le \mu K \le \frac{L}{2}$. Hence, each $f_i$ is $L$-smooth and $\mu$-strongly convex. Also, we note that $x^* = 0$ and $F_2(x^*) = 0$.

By Lemma F.2, the final iterate obtained by running Algorithm 2 is given by

$$x_n^K = (1 - \eta\mu K)^{nK}x_0 + \frac{G}{\mu K} \cdot \frac{1 - (1 - \eta\mu K)^{nK}}{1 - (1 - \eta\mu K)^n}\left(1 - (1 - \eta\mu K)^{\frac{n-1}{2}}\right)^2. \tag{7}$$

For any $\eta \in \left[\frac{1}{\mu nK}, \frac{2}{L}\right)$, it follows that $\frac{1}{n} \le \eta\mu K < \frac{2\mu K}{L} = \frac{2K}{\kappa} \le 1$. Then, we have $(1 - \eta\mu K)^{nK} \le \left(1 - \frac{1}{n}\right)^{nK} \le e^{-K} \le e^{-1}$. Moreover, $(1 - \eta\mu K)^{\frac{n-1}{2}} \le \left(1 - \frac{1}{n}\right)^{\frac{n-1}{2}} \le e^{-\frac{n-1}{2n}} \le e^{-\frac{1}{4}}$ holds for $n \ge 2$. Substituting these inequalities into equation (7) and setting $x_0 = 0$, we have

$$x_n^K \ge \frac{G}{\mu K} \cdot \frac{1 - e^{-1}}{1}(1 - e^{-\frac{1}{4}})^2.$$

Thus, we obtain the following optimality gap:

$$F_2(x_n^K) - F_2(x^*) = \frac{\mu K}{2}(x_n^K)^2 \gtrsim \frac{G^2}{\mu K}.$$

### B.1.3. CONSTRUCTION OF $F_3$

Let $F_3(x) = \frac{L}{2}x^2$ with component functions $f_i(x) = F_3(x)$ for all $i \in [n]$. It is clear that $F_1$ satisfies Assumption 2.3, Assumption 2.4 with $P = 0$, and its component functions share an identical Hessian. Also, we note that $x^* = 0$ and $F_3(x^*) = 0$.

For all $\eta \in \left[\frac{2}{L}, \infty\right)$, the final iterate is given by

$$x_n^K = (1 - \eta L)^{nK}x_0.$$

In this regime, the step size is excessively large, resulting in

$$1 - \eta L \le 1 - \frac{2}{L} \cdot L \le -1,$$

which implies $\left|(1 - \eta L)^{nK}\right| \ge 1$. Thus, the iterate does not converge and satisfies $\left|x_n^K\right| \ge |x_0|$.

By setting the initialization $x_0 = \frac{G}{\sqrt{\mu L K}}$, we have

$$F_3(x_n^K) - F_3(x^*) = \frac{L}{2}(x_n^K)^2 \ge \frac{L}{2}(x_0)^2 \gtrsim \frac{G^2}{\mu K}.$$

## B.2. Proof of Theorem 3.3

**Theorem 3.3.** *For any $n \geq 3$, $\kappa \geq 2$, and $K \leq \frac{1}{16\pi}\kappa$, there exists a 4-dimensional function $F$ satisfying Assumptions 2.3 and 2.4 with $P = 1$, where each component function is $\mu$-strongly convex, along with an initialization point $\boldsymbol{x}_0$, such that for any constant step size $\eta$, the final iterate $\boldsymbol{x}_n^K$ obtained by running Algorithm 2 satisfies*

$$F(\boldsymbol{x}_n^K) - F(\boldsymbol{x}^*) \gtrsim \frac{LG^2}{\mu^2} \min\left\{1, \frac{\kappa^2}{K^4}\right\}.$$

*Proof.* Similar to the approach in Theorem 3.1, we divide the range of step sizes into three regimes. For each regime, we construct the overall functions $F_1$, $F_2$, and $F_3$ respectively, along with their respective component functions and an initial point. Finally, we aggregate these functions across different dimensions to derive the stated lower bound.

The functions $F_1$ and $F_3$ are 1-dimensional, and $F_2$ is a 2-dimensional function. Each function is carefully designed to satisfy the following properties:

- (Small step size regime) There exists an initialization point $x_0 = \text{poly}(\mu, L, n, K, G)$ such that for any choice of $\eta \in \left(0, \frac{1}{\mu n K}\right)$, the final iterate $x_n^K$ obtained by running Algorithm 2 satisfies $F_1(x_n^K) - F_1(x^*) \gtrsim \frac{LG^2}{\mu^2} \min\left\{1, \frac{\kappa^2}{K^4}\right\}$.

- (Moderate step size regime) There exists an initialization point $(y_0, z_0) = \text{poly}(\mu, L, n, K, G)$ such that for any choice of $\eta \in \left[\frac{1}{\mu n K}, \frac{2}{L}\right)$, the final iterate $(y_n^K, z_n^K)$ obtained by running Algorithm 2 satisfies $F_2(y_n^K, z_n^K) - F_2(y^*, z^*) \gtrsim \frac{LG^2}{\mu^2} \min\left\{1, \frac{\kappa^2}{K^4}\right\}$.

- (Large step size regime) There exists an initialization point $w_0 = \text{poly}(\mu, L, n, K, G)$ such that for any choice of $\eta \in \left[\frac{2}{L}, \infty\right)$, the final iterate $w_n^K$ obtained by running Algorithm 2 satisfies $F_3(w_n^K) - F_3(w^*) \gtrsim \frac{LG^2}{\mu^2} \min\left\{1, \frac{\kappa^2}{K^4}\right\}$.

Here, $x^*$, $(y^*, z^*)$, and $w^*$ denote the minimizers of $F_1$, $F_2$, and $F_3$, respectively. All these functions are designed to satisfy Assumption 2.3. $F_1$ and $F_3$ satisfy Assumption 2.4 with $G = P = 0$, and $F_2$ satisfies with $P = 1$. Moreover, each component function within each overall function is $\mu$-strongly convex. Detailed constructions of $F_1$, $F_2$, and $F_3$, as well as the verification of the assumptions and the stated properties are presented in Appendices B.2.1 to B.2.3.

By following a similar approach to the proof of Theorem 3.1, we can conclude that the aggregated 4-dimensional function $F(\boldsymbol{x}) := F(x, y, z, w) = F_1(x) + F_2(y, z) + F_3(w)$ and its component functions satisfy Assumption 2.3. Additionally,

$$\begin{aligned}
\|\nabla f_i(\boldsymbol{x}) - \nabla F(\boldsymbol{x})\| &\leq \|\nabla f_{1i}(x) - \nabla F_1(x)\| + \|\nabla f_{2i}(y) - \nabla F_2(y)\| + \|\nabla f_{3i}(z) - \nabla F_3(z)\| \\
&\leq 0 + (G + \|\nabla F_2(y)\|) + 0 \leq G + \|\nabla F(\boldsymbol{x})\|,
\end{aligned}$$

thus satisfying Assumption 2.4 with $P = 1$. Also, since each dimension is independent, it is obvious that $\boldsymbol{x}^* = (x^*, y^*, z^*, w^*)$ minimizes $F$. Moreover, by choosing the initialization point as $\boldsymbol{x}_0 = (x_0, y_0, z_0, w_0)$, the final iterate $\boldsymbol{x}_n^K = (x_n^K, y_n^K, z_n^K, w_n^K)$ obtained by running Algorithm 2 on $F$ satisfies

$$F(\boldsymbol{x}_n^K) - F(\boldsymbol{x}^*) \gtrsim \frac{LG^2}{\mu^2} \min\left\{1, \frac{\kappa^2}{K^4}\right\},$$

regardless of the choice of $\eta > 0$.

This concludes the proof of Theorem 3.3. $\qquad\square$

In the following subsections, we present the specific construction of $F_1$, $F_2$, and $F_3$, and demonstrate that each satisfies the stated lower bound within its corresponding step size regime. For simplicity of notation, we omit the index of the overall function when referring to its component functions, e.g., we write $f_i(x)$ instead of $f_{1i}(x)$. Moreover, we use the common variable notation $x$ (and $y$) while constructing functions for each dimension, though we use different variables in the "dimension-aggregation" step.

### B.2.1. CONSTRUCTION OF $F_1$

Let $F_1(x) = \frac{\mu}{2}x^2$ with component functions $f_i(x) = F_1(x)$ for all $i \in [n]$. It is clear that $F_1$ satisfies Assumption 2.3, Assumption 2.4 with $G = P = 0$, and has $\mu$-strongly convex component functions. Also, we note that $x^* = 0$ and $F_1(x^*) = 0$.

Let the initialization be $x_0 = \sqrt{\kappa} \min\left\{1, \frac{\kappa}{K^2}\right\} \frac{G}{\mu}$. For all $\eta \in \left(0, \frac{1}{\mu n K}\right)$, the final iterate is given by

$$x_n^K = (1 - \eta\mu)^{nK} x_0 \geq \left(1 - \frac{1}{nK}\right)^{nK} x_0 \geq \sqrt{\kappa} \min\left\{1, \frac{\kappa}{K^2}\right\} \frac{G}{4\mu},$$

where the last inequality uses the fact that $(1 - \frac{1}{m})^m \geq \frac{1}{4}$ for all $m \geq 2$.

Thus, we have

$$F_1(x_n^K) - F_1(x^*) = \frac{\mu}{2}(x_n^K)^2 \gtrsim \frac{LG^2}{\mu^2} \min\left\{1, \frac{\kappa^2}{K^4}\right\}.$$

### B.2.2. CONSTRUCTION OF $F_2$

In this subsection, we let $L'$ denote $L/2$. We introduce the design of each component function as follows:

$$f_i(x, y) = \begin{cases} \frac{\mu}{2}x^2 + \frac{L'}{2}y^2 - Gx & \text{if } i = 1, \\ \left(f_1 \circ (R_{i-1})^{-1}\right)(x, y) & \text{if } 2 \leq i \leq n, \end{cases}$$

where

$$R_i := \begin{bmatrix} \cos\theta_i & -\sin\theta_i \\ \sin\theta_i & \cos\theta_i \end{bmatrix}$$

is the matrix for the counter-clock wise rotation in $\mathbb{R}^2$ by an angle $\theta_i = i\delta$ with $\delta := \frac{2\pi}{n}$.

Using these component functions, the overall function $F_2 := \frac{1}{n}\sum_{i=1}^n f_i$ is given by $\frac{\mu+L'}{4}(x^2 + y^2)$. This result can be verified by expanding the closed form of $f_i$:

$$\begin{aligned} f_i(x, y) &= \frac{\mu}{2}(x\cos\theta_{i-1} + y\sin\theta_{i-1})^2 + \frac{L'}{2}(-x\sin\theta_{i-1} + y\cos\theta_{i-1})^2 - G(x\cos\theta_{i-1} + y\sin\theta_{i-1}) \\ &= \frac{1}{2}\left(\mu\cos^2\theta_{i-1} + L'\sin^2\theta_{i-1}\right)x^2 + \frac{1}{2}\left(\mu\sin^2\theta_{i-1} + L'\cos^2\theta_{i-1}\right)y^2 \\ &\quad + (\mu - L')\sin\theta_{i-1}\cos\theta_{i-1}xy - G(x\cos\theta_{i-1} + y\sin\theta_{i-1}). \end{aligned}$$

Since $n \geq 3$, we can utilize Lemmas B.1 and B.2, and obtain

$$\frac{1}{n}\sum_{i=1}^n \sin\theta_{i-1} = \frac{1}{n}\sum_{i=1}^n \cos\theta_{i-1} = 0,$$

$$\frac{1}{n}\sum_{i=1}^n \sin^2\theta_{i-1} = \frac{1}{n}\sum_{i=1}^n \cos^2\theta_{i-1} = \frac{1}{2},$$

$$\frac{1}{n}\sum_{i=1}^n \sin\theta_{i-1}\cos\theta_{i-1} = \frac{1}{2n}\sum_{i=1}^n \sin\theta_{2(i-1)} = 0.$$

Using these results, the overall function is simplified to

$$F_2(x, y) = \frac{\mu + L'}{4}(x^2 + y^2),$$

which has a minimizer $(x^*, y^*) = (0, 0)$.

Note that each component function $f_i$ is obtained by rotating $f_1$, and hence $f_i$ inherits the properties of $f_1$. We can easily check that $f_1$ is both $\mu$-strongly convex and $L$-smooth. Also, the gradient difference between the component function $f_1$ and the overall function $F_2$ can be expressed as

$$
\begin{aligned}
\|\nabla f_1(x,y) - \nabla F_2(x,y)\| &= \left\| \left( (\mu x - G) - \frac{\mu + L'}{2}x, Ly - \frac{\mu + L'}{2}y \right) \right\| \\
&= \left\| \left( \frac{\mu - L'}{2}x - G, \frac{L' - \mu}{2}y \right) \right\| \\
&\leq G + \left\| \left( \frac{\mu - L'}{2}x, \frac{L' - \mu}{2}y \right) \right\| \\
&= G + \left\| \left( \frac{L' - \mu}{2}x, \frac{L' - \mu}{2}y \right) \right\| \\
&\leq G + \left\| \left( \frac{L' + \mu}{2}x, \frac{L' + \mu}{2}y \right) \right\| = G + \|\nabla F_2(x,y)\|,
\end{aligned}
$$

proving that the construction satisfies Assumption 2.4 with $P = 1$.

Before delving into the detailed proof, we outline the intuition for the construction. We start by designing a step-size-dependent initialization point $(u_0(\eta), v_0(\eta))$, where $\eta \in \left[ \frac{1}{\mu n K}, \frac{2}{L} \right)$. For $i \in [n]$, we define $(u_i(\eta), v_i(\eta))$ as the result of running a single step of gradient descent on $f_i$ with a step size $\eta$, starting from $(u_{i-1}(\eta), v_{i-1}(\eta))$.

The key idea is to carefully design $(u_0(\eta), v_0(\eta))$ so that each subsequent iterate $(u_i(\eta), v_i(\eta))$ is obtained by rotating $(u_{i-1}(\eta), v_{i-1}(\eta))$ by an angle $\delta = \frac{2\pi}{n}$. This aligns with our construction of the component functions $f_i$, which are also generated by continually rotating $f_1$ by the same angle $\delta$. As a result, the relative position between each iterate and the component function used to compute the next iterate is preserved throughout the entire update process. This symmetry ensures that the trajectory of the iterates $(u_i(\eta), v_i(\eta))$ forms a regular $n$-sided polygon. Consequently, after running Algorithm 2, the final iterate and the initialization point $(u_0(\eta), v_0(\eta))$ are identical.

At the last step of the proof, we will show that the choice of the initialization point $(u_0(\frac{1}{\mu n K}), v_0(\frac{1}{\mu n K}))$ can be made in a step-size-independent manner without significantly affecting the final optimality gap, even when the step size $\eta$ is chosen from $\left( \frac{1}{\mu n K}, \frac{2}{L} \right)$ rather than being fixed at $\frac{1}{\mu n K}$.

We now proceed to describe the exact construction of $(u_0(\eta), v_0(\eta))$. Consider the gradient of the component function $f_1(x,y)$, which is given by:

$$
\nabla_x f_1(x,y) = \mu x - G \text{ and } \nabla_y f_1(x,y) = L'y.
$$

A single iteration of gradient descent on $f_1$ using the step size $\eta$ yields:

$$
\begin{aligned}
u_1(\eta) &= u_0(\eta) - \eta(\mu u_0(\eta) - G), \\
v_1(\eta) &= v_0(\eta) - \eta L' v_0(\eta).
\end{aligned}
\tag{8}
$$

To maintain the rotational relationship between successive iterates, we require that the updated iterate $(u_1(\eta), v_1(\eta))$ satisfies the following relationship:

$$
\begin{bmatrix} u_1(\eta) \\ v_1(\eta) \end{bmatrix} = \begin{bmatrix} \cos\delta & -\sin\delta \\ \sin\delta & \cos\delta \end{bmatrix} \begin{bmatrix} u_0(\eta) \\ v_0(\eta) \end{bmatrix}.
\tag{9}
$$

As mentioned earlier, this rotational relationship ensures that the trajectory of the iterates forms a regular $n$-sided polygon. Recall that the component function $f_i$ is defined as:

$$
f_i(x,y) = \left( f_1 \circ (R_{i-1})^{-1} \right)(x,y).
$$

Additionally, let $\Lambda = \begin{bmatrix} \mu & 0 \\ 0 & L \end{bmatrix}$. Since

$$
\nabla f_1(x,y) = \Lambda \begin{bmatrix} x \\ y \end{bmatrix} - G \begin{bmatrix} 1 \\ 0 \end{bmatrix},
$$

the gradient of $f_i$ can then be expressed as

$$\nabla f_i(x, y) = R_{i-1} \left( \Lambda (R_{i-1})^{-1} \begin{bmatrix} x \\ y \end{bmatrix} - G \begin{bmatrix} 1 \\ 0 \end{bmatrix} \right). \tag{10}$$

If $(u_1(\eta), v_1(\eta))$ satisfies both equation (8) and equation (9), then the subsequent iterate $(u_2(\eta), v_2(\eta))$ satisfies:

$$
\begin{aligned}
\begin{bmatrix} u_2(\eta) \\ v_2(\eta) \end{bmatrix} &= \left( \boldsymbol{I} - \eta R_1 \Lambda (R_1)^{-1} \right) \begin{bmatrix} u_1(\eta) \\ v_1(\eta) \end{bmatrix} + \eta G R_1 \begin{bmatrix} 1 \\ 0 \end{bmatrix} \\
&= \left( \boldsymbol{I} - \eta R_1 \Lambda (R_1)^{-1} \right) R_1 \begin{bmatrix} u_0(\eta) \\ v_0(\eta) \end{bmatrix} + \eta G R_1 \begin{bmatrix} 1 \\ 0 \end{bmatrix} \\
&= R_1 \left( (\boldsymbol{I} - \eta \Lambda) \begin{bmatrix} u_0(\eta) \\ v_0(\eta) \end{bmatrix} + \eta G \begin{bmatrix} 1 \\ 0 \end{bmatrix} \right) = R_1 \begin{bmatrix} u_1(\eta) \\ v_1(\eta) \end{bmatrix} = R_2 \begin{bmatrix} u_0(\eta) \\ v_0(\eta) \end{bmatrix}.
\end{aligned}
$$

Thus, if the initial point $(u_0(\eta), v_0(\eta))$ and its successive iterate $(u_1(\eta), v_1(\eta))$ satisfies the rotational relationship, this relation persists throughout the entire update process. Consequently, the trajectory of the iterates forms a regular $n$-sided polygon.

To enforce this rotational relationship, we solve the following system of equations:

$$
\begin{aligned}
u_0(\eta) \cos \delta - v_0(\eta) \sin \delta &= (1 - \eta \mu) u_0(\eta) + \eta G, \\
u_0(\eta) \sin \delta + v_0(\eta) \cos \delta &= (1 - \eta L') v_0(\eta).
\end{aligned}
$$

From these, we derive:

$$u_0(\eta) = \frac{\eta L' - (1 - \cos \delta)}{(1 - \cos \delta)(2 - (\mu + L')\eta) + \eta^2 \mu L'} \cdot \eta G, \tag{11}$$

$$v_0(\eta) = -\frac{\sin \delta}{(1 - \cos \delta)(2 - (\mu + L')\eta) + \eta^2 \mu L'} \cdot \eta G. \tag{12}$$

Note that the numerator of $u_0(\eta)$ is positive as shown below:

$$
\begin{aligned}
\eta L' - (1 - \cos \delta) &\overset{(a)}{\geq} \frac{\eta L'}{2} + \frac{L'}{2\mu n K} - \frac{\delta^2}{2} \\
&\overset{(b)}{=} \frac{\eta L'}{2} + \frac{L}{4\mu n K} - \frac{2\pi^2}{n^2} \\
&= \frac{\eta L'}{2} + \frac{nL - 8\pi^2 \mu K}{4\mu n^2 K} \\
&\overset{(c)}{>} \frac{\eta L'}{2}, \tag{13}
\end{aligned}
$$

where we apply $\eta \geq \frac{1}{\mu n K}$ and the inequality $1 - \cos \theta \leq \frac{\theta^2}{2}$ at $(a)$, substitute $L' = \frac{L}{2}$ and $\delta = \frac{2\pi}{n}$ at $(b)$, and apply $n \geq 3 > \frac{\pi}{2}$ and $\kappa \geq 16\pi K$ at $(c)$. Also, we have $2 - (\mu + L')\eta \geq 0$ for $\eta < \frac{2}{L}$. Thus, $u_0(\eta)$ is always positive and $v_0(\eta)$ is always negative for $\eta \in \left[ \frac{1}{\mu n K}, \frac{2}{L} \right)$.

Let $(u_i^k(\eta), v_i^k(\eta))$ denote the $i$-th iterate at the $k$-th epoch of Algorithm 2 using the step size $\eta$ and the initialization point $(u_0(\eta), v_0(\eta))$. By definition, $(u_n^K(\eta), v_n^K(\eta))$ represents the final iterate after $K$ epochs. Due to rotational symmetry, $(u_n^K(\eta), v_n^K(\eta))$ is identical to $(u_0(\eta), v_0(\eta))$. Thus, the distance between the final iterate and the minimizer of $F_2$ can be lower bounded by the $x$-coordinate of the initialization point:

$$\left\| (u_n^K(\eta), v_n^K(\eta)) \right\| = \| (u_0(\eta), v_0(\eta)) \| \geq u_0(\eta).$$

We now derive a lower bound for $u_0(\eta)$. We begin by upper bounding its denominator:

$$(1 - \cos\delta)(2 - (\mu + L')\eta) + \eta^2\mu L' < \frac{\delta^2}{2} \cdot 2 + \eta^2\mu L' = \frac{4\pi^2}{n^2} + \eta^2\mu L'.$$

Substituting this result into equation (11) gives

$$u_0(\eta) \geq \frac{(\eta L' - (1 - \cos\delta))\eta G}{\frac{4\pi^2}{n^2} + \eta^2\mu L'} > \frac{\eta^2 L' G}{\frac{8\pi^2}{n^2} + 2\eta^2\mu L'} := \varphi(\eta),$$

where we apply equation (13) at the second inequality. We can easily check that $\varphi(\eta)$ is an increasing function of $\eta$ and thus the minimum value is attained at $\eta = \frac{1}{\mu n K}$. Substituting $\eta = \frac{1}{\mu n K}$, we have

$$\varphi(\frac{1}{\mu n K}) = \frac{L'G}{\frac{8\pi^2}{n^2} \cdot \mu^2 n^2 K^2 + 2\mu L'}$$

$$= \frac{LG}{16\pi^2\mu^2 K^2 + 2\mu L} \quad (\because L' = \frac{L}{2})$$

$$= \frac{G}{\mu} \cdot \frac{\frac{L}{\mu K^2}}{16\pi^2 + \frac{2L}{\mu K^2}}.$$

In summary, the distance between the final iterate and the minimizer of $F_2$ is bounded as:

$$\left\|(u_n^K(\eta), v_n^K(\eta))\right\| = \left\|(u_0(\eta), v_0(\eta))\right\| \geq u_0(\eta) \geq \varphi(\eta) \geq \varphi(\frac{1}{\mu n K}) \overset{(a)}{\geq} \frac{G}{32\pi^2\mu}\min\left\{1, \frac{\kappa}{K^2}\right\}, \tag{14}$$

where $(a)$ is derived through the following process:

$$\varphi(\frac{1}{\mu n K}) = \frac{G}{\mu}\frac{\frac{L}{\mu K^2}}{16\pi^2 + \frac{2L}{\mu K^2}} = \frac{G}{2\mu}\frac{\frac{\kappa}{8\pi^2 K^2}}{1 + \frac{\kappa}{8\pi^2 K^2}} \overset{(b)}{\geq} \frac{G}{4\mu}\min\left\{1, \frac{\kappa}{8\pi^2 K^2}\right\} \geq \frac{G}{32\pi^2\mu}\min\left\{1, \frac{\kappa}{K^2}\right\}.$$

Here, we use the inequality $\frac{u}{1+u} \geq \frac{1}{2}\min\{1, u\}$ for all $u \geq 0$ at $(b)$. The function optimality gap can then be bounded as:

$$F_2(u_n^K(\eta), v_n^K(\eta)) - F_2(x^*, y^*) = \frac{\mu + L'}{4}\left\|(u_n^K(\eta), v_n^K(\eta))\right\|^2 \gtrsim \frac{LG^2}{\mu^2}\min\left\{1, \frac{\kappa^2}{K^4}\right\},$$

for $\eta \in \left[\frac{1}{\mu n K}, \frac{2}{L}\right)$. However, one caveat is that the initialization point $(u_0(\eta), v_0(\eta))$ depends on the choice of $\eta$. Our goal is to identify a unified, step-size-independent initialization point $(x_0, y_0)$ that achieves the same lower bound (up to a scaling factor). Specifically, we aim to ensure:

$$F_2(x_n^K, y_n^K) \gtrsim \frac{LG}{\mu^2}\min\left\{1, \frac{\kappa^2}{K^4}\right\}$$

for any choice of $\eta \in \left[\frac{1}{\mu n K}, \frac{2}{L}\right)$.

We claim that this goal can be achieved by setting the initialization point as $(x_0, y_0) = (u_0(\frac{1}{\mu n K}), v_0(\frac{1}{\mu n K}))$. To prove this claim, consider two sequences of iterates: $\{(x_i^k, y_i^k)\}_{i\in[n], k\in[K]}$ and $\{(u_i^k(\eta), v_i^k(\eta))\}_{i\in[n], k\in[K]}$. Both sequences are generated using the same permutation $\mathrm{id}_n$ and the same step size $\eta$, but they differ in their initialization points. Specifically, $(x_i^k, y_i^k)$ starts from the initial point $(u_0(\frac{1}{\mu n K}), v_0(\frac{1}{\mu n K}))$, while $(u_i^k(\eta), v_i^k(\eta))$ starts from the initial point $(u_0(\eta), v_0(\eta))$.

Recall the gradient of $f_i$ from equation (10):

$$\nabla f_i(x, y) = R_{i-1}\left(\Lambda(R_{i-1})^{-1}\begin{bmatrix} x \\ y \end{bmatrix} - G\begin{bmatrix} 1 \\ 0 \end{bmatrix}\right).$$

The update rule for the iterates generated by IGD is:

$$\begin{bmatrix} x_i^k \\ y_i^k \end{bmatrix} = \left(\boldsymbol{I} - \eta R_{i-1}\Lambda(R_{i-1})^{-1}\right)\begin{bmatrix} x_{i-1}^k \\ y_{i-1}^k \end{bmatrix} + \eta G R_{i-1}\begin{bmatrix} 1 \\ 0 \end{bmatrix} \text{ and}$$

$$\begin{bmatrix} u_i^k(\eta) \\ v_i^k(\eta) \end{bmatrix} = \left(\boldsymbol{I} - \eta R_{i-1}\Lambda(R_{i-1})^{-1}\right)\begin{bmatrix} u_{i-1}^k(\eta) \\ v_{i-1}^k(\eta) \end{bmatrix} + \eta G R_{i-1}\begin{bmatrix} 1 \\ 0 \end{bmatrix}.$$

Taking the difference between the two sequences of iterates, we have:

$$\begin{bmatrix} x_i^k - u_i^k(\eta) \\ y_i^k - v_i^k(\eta) \end{bmatrix} = \left(\boldsymbol{I} - \eta R_{i-1}\Lambda(R_{i-1})^{-1}\right)\begin{bmatrix} x_{i-1}^k - u_{i-1}^k(\eta) \\ y_{i-1}^k - v_{i-1}^k(\eta) \end{bmatrix}.$$

Since $R_{i-1}$ is an unitary matrix, it follows that $R_{i-1}\Lambda(R_{i-1})^{-1} \succeq \mu I$. Thus, we obtain the inequality $\boldsymbol{I} - \eta R_{i-1}\Lambda R_{i-1}^{-1} \preceq (1-\eta\mu)I$, which leads to the following bound:

$$\left\|(x_i^k - u_i^k(\eta), y_i^k - v_i^k(\eta))\right\| \leq (1-\eta\mu)\left\|(x_{i-1}^k - u_{i-1}^k(\eta), y_{i-1}^k - v_{i-1}^k(\eta))\right\|.$$

Based on this inequality, we will demonstrate that $\left\|(x_n^K - u_n^K(\eta), y_n^K - v_n^K(\eta))\right\|$ is not significant. This can be interpreted as the gap between the initialization points shrinking progressively throughout the optimization process. Specifically, we have

$$
\begin{aligned}
\left\|(x_n^K - u_n^K(\eta), y_n^K - v_n^K(\eta))\right\| &\leq (1-\eta\mu)^{nK}\left\|(x_0 - u_0(\eta), y_0 - v_0(\eta))\right\| \\
&\leq e^{-\eta\mu nK}\left\|(x_0 - u_0(\eta), y_0 - v_0(\eta))\right\| \\
&\overset{(a)}{\leq} e^{-1}\left\|(u_0(\tfrac{1}{\mu nK}) - u_0(\eta), v_0(\tfrac{1}{\mu nK}) - v_0(\eta))\right\| \\
&\leq e^{-1}\left(\left|u_0(\tfrac{1}{\mu nK}) - u_0(\eta)\right| + \left|v_0(\tfrac{1}{\mu nK}) - v_0(\eta)\right|\right) \\
&\overset{(b)}{\leq} e^{-1}\left(u_0(\eta) + \left|v_0(\tfrac{1}{\mu nK})\right| + |v_0(\eta)|\right) \\
&\overset{(c)}{\leq} e^{-1}\left(u_0(\eta) + \frac{8\pi K}{\kappa}u_0(\tfrac{1}{\mu nK}) + \frac{8\pi K}{\kappa}u_0(\eta)\right) \\
&\overset{(d)}{\leq} \frac{1 + \frac{16\pi K}{\kappa}}{e}u_0(\eta) \\
&\overset{(e)}{\leq} \frac{2}{e}u_0(\eta) \\
&\overset{(f)}{\leq} \frac{2}{e}\left\|(u_n^K(\eta), v_n^K(\eta))\right\|.
\end{aligned}
$$

Here, we apply:

- $(a)$: $\eta \geq \frac{1}{\mu nK}$ and $(x_0, y_0) = \left(u_0\left(\frac{1}{\mu nK}\right), v_0\left(\frac{1}{\mu nK}\right)\right)$.

- $(b)$, $(d)$: $u_0(\eta)$ is positive and increasing (shown in Lemma B.3).

- $(c)$: Follows from Lemma B.4.

- $(e)$: $K \leq \frac{\kappa}{16\pi}$.

- $(f)$: $u_0(\eta) = u_n^K(\eta)$.

In summary, the distance between $(x_n^K, y_n^K)$ and the minimizer of $F_2$ can be bounded as:

$$\left\|(x_n^K, y_n^K)\right\| \geq \left\|(u_n^K(\eta), v_n^K(\eta))\right\| - \left\|(x_n^K - u_n^K(\eta), y_n^K - v_n^K(\eta))\right\| \geq \left(1 - \frac{2}{e}\right)\left\|(u_n^K(\eta), v_n^K(\eta))\right\|,$$

where we have already derived that $\left\|(u_n^K(\eta), v_n^K(\eta))\right\| \geq \frac{G}{32\pi^2\mu} \min\left\{1, \frac{\kappa}{K^2}\right\}$ in equation (14). Thus, we conclude

$$\left\|(x_n^K, y_n^K)\right\| \geq \frac{(1 - 2e^{-1})\,G}{32\pi^2\mu} \min\left\{1, \frac{\kappa}{K^2}\right\}$$

and consequently,

$$F_2(x_n^K, y_n^K) - F_2(x^*, y^*) = \frac{L' + \mu}{4} \left\|(x_n^K, y_n^K)\right\|^2 \geq \frac{L}{8} \left\|(x_n^K, y_n^K)\right\|^2 \gtrsim \frac{LG^2}{\mu^2} \min\left\{1, \frac{\kappa^2}{K^4}\right\}.$$

### B.2.3. CONSTRUCTION OF $F_3$

Let $F_3(x) = \frac{L}{2}x^2$ with component functions $f_i(x) = F_3(x)$ for all $i \in [n]$. It is clear that $F_1$ satisfies Assumption 2.3, Assumption 2.4 with $G = P = 0$ and has $\mu$-strongly convex component functions. Also, we note that $x^* = 0$ and $F_3(x^*) = 0$.

For all $\eta \in \left[\frac{2}{L}, \infty\right)$, the final iterate is given by

$$x_n^K = (1 - \eta L)^{nK} x_0.$$

In this regime, the step size is excessively large, resulting in

$$1 - \eta L \leq 1 - \frac{2}{L} \cdot L \leq -1,$$

which implies $\left|(1 - \eta L)^{nK}\right| \geq 1$. Thus, the iterate does not converge and satisfies $\left|x_n^K\right| \geq |x_0|$.

By setting the initialization $x_0 = \frac{G}{\mu} \min\left\{1, \frac{\kappa}{K^2}\right\}$, we have

$$F_3(x_n^K) - F_3(x^*) = \frac{L}{2}(x_n^K)^2 \geq \frac{L}{2}(x_0)^2 \gtrsim \frac{LG^2}{\mu^2} \min\left\{1, \frac{\kappa^2}{K^4}\right\}.$$

### B.3. Proof of Theorem 3.5

**Theorem 3.5.** *For any $n \geq 4$, $\kappa \geq 4$, and $K \leq \frac{\kappa}{4}$, there exists a 2-dimensional function $F$ satisfying Assumptions 2.3 and 2.4 with $P = 3$ such that for any constant step size $\eta$, the final iterate $\boldsymbol{x}_n^K$ obtained by running Algorithm 2 starting from the initialization point $\boldsymbol{x}_0 = (D, 0)$ satisfies*

$$F(\boldsymbol{x}_n^K) - F(\boldsymbol{x}^*) \gtrsim \min\left\{\mu D^2, \frac{G^2}{L}\left(1 + \frac{L}{2\mu nK}\right)^n\right\}.$$

*Proof.* Similar to the approach in Theorem 3.1, we divide the range of step sizes into two regimes. For each regime, we construct the overall functions $F_1$ and $F_2$, respectively, along with their respective component functions and an initial point. Finally, we aggregate these functions across different dimensions to derive the stated lower bound.

Each function is 1-dimensional and carefully designed to satisfy the following properties:

- (Small step size regime) For any choice of the initialization point $x_0 = D$ and step size $\eta \in \left(0, \frac{1}{\mu nK}\right)$, the final iterate $x_n^K$ obtained by running Algorithm 2 satisfies $F_1(x_n^K) - F_1(x^*) \gtrsim \mu D^2$.

- (Moderate & Large step size regime) There exists an initialization point $y_0 = \text{poly}(\mu, L, n, K, G)$ such that for any choice of $\eta \in \left[\frac{1}{\mu nK}, \infty\right)$, the final iterate $y_n^K$ obtained by running Algorithm 2 satisfies $F_2(y_n^K) - F_2(y^*) \gtrsim \frac{G^2}{L}\left(1 + \frac{L}{2\mu nK}\right)^n$.

Here, $x^*$ and $y^*$ denote the minimizer of $F_1$ and $F_2$, respectively. Both functions are designed to satisfy Assumption 2.3. $F_1$ satisfies Assumption 2.4 with $G = P = 0$, and $F_2$ satisfies with $P = 3$. Detailed constructions of $F_1$ and $F_2$, as well as the verification of the assumptions and the stated properties are presented in Appendices B.3.1 and B.3.2.

By following a similar approach to the proof of Theorems 3.1 and 3.3, we can conclude that the aggregated 2-dimensional function $F(\boldsymbol{x}) := F(x, y) = F_1(x) + F_2(y)$ and its component functions satisfy the stated assumptions. Also, since each dimension is independent, it is obvious that $\boldsymbol{x}^* = (x^*, y^*)$ minimizes $F$. Finally, by starting from the initialization point as $\boldsymbol{x}_0 = (D, 0)$, the final iterate $\boldsymbol{x}_n^K = (x_n^K, y_n^K)$ obtained by running Algorithm 2 on $F$ satisfies

$$F(\boldsymbol{x}_n^K) - F(\boldsymbol{x}^*) \gtrsim \min\left\{\mu D^2, \frac{G^2}{L}\left(1 + \frac{L}{2\mu nK}\right)^n\right\},$$

for any choice of $D \in \mathbb{R}$ and $\eta > 0$.

This concludes the proof of Theorem 3.5. $\qquad\qquad\qquad\qquad\qquad\qquad\qquad\qquad\qquad\qquad\qquad\qquad\qquad\square$

One key distinction of Theorem 3.5 compared to other lower bound theorems is the explicit inclusion of $D$ in the statement. While it is possible to express the lower bounds in other theorems with a dependency on $D$, we chose to leave this dependency only for Theorem 3.5 due to the unique behavior of the term $\frac{G^2}{L}\left(1 + \frac{L}{2\mu nK}\right)^n$.

Unlike typical bounds, this expression cannot be simplified into a clear, closed-form polynomial expression. Its proportional degree with respect to $\mu$, $L$, $n$, and $K$ varies depending on their values. In particular, when $K$ is small (e.g., near $\frac{\kappa}{n}$), the term exhibits exponential growth, scaling as $c^n \cdot \frac{G^2}{L}$ where $c$ is a constant greater than 1.1.

This exponential growth introduces challenges when attempting to express the bound without the "min" operator, as in other theorems. Specifically, the first coordinate of the initialization point, $x_0$, would need to grow to an exponential scale, which is undesirable to when comparing to the upper bound theorems that hide logarithmic dependency. For these reasons, we leave the dependency on $D$ explicitly in the bound.

In the following subsections, we present the specific construction of $F_1$ and $F_2$, and demonstrate that each satisfies the stated lower bound within its corresponding step size regime. For simplicity of notation, we omit the index of the overall function when referring to its component functions, e.g., we write $f_i(x)$ instead of $f_{1i}(x)$. Moreover, we use the common variable notation $x$ while constructing functions for each dimension, though we use different variables in the "dimension-aggregation" step.

### B.3.1. CONSTRUCTION OF $F_1$

Let $F_1(x) = \frac{\mu}{2}x^2$ with component functions $f_i(x) = F_1(x)$ for all $i \in [n]$. It is clear that $F_1$ satisfies Assumptions 2.3 and 2.5 with $G = P = 0$. Also, we note that $x^* = 0$ and $F_1(x^*) = 0$.

For all $\eta \in \left(0, \frac{1}{\mu nK}\right)$, the final iterate is given by

$$x_n^K = (1 - \eta\mu)^{nK}x_0 \geq \left(1 - \frac{1}{nK}\right)^{nK}x_0 \geq \frac{x_0}{4},$$

where the last inequality uses the fact that $(1 - \frac{1}{m})^m \geq \frac{1}{4}$ for all $m \geq 2$.

Thus, for $x_0 = D$, we have

$$F_1(x_n^K) - F_1(x^*) = \frac{\mu}{2}(x_n^K)^2 \gtrsim \mu D^2.$$

### B.3.2. CONSTRUCTION OF $F_2$

In this section, we focus on the case when $n$ is even. If $n$ is odd, we set $n - 1$ components satisfying the argument, and introduce an additional zero component function. This adjustment does not affect the final result, but only modifies the parameters $\mu$, $L$, $n$ by at most a constant factor.

Let $F_2(x) = \frac{L}{8}x^2$ with component functions

$$f_i(x) = \begin{cases} \frac{L}{2}x^2 + Gx & \text{if } i \leq n/2, \\ -\frac{L}{4}x^2 - Gx & \text{otherwise.} \end{cases}$$

Note that the first $n/2$ component functions are strongly convex, while the remaining component functions are concave. The overall function $F_2$ is $\mu$-strongly convex, since $\frac{L}{4} \geq \mu$ holds from the assumption $\kappa \geq 4$, thereby satisfying Assumption 2.3. Also, it is clear that $f_i$ satisfies Assumption 2.4 with $P = 3$. We note that $x^* = 0$ and $F_2(x^*) = 0$.

We now consider the relationship between $x_0^k$ and $x_0^{k+1}$. Applying Lemma F.3, we have

$$x_0^{k+1} = \left(1 + \frac{\eta L}{2}\right)^{\frac{n}{2}} (1 - \eta L)^{\frac{n}{2}} x_0^k + \frac{G}{L}\left(\left(1 + \frac{\eta L}{2}\right)^{\frac{n}{2}} \left(1 + (1 - \eta L)^{\frac{n}{2}}\right) - 2\right). \tag{15}$$

From $K \leq \frac{\kappa}{4}$, we have $\eta \geq \frac{1}{\mu n K} \geq \frac{4}{nL}$. We now derive the lower bound for $\left(1 + \frac{\eta L}{2}\right)^{\frac{n}{2}}$. To do this, we consider the first three terms of its binomial expansion, which is possible because $\frac{n}{2} \geq 2$:

$$\left(1 + \frac{\eta L}{2}\right)^{\frac{n}{2}} \geq \left(1 + \frac{2}{n}\right)^{\frac{n}{2}} \geq 1 + \frac{2}{n} \cdot \binom{\frac{n}{2}}{1} + \left(\frac{2}{n}\right)^2 \cdot \binom{\frac{n}{2}}{2} = 1 + 1 + \frac{4}{n^2} \cdot \frac{n(n-2)}{8} = \frac{5}{2} - \frac{1}{n} \geq \frac{9}{4},$$

where the last inequality uses $n \geq 4$. Equivalently, the following inequality holds:

$$\left(1 + \frac{\eta L}{2}\right)^{\frac{n}{2}} \geq 2 + \frac{1}{9}\left(1 + \frac{\eta L}{2}\right)^{\frac{n}{2}}.$$

Using this inequality, the numerical term in equation (15) becomes

$$\left(1 + \frac{\eta L}{2}\right)^{\frac{n}{2}} \left(1 + (1 - \eta L)^{\frac{n}{2}}\right) - 2 > \left(1 + \frac{\eta L}{2}\right)^{\frac{n}{2}} - 2 \geq \frac{1}{9}\left(1 + \frac{\eta L}{2}\right)^{\frac{n}{2}}.$$

Substituting this back to equation (15) yields

$$x_0^{k+1} \geq \left(1 + \frac{\eta L}{2}\right)^{\frac{n}{2}} (1 - \eta L)^{\frac{n}{2}} x_0^k + \frac{G}{9L}\left(1 + \frac{\eta L}{2}\right)^{\frac{n}{2}}.$$

Note that if $x_0^k$ is non-negative, we have $x_0^{k+1} \geq \frac{G}{9L}\left(1 + \frac{\eta L}{2}\right)^{\frac{n}{2}} \geq 0$. By setting the initialization point $x_0$ as 0, each $x_0^k$ remains non-negative throughout the process, and therefore the final iterate $x_n^K$ satisfies:

$$x_n^K \geq \frac{G}{9L}\left(1 + \frac{\eta L}{2}\right)^{\frac{n}{2}} \geq \frac{G}{9L}\left(1 + \frac{L}{2\mu n K}\right)^{\frac{n}{2}},$$

where we apply $\eta \geq \frac{1}{\mu n K}$ at the last step. Consequently, the optimality gap is lower bounded as:

$$F_2(x_n^K) - F_2(x^*) = \frac{L}{8}(x_n^K)^2 \gtrsim \frac{G^2}{L}\left(1 + \frac{L}{2\mu n K}\right)^n.$$

## B.4. Technical Lemmas

**Lemma B.1.** *For any $n \geq 2$, the following holds:*

$$\sum_{j=0}^{n-1} e^{\frac{2\pi j}{n}\mathrm{i}} = 0.$$

*where* $\mathrm{i}$ *denotes the imaginary unit. In particular, the following equations hold:*

$$\sum_{j=0}^{n-1} \cos\frac{2\pi j}{n} = 0, \quad and \quad \sum_{j=0}^{n-1} \sin\frac{2\pi j}{n} = 0.$$

*Proof.* Let $\zeta = e^{\frac{2\pi}{n}i}$. Then, $\zeta^n - 1 = e^{2\pi i} - 1 = 0$ holds. Moreover, we have

$$\zeta^n - 1 = \left( \sum_{j=0}^{n-1} \zeta^j \right)(\zeta - 1) = 0.$$

Since $\zeta - 1 \neq 0$, it follows that $\sum_{j=0}^{n-1} \zeta^j = 0$. This leads to the results $\sum_{j=0}^{n-1} \cos \frac{2\pi j}{n} = 0$ and $\sum_{j=0}^{n-1} \sin \frac{2\pi j}{n} = 0$. □

**Lemma B.2.** *For any $n \geq 3$, the following equations hold:*

$$\frac{1}{n} \sum_{j=0}^{n-1} \cos^2 \frac{2\pi j}{n} = \frac{1}{2}, \quad \frac{1}{n} \sum_{j=0}^{n-1} \sin^2 \frac{2\pi j}{n} = \frac{1}{2}, \text{ and } \quad \frac{1}{n} \sum_{j=0}^{n-1} \sin \frac{4\pi j}{n} = 0.$$

*Proof.* First, notice that

$$\cos^2 \frac{2\pi j}{n} = \frac{1}{2}\left(1 + \cos \frac{4\pi j}{n}\right).$$
$$\sin^2 \frac{2\pi j}{n} = \frac{1}{2}\left(1 - \cos \frac{4\pi j}{n}\right).$$

Hence, it suffices to prove $\sum_{j=0}^{n-1} \cos \frac{4\pi j}{n} = 0$ and $\sum_{j=0}^{n-1} \sin \frac{4\pi j}{n} = 0$. Let $\zeta = e^{\frac{4\pi}{n}i}$ where i denote the imaginary number. Then, $\zeta^n - 1 = e^{4\pi i} - 1 = 0$ holds. Moreover, we have

$$\zeta^n - 1 = \left( \sum_{j=0}^{n-1} \zeta^j \right)(\zeta - 1) = 0.$$

Since $\zeta \neq 1$ for $n \geq 3$, it follows that $\sum_{j=0}^{n-1} \zeta^j = 0$. This leads to the results $\sum_{j=0}^{n-1} \cos \frac{4\pi j}{n} = 0$ and $\sum_{j=0}^{n-1} \sin \frac{4\pi j}{n} = 0$. □

**Lemma B.3.** *For $\eta \in \left[ \frac{1}{\mu n K}, \frac{2}{L} \right)$, $u_0(\eta)$ is an increasing function of $\eta$.*

*Proof.* Recall the expression for $u_0(\eta)$ given in equation (11):

$$u_0(\eta) = \frac{\eta L' - (1 - \cos \delta)}{(1 - \cos \delta)(2 - (\mu + L')\eta) + \eta^2 \mu L'} \cdot \eta G.$$

For simplicity of the notation, let $a = 1 - \cos \delta$, $b(\eta) = (\eta L' - a)\eta$, and $c(\eta) = a(2 - \eta(\mu + L')) + \eta^2 \mu L'$. Then, $u_0(\eta)$ can be expressed as $\frac{b(\eta)}{c(\eta)} G$ and $u_0'(\eta)$ becomes $(b'(\eta)c(\eta) - b(\eta)c'(\eta)) G/c(\eta)^2$. It suffices to prove the numerator $b'(\eta)c(\eta) - b(\eta)c'(\eta)$ is non-negative.

Expanding the numerator, we obtain

$$\begin{aligned}
b'(\eta)c(\eta) - b(\eta)c'(\eta) &= (2\eta L' - a)(\eta^2 \mu L' - \eta a(\mu + L') + 2a) - (\eta^2 L' - \eta a)(2\eta \mu L' - a(\mu + L')) \\
&= (2\eta^3 \mu L'^2 - \eta^2 a L'(3\mu + 2L') + \eta a(4L' + a(\mu + L')) - 2a^2) \\
&\quad - (2\eta^3 \mu L'^2 - \eta^2 a L'(3\mu + L') + \eta a^2(\mu + L')) \\
&= -\eta^2 a L'^2 + 4\eta a L' - 2a^2 \\
&= a(4\eta L' - \eta^2 L'^2 - 2a).
\end{aligned} \tag{16}$$

Since $\eta L' = \frac{\eta L}{2} \leq 1$, we have $\eta^2 L'^2 \leq \eta L'$. Moreover, we have $a = 1 - \cos \delta \leq \frac{\eta L'}{2}$ from equation (13).

Substituting these results into equation (16), we have

$$b'(\eta)c(\eta) - b(\eta)c'(\eta) = a(4\eta L' - \eta^2 L'^2 - 2a) \geq a(4\eta L' - \eta L' - \eta L') = 2\eta a L' \geq 0.$$

Therefore, we conclude that $u_0(\eta)$ is an increasing function with respect to $\eta$. □

**Lemma B.4.** *For $\eta \in \left[ \frac{1}{\mu n K}, \frac{2}{L} \right)$, the absolute value of $v_0(\eta)$ is bounded by $u_0(\eta)$ as follows:*

$$|v_0(\eta)| \leq \frac{8\pi K}{\kappa} u_0(\eta).$$

*Proof.* Starting from equations (11) and (12), we have

$$
\begin{aligned}
|v_0(\eta)| &= \frac{\sin \delta}{\eta L' - (1 - \cos \delta)} u_0(\eta) \\
&\leq \frac{2\pi}{n} \cdot \frac{2}{\eta L'} u_0(\eta) \\
&= \frac{4\pi}{\eta n L'} u_0(\eta) \\
&= \frac{8\pi}{\eta n L} u_0(\eta),
\end{aligned}
$$

where we employ $\sin \delta \leq \delta = \frac{2\pi}{n}$, $1 - \cos \delta \leq \frac{\eta L'}{2}$ from equation (13), and $L' = \frac{L}{2}$. Finally, applying the condition $\eta \geq \frac{1}{\mu n K}$ completes the proof of desired inequality. $\qquad \square$

# C. Proofs for Small Epoch Upper Bounds

In this section, we provide detailed proofs for Theorem 3.2, Proposition 3.4, and Theorem 3.7 which correspond to upper bound results in the small epoch regime.

## C.1. Proof of Theorem 3.2

**Theorem 3.2.** *Let $n \geq 1$, $\frac{\kappa}{n} \lesssim K \leq \kappa$, and an initialization point $x_0$. Suppose $F$ is a 1-dimensional function satisfying Assumptions 2.3 and 2.5. Assume that each component function $f_i$ shares the same Hessian for all $i \in [n]$ and $x \in \mathbb{R}$. Then, for any choice of permutation $\sigma_k$ in each epoch, the final iterate $x_n^K$ obtained by Algorithm 1 with the step size $\eta = \frac{1}{\mu n K} \max \left\{ \log \left( \frac{L|x_0 - x^*|}{G_*} \right), 1 \right\}$ satisfies*

$$F(x_n^K) - F(x^*) \lesssim \frac{G_*^2}{\mu K}.$$

*Proof.* Since each $f_i$ has the identical Hessian, we have $\nabla^2 f_i(x) = \nabla^2 F(x)$ for every $x \in \mathbb{R}$. Consequently, for all $i \in [n]$, we can express the gradient difference as follows:

$$\nabla f_i(x) - \nabla F(x) = \nabla f_i(x^*) - \nabla F(x^*) + \int_{x^*}^{x} \left( \nabla^2 f_i(\alpha) - \nabla^2 F(\alpha) \right) \, \mathrm{d}\alpha = \nabla f_i(x^*).$$

For simplicity, let $a_i = -\nabla f_i(x^*)$ for $i \in [n]$. Then, from the definition of $F(x) = \frac{1}{n} \sum_{i=1}^{n} f_i(x)$, we have $\sum_{i=1}^{n} a_i = 0$. Furthermore, it follows from Assumption 2.5 that $|a_i| \leq G_*$. To further classify the indices, we define

$$I_+ = \{ i \in [n] \,|\, a_i \geq 0 \} \text{ and } I_- = \{ i \in [n] \,|\, a_i < 0 \} .$$

Here, $I_+$ represents the collection of component functions whose minima are greater than or equal to $x^*$, while $I_-$ consists of the remaining functions.

We begin by presenting the following lemma:

**Lemma C.1.** *Let $p, q \in \mathbb{R}$ with $p < q$, and let $p'$ and $q'$ denote the results of performing a single step of gradient descent on a $\mu$-strongly convex and $L$-smooth 1-dimensional function $f$, starting from $p$ and $q$, respectively, with a step size $\eta < \frac{1}{L}$. Then, it holds that $0 < q' - p' \leq (1 - \eta\mu)(q - p)$.*

The proof for Lemma C.1 is presented in Appendix C.4. Now, let $z_0 = x^*$, initialized at the minima of the overall function $F$, and define $z_i^k$ as the $i$-th iterate of the $k$-th epoch, using the same permutations employed for $x_i^k$ but instead starting from the initial point $z_0$. Since the distance between $x_i^k$ and $z_i^k$ decreases by at least a factor of $(1 - \eta\mu)$ at each iteration (Lemma C.1), we have

$$\left| x_n^K - z_n^K \right| \leq (1 - \eta\mu)^{nK} \left| x_0 - z_0^1 \right| \leq e^{-\eta\mu nK} \left| x_0 - x^* \right| \leq \frac{G_*}{L}, \tag{17}$$

where we substitute $\eta = \frac{1}{\mu n K} \max \left\{ \log \left( \frac{L|x_0 - x^*|}{G_*} \right), 1 \right\}$ in the last step. This demonstrates that $x_n^K$ and $z_n^K$ remain sufficiently close. For the rest of the analysis, we mainly focus on how far $z_n^K$ can deviate from $x^*$. The bound for $F(x_n^K)$ will later be controlled by leveraging $L$-smoothness between $x_n^K$ and $z_n^K$.

In the special case where $I_- = \emptyset$, all $a_i$ are equal to 0 since $\sum_{i=1}^{n} a_i = 0$. In this scenario, $z_n^K$ remains at $x^*$ because $\nabla f_i(z_0^1) = 0$ holds for all $i \in [n]$, resulting in $z_n^K = x^*$. Using this, we have

$$F(x_n^K) \leq F(z_n^K) + \left\langle \nabla F(z_n^K), x_n^K - z_n^K \right\rangle + \frac{L}{2} \left| x_n^K - z_n^K \right|^2 \quad (\because L\text{-smoothness})$$

$$= F(x^*) + \frac{L}{2} \left( \frac{G_*}{L} \right)^2 = F(x^*) + \frac{G_*^2}{2L} \leq F(x^*) + \frac{G_*^2}{2\mu K},$$

where we apply $K \leq \kappa$ in the last inequality. This concludes the proof for this special case. For the remainder of the proof, we assume $I_- \neq \emptyset$.

For each $k \in [K]$, define $z_+^k$ as the maximum possible final iterate obtained after running Algorithm 1 starting from $x^*$, i.e., the largest value among the $(n!)^k$ possible options. Similarly, for each $k \in [K]$, let $z_-^k$ denote the minimum among the $(n!)^k$ options, also starting from $x^*$. Consequently, by convexity of $F$, $F(z_n^K)$ can naturally be upper bounded by

$$\max \left\{ F(z_+^K), F(z_-^K) \right\}.$$

The following lemma helps us to establish upper bounds for $z_+^K - x^*$ and $-(z_-^K - x^*)$.

**Lemma C.2.** *Let $\{\sigma_k^+\}_{k=1}^K$ denote the sequence of permutations applied over $K$ epochs to generate $z_+^K$. These permutations and the corresponding $z_+^K$ satisfy the following properties:*

- *The permutations $\{\sigma_k^+\}_{k=1}^{K-1}$, applied during the first $K-1$ epochs, produce $z_+^{K-1}$.*
- *For any $k \in [K]$, all indices in $I_+$ appear before all indices in $I_-$ in the permutation $\sigma_k^+$.*
- *For any $k \in [K]$, let $z_+^{k,I_-}$ denote the $|I_-|$-th iterate in the $k$-th epoch, i.e., obtained after processing all indices in $I_-$. Then, the inequality $z_+^{k,I_-} \leq x^* \leq z_+^k$ holds.*

*For $z_-^K$ and its corresponding permutations $\{\sigma_k^-\}_{k=1}^k$, the following properties hold:*

- *The permutations $\{\sigma_k^-\}_{k=1}^{K-1}$, applied during the first $K-1$ epochs, produce $z_-^{K-1}$.*
- *For any $k \in [K]$, all indices in $I_-$ appear before all indices in $I_+$ in the permutation $\sigma_k^-$.*
- *For any $k \in [K]$, let $z_-^{k,I_+}$ denote the $|I_+|$-th iterate in the $k$-th epoch, i.e., obtained after processing all indices in $I_+$. Then, the inequality $z_-^{k,I_+} \geq x^* \geq z_-^k$ holds.*

The proof for Lemma C.2 is presented in Appendix C.4. Define $\pi_+ : [|I_+|] \to [n]$ as the ordering of $I_+$ used during the $K$-th epoch to generate $z_+^K$ from $z_+^{K,I_-}$. We then define the sequence of iterates $u_0, u_1, \ldots u_{|I_+|}$ where $u_0 = x^*$ and each subsequent $u_i$ is obtained by applying a gradient update using the component function $f_{\pi_+(i)}$ to $u_{i-1}$. We emphasize the following two key points:

1. $z_+^{K,I_-} \leq x^* = u_0$.
2. The sequences of iterates $z_+^{K,I_-}, \ldots, z_+^K$ and $u_0, \ldots u_{|I_+|}$ are generated by the same component function ordering.

From these observations, we conclude that $z_+^K \leq u_{|I_+|}$ as Lemma C.1 ensures that the relationship $p \leq q$ is preserved under gradient descent (i.e., if $p \leq q$, then $p' \leq q'$ after each update). Together with $x^* \leq z_+^K$ from Lemma C.2, we obtain $0 \leq z_+^K - x^* \leq u_{|I_+|} - x^*$.

Similarly, define $\pi_- : [|I_-|] \to [n]$ as the ordering of $I_-$ used during the $K$-th epoch to generate $z_-^K$ from $z_-^{K,I_+}$. Also, define the sequence of iterates $v_0, v_1, \ldots, v_{|I_-|}$ where $v_0 = x^*$ and each subsequent $v_i$ is obtained by applying a gradient update using the component function $f_{\pi_-(i)}$ to $v_{i-1}$. Then, we have $x^* \geq z_-^K \geq v_{|I_-|}$, leading to $0 \leq -(z_-^K - x^*) \leq -(v_{|I_-|} - x^*)$.

To summarize the process so far, we aim to upper bound $\left| z_n^K - x^* \right|$ where $z_n^K$ is the final iterate obtained using the same permutations as $x_n^K$ but starting from $z_0 = x^*$ instead of $x_0 = x_0$. Since $z_+^K$ and $z_-^K$ represent the maximum and minimum possible final iterate of $z_n^K$, respectively, the followings hold:

$$\left| z_n^K - x^* \right| \leq \max \left\{ z_+^K - x^*, -(z_-^K - x^*) \right\} \leq \max \left\{ u_{|I_+|} - x^*, -(v_{|I_-|} - x^*) \right\}$$

and therefore, by convexity of $F$,

$$F(z_n^K) \leq \max \left\{ F(u_{|I_+|}), F(v_{|I_-|}) \right\}. \tag{18}$$

We now focus on providing the upper bound for $\max \left\{ F(u_{|I_+|}), F(v_{|I_-|}) \right\}$. To this end, we introduce the following lemma:

**Lemma C.3.** *With a step size $\eta < \frac{1}{L}$, $0 \leq \nabla F(u_i) \leq 2G_*$ holds for all $i \in \{0\} \cup [|I_+|]$ and $0 \geq \nabla F(v_j) \geq -2G_*$ holds for all $j \in \{0\} \cup [|I_-|]$.*

The proof for Lemma C.3 is presented in Appendix C.4. Using Lemma C.3, we can upper bound the increments in the per-iteration function evaluation as follows:

$$
\begin{aligned}
F(u_i) &= F(u_{i-1}) + \int_{u_{i-1}}^{u_i} \nabla F(\alpha)\, \mathrm{d}\alpha \\
&\leq F(u_{i-1}) + |u_i - u_{i-1}| \cdot |\nabla F(u_{i-1} + c_i \cdot (u_i - u_{i-1}))| \\
&= F(u_{i-1}) + \eta \left| \nabla f_{\pi_+(i)}(u_{i-1}) \right| \cdot |\nabla F(u_{i-1} + c_i \cdot (u_i - u_{i-1}))| \\
&= F(u_{i-1}) + \eta \left| \nabla F(u_{i-1}) - a_{\pi_+(i)} \right| \cdot |\nabla F(u_{i-1} + c_i \cdot (u_i - u_{i-1}))| \\
&\leq F(u_{i-1}) + \eta \cdot 4 G_*^2,
\end{aligned}
\tag{19}
$$

where $0 \leq c_i \leq 1$ by Mean Value Theorem. The last inequality follows from the bounds $0 \leq \nabla F(u_{i-1}) \leq 2 G_*$ and $0 \leq a_{\pi_+(i)} \leq G_*$ (since $\pi_+(i) \in I_+$). Additionally, $\min\{\nabla F(u_{i-1}), \nabla F(u_i)\} \leq \nabla F(u_{i-1} + c_i \cdot (u_i - u_{i-1})) \leq \max\{\nabla F(u_{i-1}), \nabla F(u_i)\}$ holds as $\nabla F$ is a strictly increasing function.

Unrolling equation (19) for $i = 1, 2, \ldots, |I_+|$, we obtain:

$$
F(u_{|I_+|}) \leq F(x^*) + 4 \eta n G_*^2.
$$

By applying a similar argument, we can derive a corresponding bound for $v_{|I_-|}$:

$$
F(v_{|I_-|}) \leq F(x^*) + 4 \eta n G_*^2.
$$

Therefore, equation (18) becomes

$$
F(z_n^K) \leq F(x^*) + 4 \eta n G_*^2.
$$

We now proceed to derive the upper bound for $F(x_n^K)$. We already established in equation (17) that $\left| x_n^K - z_n^K \right| \leq G_*/L$. Consequently, by applying $L$-smoothness,

$$
\begin{aligned}
F(x_n^K) &\leq F(z_n^K) + \left\langle \nabla F(z_n^K), x_n^K - z_n^K \right\rangle + \frac{L}{2} \left| x_n^K - z_n^K \right|^2 \\
&\leq \left( F(x^*) + 4 \eta n G_*^2 \right) + 2 G_* \cdot \frac{G_*}{L} + \frac{L}{2} \left( \frac{G_*}{L} \right)^2 \\
&= F(x^*) + \frac{4 G_*^2}{\mu K} \max\left\{ \log\left( \frac{L |x_0 - x^*|}{G_*} \right), 1 \right\} + \frac{5 G_*^2}{2L},
\end{aligned}
$$

where we used the fact that $\left| \nabla F(z_n^K) \right| \leq \max\{\nabla F(z_+^K), -\nabla F(z_-^K)\} < \max\{\nabla F(u_{|I_+|}), -\nabla F(v_{|I_-|})\} \leq 2 G_*$ and $\eta = \frac{1}{\mu n K} \max\left\{ \log\left( \frac{L|x_0-x^*|}{G_*} \right), 1 \right\}$. Since $K \leq \kappa$, we have $L \geq \mu K$, and therefore,

$$
F(x_n^K) - F(x^*) \lesssim \frac{G_*^2}{\mu K}.
$$

This concludes the proof of Theorem 3.2. $\qquad\square$

## C.2. Proof of Proposition 3.4

**Proposition 3.4** (Mishchenko et al. (2020), Theorem 5). *Let $n \geq 1$, $K \gtrsim \frac{\kappa}{n}$, and $\boldsymbol{x}_0$ be the initialization point. Suppose $F$ is a function satisfying Assumptions 2.3 and 2.5 where each component function is $\mu$-strongly convex. Then, for any choice of permutation $\sigma_k$ in each epoch, the final iterate $\boldsymbol{x}_n^K$ obtained by running Algorithm 1 with a step size $\eta = \frac{2}{\mu n K} \max\left\{ \log\left( \frac{\|\boldsymbol{x}_0 - \boldsymbol{x}^*\| \mu K}{\sqrt{\kappa} G_*} \right), 1 \right\}$, satisfies*

$$
\left\| \boldsymbol{x}_n^K - \boldsymbol{x}^* \right\|^2 \lesssim \frac{L G_*^2}{\mu^3 K^2}.
$$

*Proof.* The original statement by Theorem 5 in (the appendix of) Mishchenko et al. (2020) holds only for IGD. We here extend the theorem to hold for arbitrary permutation-based SGD, and reorganize some terms to facilitate clear comparison to the proof of Theorem 3.7.

We begin by noting the specific epoch condition stated as $K \gtrsim \frac{\kappa}{n}$ in the theorem statement:

$$K \geq \frac{2\kappa}{n} \max \left\{ \log \left( \frac{\|\boldsymbol{x}_0 - \boldsymbol{x}^*\| \mu K}{\sqrt{\kappa} G_*} \right), 1 \right\}.$$

Under this condition, the specified step size $\eta = \frac{2}{\mu n K} \max \left\{ \log \left( \frac{\|\boldsymbol{x}_0 - \boldsymbol{x}^*\| \mu K}{\sqrt{\kappa} G_*} \right), 1 \right\}$ satisfies $\eta \leq \frac{1}{L}$.

For each $k \in [K]$, we use the permutation $\sigma_k$ to define a sequence of iterates $\{\boldsymbol{x}_{k,i}^*\}_{i=0}^n$ as follows:

$$\boldsymbol{x}_{k,0}^* = \boldsymbol{x}^*,$$
$$\boldsymbol{x}_{k,i}^* = \boldsymbol{x}_{k,i-1}^* - \eta \nabla f_{\sigma_k(i)}(\boldsymbol{x}^*).$$

The sequence $\boldsymbol{x}_{k,i}^*$ can be interpreted as the sequence starting from $\boldsymbol{x}^*$ obtained by using the component gradients in the same order as $\boldsymbol{x}_i^k$, but the gradients are being evaluated at $\boldsymbol{x}^*$ instead of $\boldsymbol{x}_{i-1}^k$. From $\sum_{i=1}^n \nabla f_{\sigma_k(i)}(\boldsymbol{x}^*) = n \nabla F(\boldsymbol{x}^*) = 0$, we can easily deduce that $\boldsymbol{x}_{k,n}^* = \boldsymbol{x}^* = \boldsymbol{x}_{k+1,0}^*$.

We analyze the square norm distance $\left\| \boldsymbol{x}_i^k - \boldsymbol{x}_{k,i}^* \right\|^2$ using an iteration-wise recursive inequality:

$$
\begin{aligned}
\left\| \boldsymbol{x}_i^k - \boldsymbol{x}_{k,i}^* \right\|^2 &= \left\| \boldsymbol{x}_{i-1}^k - \eta \nabla f_{\sigma_k(i)}(\boldsymbol{x}_{i-1}^k) - \left( \boldsymbol{x}_{k,i-1}^* - \eta \nabla f_{\sigma_k(i)}(\boldsymbol{x}^*) \right) \right\|^2 \\
&= \left\| \boldsymbol{x}_{i-1}^k - \boldsymbol{x}_{k,i-1}^* \right\|^2 - 2\eta \left\langle \boldsymbol{x}_{i-1}^k - \boldsymbol{x}_{k,i-1}^*, \nabla f_{\sigma_k(i)}(\boldsymbol{x}_{i-1}^k) - \nabla f_{\sigma_k(i)}(\boldsymbol{x}^*) \right\rangle \\
&\quad + \eta^2 \left\| \nabla f_{\sigma_k(i)}(\boldsymbol{x}_{i-1}^k) - \nabla f_{\sigma_k(i)}(\boldsymbol{x}^*) \right\|^2 \\
&\overset{(a)}{=} \left\| \boldsymbol{x}_{i-1}^k - \boldsymbol{x}_{k,i-1}^* \right\|^2 - 2\eta \left( D_{f_{\sigma_k(i)}}(\boldsymbol{x}_{i-1}^k, \boldsymbol{x}^*) + D_{f_{\sigma_k(i)}}(\boldsymbol{x}_{k,i-1}^*, \boldsymbol{x}_{i-1}^k) - D_{f_{\sigma_k(i)}}(\boldsymbol{x}_{k,i-1}^*, \boldsymbol{x}^*) \right) \\
&\quad + \eta^2 \left\| \nabla f_{\sigma_k(i)}(\boldsymbol{x}_{i-1}^k) - \nabla f_{\sigma_k(i)}(\boldsymbol{x}^*) \right\|^2.
\end{aligned}
\tag{20}
$$

Here, $D_f(\boldsymbol{x}, \boldsymbol{y}) := f(\boldsymbol{x}) - f(\boldsymbol{y}) - \langle \nabla f(\boldsymbol{y}), \boldsymbol{x} - \boldsymbol{y} \rangle$ denotes the Bregman divergence of $f$ between $\boldsymbol{x}$ and $\boldsymbol{y}$. At $(a)$, we apply the three-point identity of the Bregman divergence.

The term $D_{f_{\sigma_k(i)}}(\boldsymbol{x}_{k,i-1}^*, \boldsymbol{x}_{i-1}^k)$ in equation (20) can be bounded as follows:

$$D_{f_{\sigma_k(i)}}(\boldsymbol{x}_{k,i-1}^*, \boldsymbol{x}_{i-1}^k) \geq \frac{\mu}{2} \left\| \boldsymbol{x}_{k,i-1}^* - \boldsymbol{x}_{i-1}^k \right\|^2,$$

by the $\mu$-strong convexity of the component function. Moreover, from Lemma 2.29 of Garrigos & Gower (2023), we have

$$\left\| \nabla f_{\sigma_k(i)}(\boldsymbol{x}_{i-1}^k) - \nabla f_{\sigma_k(i)}(\boldsymbol{x}^*) \right\|^2 \leq 2L D_{f_{\sigma_k(i)}}(\boldsymbol{x}_{i-1}^k, \boldsymbol{x}^*).$$

Substituting these inequalities into equation (20), we derive

$$
\begin{aligned}
\left\| \boldsymbol{x}_i^k - \boldsymbol{x}_{k,i}^* \right\|^2 &\leq \left\| \boldsymbol{x}_{i-1}^k - \boldsymbol{x}_{k,i-1}^* \right\|^2 - 2\eta(1 - \eta L) D_{f_{\sigma_k(i)}}(\boldsymbol{x}_{i-1}^k, \boldsymbol{x}^*) - \eta\mu \left\| \boldsymbol{x}_{i-1}^k - \boldsymbol{x}_{k,i-1}^* \right\|^2 + 2\eta D_{f_{\sigma_k(i)}}(\boldsymbol{x}_{k,i-1}^*, \boldsymbol{x}^*) \\
&\overset{(a)}{\leq} (1 - \eta\mu) \left\| \boldsymbol{x}_{i-1}^k - \boldsymbol{x}_{k,i-1}^* \right\|^2 + 2\eta D_{f_{\sigma_k(i)}}(\boldsymbol{x}_{k,i-1}^*, \boldsymbol{x}^*) \\
&\overset{(b)}{\leq} (1 - \eta\mu) \left\| \boldsymbol{x}_{i-1}^k - \boldsymbol{x}_{k,i-1}^* \right\|^2 + \eta^3 L n^2 G_*^2,
\end{aligned}
\tag{21}
$$

where we apply $1 - \eta L \geq 0$ and $D_{f_{\sigma_k(i)}}(\boldsymbol{x}_{i-1}^k, \boldsymbol{x}^*) \geq 0$ at $(a)$. At $(b)$, we utilize the $L$-smoothness of the component function and the triangle inequality:

$$D_{f_{\sigma_k(i)}}(\boldsymbol{x}_{k,i-1}^*, \boldsymbol{x}^*) \leq \frac{L}{2} \left\| \boldsymbol{x}_{k,i-1}^* - \boldsymbol{x}^* \right\|^2 = \frac{\eta^2 L}{2} \left\| \sum_{j=1}^{i-1} \nabla f_{\sigma_k(j)}(\boldsymbol{x}^*) \right\|^2 \leq \frac{\eta^2 L}{2} \cdot (n G_*)^2.$$

Thus, by unrolling equation (21) over all $k \in [K]$ and all $i \in [n]$, and noting that $\boldsymbol{x}^*_{k,n} = \boldsymbol{x}^* = \boldsymbol{x}^*_{k+1,0}$, we obtain

$$\left\| \boldsymbol{x}^K_n - \boldsymbol{x}^* \right\|^2 \leq (1 - \eta\mu)^{nK} \left\| \boldsymbol{x}^1_0 - \boldsymbol{x}^* \right\|^2 + \eta^3 L n^2 G^2_* \sum_{t=1}^{nK} (1 - \eta\mu)^{t-1}$$

$$= (1 - \eta\mu)^{nK} \left\| \boldsymbol{x}_0 - \boldsymbol{x}^* \right\|^2 + \eta^3 L n^2 G^2_* \frac{1 - (1 - \eta\mu)^{nK}}{\eta\mu}$$

$$\leq e^{-\eta\mu nK} \left\| \boldsymbol{x}_0 - \boldsymbol{x}^* \right\|^2 + \frac{\eta^2 L n^2 G^2_*}{\mu}.$$

We now substitute $\eta = \frac{2}{\mu nK} \max \left\{ \log \left( \frac{\|\boldsymbol{x}_0 - \boldsymbol{x}^*\| \mu K}{\sqrt{\kappa} G_*} \right), 1 \right\}$. When $\|\boldsymbol{x}_0 - \boldsymbol{x}^*\|$ is sufficiently large, the above inequality simplifies to

$$\left\| \boldsymbol{x}^K_n - \boldsymbol{x}^* \right\|^2 \leq \frac{L G^2_*}{\mu^3 K^2} \left( 1 + 4 \log^2 \left( \frac{\|\boldsymbol{x}_0 - \boldsymbol{x}^*\| \mu K}{\sqrt{\kappa} G_*} \right) \right) \lesssim \frac{L G^2_*}{\mu^3 K^2}.$$

On the other hand, when $\|\boldsymbol{x}_0 - \boldsymbol{x}^*\|$ is small so that 1 is chosen after the max operation, the above inequality simplifies to

$$\left\| \boldsymbol{x}^K_n - \boldsymbol{x}^* \right\|^2 \leq \frac{1}{e^2} \cdot e^2 \frac{\kappa G^2_*}{\mu^2 K^2} + \frac{4 L G^2_*}{\mu^3 K^2} \lesssim \frac{L G^2_*}{\mu^3 K^2},$$

where we use $\|\boldsymbol{x}_0 - \boldsymbol{x}^*\| \leq e \cdot \frac{\sqrt{\kappa} G_*}{\mu K}$.

In particular, using the $L$-smoothness of $F$, the function optimality gap can be bounded as:

$$F(\boldsymbol{x}^K_n) - F(\boldsymbol{x}^*) \leq \frac{L}{2} \left\| \boldsymbol{x}^K_n - \boldsymbol{x}^* \right\|^2 \lesssim \frac{L^2 G^2_*}{\mu^3 K^2}.$$

This ends the proof of Proposition 3.4. $\qquad\qquad\qquad\qquad\qquad\qquad\qquad\qquad\qquad\qquad\qquad\square$

### C.3. Proof of Theorem 3.7

**Theorem 3.7** (Herding at Optimum). *Let $n \geq 1$, $K \gtrsim \frac{\kappa}{n}$, and $\boldsymbol{x}_0$ be the initialization point. Suppose $F$ is a function satisfying Assumptions 2.3 and 2.5 where each component function is $\mu$-strongly convex. Then, there exists a permutation $\sigma$ such that the final iterate $\boldsymbol{x}^K_n$ obtained by running Algorithm 1 with $K$ epochs of $\sigma$ and a step size $\eta = \frac{2}{\mu nK} \max \left\{ \log \left( \frac{\|\boldsymbol{x}_0 - \boldsymbol{x}^*\| \mu nK}{\sqrt{\kappa} H G_*} \right), 1 \right\}$, satisfies*

$$\left\| \boldsymbol{x}^K_n - \boldsymbol{x}^* \right\|^2 \lesssim \frac{H^2 L G^2_*}{\mu^3 n^2 K^2}.$$

*Proof.* We begin by noting the specific epoch condition stated as $K \gtrsim \frac{\kappa}{n}$ in the theorem statement:

$$K \geq \frac{2\kappa}{n} \max \left\{ \log \left( \frac{\|\boldsymbol{x}_0 - \boldsymbol{x}^*\| \mu nK}{\sqrt{\kappa} H G_*} \right), 1 \right\}.$$

Under this condition, the specified step size $\eta = \frac{2}{\mu nK} \max \left\{ \log \left( \frac{\|\boldsymbol{x}_0 - \boldsymbol{x}^*\| \mu nK}{\sqrt{\kappa} H G_*} \right), 1 \right\}$ satisfies $\eta \leq \frac{1}{L}$.

Next, we consider the scaled gradient of each component function at $\boldsymbol{x}^*$:

$$\left\{ \frac{\nabla f_1(\boldsymbol{x}^*)}{G_*}, \frac{\nabla f_2(\boldsymbol{x}^*)}{G_*}, \dots, \frac{\nabla f_n(\boldsymbol{x}^*)}{G_*} \right\}.$$

From Assumption 2.5, we have $\|\nabla f_i(\boldsymbol{x}^*)\| \leq G_*$ for all $i \in [n]$. Thus, the norm of each element is bounded by 1. Also, since $\sum_{i=1}^n \nabla f_i(\boldsymbol{x}^*) = n\nabla F(\boldsymbol{x}^*) = 0$, it follows that these elements sum to 0. Therefore, we can apply the Herding algorithm, as stated in Lemma 3.6, to obtain a permutation $\sigma^* : [n] \to [n]$ satisfying

$$\max_{i \in [n]} \left\| \sum_{j=1}^i \nabla f_{\sigma^*(j)}(\boldsymbol{x}^*) \right\| \leq H G_*, \qquad\qquad\qquad\qquad (22)$$

where $H = \tilde{\mathcal{O}}(\sqrt{d})$. We will demonstrate that this permutation $\sigma^*$ is the desired one: the final iterate $\boldsymbol{x}_n^K$ obtained by running Algorithm 1 for $K$ epochs of $\sigma^*$ satisfies the desired upper bound.

Using $\sigma^*$, we define a sequence of iterates $\{\boldsymbol{x}_i^*\}_{i=0}^n$ as follows:

$$\boldsymbol{x}_0^* = \boldsymbol{x}^*,$$
$$\boldsymbol{x}_i^* = \boldsymbol{x}_{i-1}^* - \eta \nabla f_{\sigma^*(i)}(\boldsymbol{x}^*).$$

Note that the sequence is obtained by using the component gradients at the minimizer $\boldsymbol{x}^*$. From $\sum_{i=1}^n \nabla f_{\sigma^*(i)}(\boldsymbol{x}^*) = n\nabla F(\boldsymbol{x}^*) = 0$, we can easily deduce that $\boldsymbol{x}_n^* = \boldsymbol{x}_0^* = \boldsymbol{x}^*$.

The proof follows the approach used in Theorem 1 in Mishchenko et al. (2020) with several modifications using the property from the Herding algorithm. We analyze the square norm distance $\left\|\boldsymbol{x}_i^k - \boldsymbol{x}_i^*\right\|^2$ using an iteration-wise recursive inequality:

$$
\begin{aligned}
\left\|\boldsymbol{x}_i^k - \boldsymbol{x}_i^*\right\|^2 &= \left\|\boldsymbol{x}_{i-1}^k - \eta \nabla f_{\sigma^*(i)}(\boldsymbol{x}_{i-1}^k) - \left(\boldsymbol{x}_{i-1}^* - \eta \nabla f_{\sigma^*(i)}(\boldsymbol{x}^*)\right)\right\|^2 \\
&= \left\|\boldsymbol{x}_{i-1}^k - \boldsymbol{x}_{i-1}^*\right\|^2 - 2\eta \left\langle \boldsymbol{x}_{i-1}^k - \boldsymbol{x}_{i-1}^*, \nabla f_{\sigma^*(i)}(\boldsymbol{x}_{i-1}^k) - \nabla f_{\sigma^*(i)}(\boldsymbol{x}^*)\right\rangle \\
&\quad + \eta^2 \left\|\nabla f_{\sigma^*(i)}(\boldsymbol{x}_{i-1}^k) - \nabla f_{\sigma^*(i)}(\boldsymbol{x}^*)\right\|^2 \\
&\overset{(a)}{=} \left\|\boldsymbol{x}_{i-1}^k - \boldsymbol{x}_{i-1}^*\right\|^2 - 2\eta \left(D_{f_{\sigma^*(i)}}(\boldsymbol{x}_{i-1}^k, \boldsymbol{x}^*) + D_{f_{\sigma^*(i)}}(\boldsymbol{x}_{i-1}^*, \boldsymbol{x}_{i-1}^k) - D_{f_{\sigma^*(i)}}(\boldsymbol{x}_{i-1}^*, \boldsymbol{x}^*)\right) \\
&\quad + \eta^2 \left\|\nabla f_{\sigma^*(i)}(\boldsymbol{x}_{i-1}^k) - \nabla f_{\sigma^*(i)}(\boldsymbol{x}^*)\right\|^2.
\end{aligned}
\tag{23}
$$

Here, $D_f(\boldsymbol{x}, \boldsymbol{y}) := f(\boldsymbol{x}) - f(\boldsymbol{y}) - \langle \nabla f(\boldsymbol{y}), \boldsymbol{x} - \boldsymbol{y}\rangle$ denotes the Bregman divergence of $f$ between $\boldsymbol{x}$ and $\boldsymbol{y}$. At $(a)$, we apply the three-point identity of the Bregman divergence.

The term $D_{f_{\sigma^*(i)}}(\boldsymbol{x}_{i-1}^*, \boldsymbol{x}_{i-1}^k)$ in equation (23) can be bounded as follows:

$$D_{f_{\sigma^*(i)}}(\boldsymbol{x}_{i-1}^*, \boldsymbol{x}_{i-1}^k) \geq \frac{\mu}{2}\left\|\boldsymbol{x}_{i-1}^* - \boldsymbol{x}_{i-1}^k\right\|^2,$$

by the $\mu$-strong convexity of the component function. Moreover, from Lemma 2.29 of Garrigos & Gower (2023), we have

$$\left\|\nabla f_{\sigma^*(i)}(\boldsymbol{x}_{i-1}^k) - \nabla f_{\sigma^*(i)}(\boldsymbol{x}^*)\right\|^2 \leq 2L D_{f_{\sigma^*(i)}}(\boldsymbol{x}_{i-1}^k, \boldsymbol{x}^*).$$

Substituting these inequalities into equation (23), we derive

$$
\begin{aligned}
\left\|\boldsymbol{x}_i^k - \boldsymbol{x}_i^*\right\|^2 &\leq \left\|\boldsymbol{x}_{i-1}^k - \boldsymbol{x}_{i-1}^*\right\|^2 - 2\eta(1-\eta L)D_{f_{\sigma^*(i)}}(\boldsymbol{x}_{i-1}^k, \boldsymbol{x}^*) - \eta\mu\left\|\boldsymbol{x}_{i-1}^k - \boldsymbol{x}_{i-1}^*\right\|^2 + 2\eta D_{f_{\sigma^*(i)}}(\boldsymbol{x}_{i-1}^*, \boldsymbol{x}^*) \\
&\overset{(a)}{\leq} (1-\eta\mu)\left\|\boldsymbol{x}_{i-1}^k - \boldsymbol{x}_{i-1}^*\right\|^2 + 2\eta D_{f_{\sigma^*(i)}}(\boldsymbol{x}_{i-1}^*, \boldsymbol{x}^*) \\
&\overset{(b)}{\leq} (1-\eta\mu)\left\|\boldsymbol{x}_{i-1}^k - \boldsymbol{x}_{i-1}^*\right\|^2 + H^2\eta^3 LG_*^2,
\end{aligned}
\tag{24}
$$

where we apply $1 - \eta L \geq 0$ and $D_{f_{\sigma^*(i)}}(\boldsymbol{x}_{i-1}^k, \boldsymbol{x}^*) \geq 0$ at $(a)$. At $(b)$, we utilize the $L$-smoothness of the component function and the property of the Herding algorithm, given in equation (22):

$$D_{f_{\sigma^*(i)}}(\boldsymbol{x}_{i-1}^*, \boldsymbol{x}^*) \leq \frac{L}{2}\left\|\boldsymbol{x}_{i-1}^* - \boldsymbol{x}^*\right\|^2 = \frac{\eta^2 L}{2}\left\|\sum_{j=1}^{i-1} \nabla f_{\sigma^*(j)}(\boldsymbol{x}^*)\right\|^2 \leq \frac{\eta^2 L}{2}\cdot(HG_*)^2.$$

Thus, by unrolling equation (24) over all $k \in [K]$ and all $i \in [n]$, and noting that $\boldsymbol{x}_n^* = \boldsymbol{x}_0^* = \boldsymbol{x}^*$, we obtain

$$
\begin{aligned}
\left\|\boldsymbol{x}_n^K - \boldsymbol{x}^*\right\|^2 &\leq (1-\eta\mu)^{nK}\left\|\boldsymbol{x}_0^1 - \boldsymbol{x}^*\right\|^2 + H^2\eta^3 LG_*^2 \sum_{t=1}^{nK}(1-\eta\mu)^{t-1} \\
&= (1-\eta\mu)^{nK}\left\|\boldsymbol{x}_0 - \boldsymbol{x}^*\right\|^2 + H^2\eta^3 LG_*^2 \frac{1-(1-\eta\mu)^{nK}}{\eta\mu}
\end{aligned}
$$

$$\leq e^{-\eta\mu nK}\left\|\boldsymbol{x}_0 - \boldsymbol{x}^*\right\|^2 + \frac{H^2\eta^2 LG_*^2}{\mu}.$$

We now substitute $\eta = \frac{2}{\mu nK}\max\left\{\log\left(\frac{\|\boldsymbol{x}_0-\boldsymbol{x}^*\|\mu nK}{\sqrt{\kappa}HG_*}\right), 1\right\}$. When $\|\boldsymbol{x}_0 - \boldsymbol{x}^*\|$ is sufficiently large, the above inequality simplifies to

$$\left\|\boldsymbol{x}_n^K - \boldsymbol{x}^*\right\|^2 \leq \frac{H^2 LG_*^2}{\mu^3 n^2 K^2}\left(1 + 4\log^2\left(\frac{\|\boldsymbol{x}_0-\boldsymbol{x}^*\|\mu nK}{\sqrt{\kappa}HG_*}\right)\right) \lesssim \frac{H^2 LG_*^2}{\mu^3 n^2 K^2}.$$

On the other hand, when $\|\boldsymbol{x}_0 - \boldsymbol{x}^*\|$ is small so that 1 is chosen after the max operation, the above inequality simplifies to

$$\left\|\boldsymbol{x}_n^K - \boldsymbol{x}^*\right\|^2 \leq \frac{1}{e^2} \cdot e^2 \frac{\kappa H^2 G_*^2}{\mu^2 n^2 K^2} + \frac{4H^2 LG_*^2}{\mu^3 n^2 K^2} \lesssim \frac{H^2 LG_*^2}{\mu^3 n^2 K^2},$$

where we use $\|\boldsymbol{x}_0 - \boldsymbol{x}^*\| \leq e \cdot \frac{\sqrt{\kappa}HG_*}{\mu nK}$.

In particular, using the $L$-smoothness of $F$, the function optimality gap can be bounded as:

$$F(\boldsymbol{x}_n^K) - F(\boldsymbol{x}^*) \leq \frac{L}{2}\left\|\boldsymbol{x}_n^K - \boldsymbol{x}^*\right\|^2 \lesssim \frac{H^2 L^2 G_*^2}{\mu^3 n^2 K^2}.$$

This ends the proof of Theorem 3.7. $\qquad\square$

### C.4. Technical Lemmas

**Lemma C.1.** *Let $p, q \in \mathbb{R}$ with $p < q$, and let $p'$ and $q'$ denote the results of performing a single step of gradient descent on a $\mu$-strongly convex and $L$-smooth 1-dimensional function $f$, starting from $p$ and $q$, respectively, with a step size $\eta < \frac{1}{L}$. Then, it holds that $0 < q' - p' \leq (1 - \eta\mu)(q - p)$.*

*Proof.* Using the gradient descent update rule, we obtain:

$$p' = p - \eta\nabla f(p),$$
$$q' = q - \eta\nabla f(q).$$

The difference between $q'$ and $p'$ can then be written as:

$$
\begin{aligned}
q' - p' &= (q - p) - \eta\left(\nabla f(q) - \nabla f(p)\right) \\
&= (q - p) - \eta\int_p^q \nabla^2 f(u)\,\mathrm{d}u.
\end{aligned}
\tag{25}
$$

Since $\nabla^2 f(u)$ satisfies $\mu \leq \nabla^2 f(u) \leq L$, we have $\mu(q - p) \leq \int_p^q \nabla^2 f(u)\,\mathrm{d}u \leq L(q - p)$. Substituting this inequality to equation (25) yields

$$0 < (1 - \eta L)(q - p) \leq q' - p' \leq (1 - \eta\mu)(q - p),$$

where the first inequality holds due to $\eta < \frac{1}{L}$. $\qquad\square$

**Lemma C.2.** *Let $\{\sigma_k^+\}_{k=1}^K$ denote the sequence of permutations applied over $K$ epochs to generate $z_+^K$. These permutations and the corresponding $z_+^K$ satisfy the following properties:*

- *The permutations $\{\sigma_k^+\}_{k=1}^{K-1}$, applied during the first $K - 1$ epochs, produce $z_+^{K-1}$.*

- *For any $k \in [K]$, all indices in $I_+$ appear before all indices in $I_-$ in the permutation $\sigma_k^+$.*

- *For any $k \in [K]$, let $z_+^{k,I_-}$ denote the $|I_-|$-th iterate in the $k$-th epoch, i.e., obtained after processing all indices in $I_-$. Then, the inequality $z_+^{k,I_-} \leq x^* \leq z_+^k$ holds.*

*For $z_-^K$ and its corresponding permutations $\{\sigma_k^-\}_{k=1}^k$, the following properties hold:*

- *The permutations $\{\sigma_k^-\}_{k=1}^{K-1}$, applied during the first $K-1$ epochs, produce $z_-^{K-1}$.*
- *For any $k \in [K]$, all indices in $I_-$ appear before all indices in $I_+$ in the permutation $\sigma_k^-$.*
- *For any $k \in [K]$, let $z_-^{k,I_+}$ denote the $|I_+|$-th iterate in the $k$-th epoch, i.e., obtained after processing all indices in $I_+$. Then, the inequality $z_-^{k,I_+} \geq x^* \geq z_-^k$ holds.*

*Proof.* We provide the proof for $z_+^K$ and its corresponding permutations $\{\sigma_k^+\}_{k=1}^K$. The proof for $z_-^K$ and $\{\sigma_k^-\}_{k=1}^K$ is analogous, as flipping the sign of $a$'s leads to identical circumstances.

**Step 1: The First Property.** Let $w_+^{K-1}$ denote the iterate obtained by running Algorithm 1 with the sequence of permutations $\{\sigma_k^+\}_{k=1}^{K-1}$, starting from $x^*$ with a step size $\eta$. Since $z_+^{K-1}$ is defined as the maximum possible iterate after running Algorithm 1 with $K-1$ epochs, it follows that $w_+^{K-1} \leq z_+^{K-1}$.

Assume for contradiction that $w_+^{K-1} < z_+^{K-1}$. By Lemma C.1, the iterate obtained by applying $\sigma_K^+$ starting from $z_+^{K-1}$ exceeds $z_+^K$. This contradicts the definition of $z_+^K$, which is the maximum possible final iterate after $K$ epochs. Therefore, we conclude that $w_+^{k-1} = z_+^{k-1}$.

By recursively applying this reasoning, we deduce that for all $l \in [K]$, running Algorithm 1 with permutations $\{\sigma_k^+\}_{k=1}^l$ generates $z_+^l$.

**Step 2: The Second Property.** We now prove the following claim:

**Claim.** *Consider two steps of gradient updates using two component functions $f_i(x)$ and $f_j(x)$ with $a_i < a_j$, starting from the initialization $u$. Then, regardless of the choice of the step size $\eta$, applying $f_i$ first, followed by $f_j$, results in a larger iterate than applying $f_j$ first, followed by $f_i$.*

*Proof of the claim.* The update equations are:

$$u_i = u - \eta\left(\nabla F(u) - a_i\right), \qquad u_j = u - \eta\left(\nabla F(u) - a_j\right),$$
$$u_{ij} = u_i - \eta\left(\nabla F(u_i) - a_j\right), \qquad u_{ji} = u_j - \eta\left(\nabla F(u_j) - a_i\right).$$

Since $a_i < a_j$, we have $u_i < u_j$. Also, because $\nabla F$ is a monotonically increasing function, it follows that $\nabla F(u_i) < \nabla F(u_j)$. Now, we can check that subtracting $u_{ji}$ from $u_{ij}$ yields positive difference:

$$
\begin{aligned}
u_{ij} - u_{ji} &= \left(u_i - \eta\left(\nabla F(u_i) - a_j\right)\right) - \left(u_j - \eta\left(\nabla F(u_j) - a_i\right)\right) \\
&= \eta\left(\nabla F(u_j) - \nabla F(u_i)\right) + \underbrace{\left(u_i + \eta a_j\right) - \left(u_j + \eta a_i\right)}_{=0} \\
&= \eta\left(\nabla F(u_j) - \nabla F(u_i)\right) > 0.
\end{aligned}
$$

Thus, $u_{ji} > u_{ij}$ holds, completing the proof of the claim. $\qquad\square$

From the claim, we conclude in $\sigma_k^+$, all indices in $I_-$ (indices corresponding to negative $a$ values) must appear before indices in $I_+$ (indices corresponding to positive $a$ values). Otherwise, if there exists an index in $I_-$ that immediately follows an index in $I_+$, switching these two indices would result in a larger final iterate (due to Lemma C.1), contradicting the optimality of $\sigma_k^+$. This concludes the proof of the second property.

**Step 3: The Third Property.** Define $M := \sum_{i \in I_+} a_i = -\sum_{i \in I_-} a_i$. We claim that:

**Claim.** *If $0 \leq z_+^k - x^* \leq \eta M$, then $-\eta M \leq z_+^{k+1,I_-} - x^* \leq 0$ holds.*

*Proof of the claim.* Note that the iterate $z_+^{k+1,I_-}$ is obtained by applying gradient update starting from $z_+^k$ using the first $I_-$ component functions of the permutation $\sigma_k$. Let $\sigma_k^f$ denote the first $I_-$ parts of the permutation $\sigma_k$. We verify the bound as follows:

**Lower Bound:** $-\eta M \leq z_+^{k+1,I_-} - x^*$.

By Lemma C.1, the iterate $z_+^{k+1,I_-}$ is at least as large as the iterate obtained by applying gradient updates following $\sigma_k^f$, starting from $x^*$.

Also, if $p < x^*$ holds, then

$$-\nabla F(p) = \nabla F(x^*) - \nabla F(p) = \int_p^{x^*} \nabla^2 F(\alpha) \, d\alpha \le L(x^* - p).$$

Hence, $p - \eta\nabla F(p) \le x^*$ holds. Thus, if the iterate falls below $x^*$, the next iterate obtained by applying the gradient update from the component in $I_-$ will also remain below $x^*$.

This property guarantees that when the gradient update starts $x^*$ and follows $\sigma_k^f$, every iterate remains below $x^*$. Moreover, the total contribution of the gradient updates towards the negative direction by indices in $I_-$ when starting from $x^*$ is at most $-\eta\sum_{i \in I_-} a_i = \eta M$. Hence, $z_+^{k+1,I_-} - x^* \ge -\eta M$ holds.

**Upper Bound:** $z_+^{k+1,I_-} - x^* \le 0.$

Again, by Lemma C.1, the iterate $z_+^{k+1,I_-}$ is at most the iterate obtained by applying gradient updates following $\sigma_k^f$, starting from $x^* + \eta M$.

Assume by contradiction that $z_+^{k+1,I_-} > x^*$ holds. This means that the iterate obtained by following $\sigma_k^f$ starting from $x^* + \eta M$ is also greater than $x^*$. Due to the property stated in the proof of lower bounding $z_+^{k+1,I_-}$, all intermediate iterates should be greater than $x^*$ as well. This leads to a contradiction, as the total contribution of the gradient updates towards the negative direction by indices in $I_-$ when starting from $x^* + \eta M$ will exceed $\eta M$, leading $z_+^{k+1,I_-}$ to fall below $x^*$. Hence, $z_+^{k+1,I_-} - x^* \le 0$ holds.

Combining these two bounds, we obtain

$$-\eta M \le z_+^{k+1,I_-} - x^* \le 0,$$

and this ends the proof of the claim. $\qquad\square$

The claim shows that if $0 \le z_+^k - x^* \le \eta M$, then $-\eta M \le z_+^{k+1,I_-} - x^* \le 0$ holds. By analogous reasoning, if $-\eta M \le z_+^{k+1,I_-} - x^* \le 0$, then $0 \le z_+^{k+1} - x^* \le \eta M$ holds. Combining these two statements, we have: *if* $0 \le z_+^k - x^* \le \eta M$, *then* $0 \le z_+^{k+1} - x^* \le \eta M$ *and* $z_+^{k+1,I_-} \le x^* \le z_+^{k+1}$ *hold.*

Using these, we now proceed by induction to prove the third property. For the base case, the initialization point is $z_0 = x^*$, satisfying the initial condition by $z_0 - x^* = 0$. By induction, it follows that

$$z_+^{k,I_-} \le x^* \le z_+^k.$$

for all $k \in [K]$. This concludes the proof of the third property. $\qquad\square$

**Lemma C.3.** *With a step size $\eta < \frac{1}{L}$, $0 \le \nabla F(u_i) \le 2G_*$ holds for all $i \in \{0\} \cup [|I_+|]$ and $0 \ge \nabla F(v_j) \ge -2G_*$ holds for all $j \in \{0\} \cup [|I_-|]$.*

*Proof.* Recall that the sequence of iterate $\{u_i\}_{i=0}^{|I_+|}$ is defined as $u_0 = x^*$ and each subsequent $u_i$ is obtained by applying a gradient update using the component function $f_{\pi_+(i)}$ to $u_{i-1}$. Specifically, we have

$$u_i = u_{i-1} - \eta\nabla f_{\pi_+(i)}(u_{i-1})$$
$$= u_{i-1} - \eta\left(\nabla F(u_{i-1}) - a_{\pi_+(i)}\right),$$

for $i \in [|I_+|]$.

Now, we will prove by induction that $0 \le \nabla F(u_i) \le 2G_*$ holds for all $i \in [|I_+|]$. Initially, we have $u_0 = x^*$ and thus $\nabla F(u_0) = 0$. Now, assume that $0 \le \nabla F(u_{j-1}) \le 2G_*$. We divide the proof into two cases based on the value of $\nabla F(u_{j-1})$.

**Case 1.** $\nabla F(u_{j-1}) \le a_{\pi_+(j)}.$

In this case, the update equation becomes:

$$u_j = u_{j-1} - \eta \left( \nabla F(u_{j-1}) - a_{\pi_+(j)} \right) \geq u_{j-1},$$

meaning that the iterate increases. Since $\nabla F$ is an increasing function, we have $\nabla F(u_j) \geq \nabla F(u_{j-1}) \geq 0$.

Also, using the fact that all $|a_i|$ is bounded by $G_*$, we can bound the difference of the gradient between successive iterates via the $L$-smoothness of $F$:

$$|\nabla F(u_j) - \nabla F(u_{j-1})| \leq L |u_j - u_{j-1}| \leq \eta L G_* < G_*.$$

Thus, the deviation of $\nabla F(u_j)$ from $\nabla F(u_{j-1})$ is at most $G_*$, leading to the following inequality:

$$\nabla F(u_j) \leq \nabla F(u_{j-1}) + G_* \leq a_{\pi_+(j)} + G_* \leq 2G_*.$$

**Case 2.** $\nabla F(u_{j-1}) > a_{\pi_+(j)}$.

In this case, the update equation becomes:

$$u_j = u_{j-1} - \eta \left( \nabla F(u_{j-1}) - a_{\pi_+(j)} \right) \leq u_{j-1},$$

meaning that the iterate decreases. Since $\nabla F$ is an increasing function, we have $\nabla F(u_j) \leq \nabla F(u_{j-1}) \leq 2G_*$.

Furthermore, by $L$-smoothness of $F$, we have $\nabla F(u_{j-1}) = \nabla F(u_{j-1}) - \nabla F(x^*) \leq L(u_{j-1} - x^*)$. Then, we can ensure that $u_j$ is greater than or equal to $x^*$ as follows:

$$\begin{aligned} u_j &= u_{j-1} - \eta \nabla \left( F(u_{j-1}) - a_{\pi_+(j)} \right) \\ &\geq u_{j-1} - \eta \nabla F(u_{j-1}) \\ &\geq u_{j-1} - \frac{1}{L} \cdot L(u_{j-1} - x^*) = x^*. \end{aligned}$$

For both cases, we have shown that $0 \leq \nabla F(u_j) \leq 2G_*$.

We can apply the same approach for $\{v_i\}_{i=1}^{|I_-|}$. The key difference is that the sign of $a_{\pi_-(j)}$ is negative. This leads to the result $0 \geq \nabla F(v_j) \geq -2G_*$ for all $j \in [|I_-|]$. This concludes the proof of Lemma C.3. □

# D. Proofs for Large Epoch Lower Bounds

## D.1. Proof of Theorem 4.1

**Theorem 4.1.** *For any $n \geq 2$, $\kappa \geq 2$, and $K \geq \kappa$, there exists a 3-dimensional function $F$ satisfying Assumptions 2.3 and 2.4 with $P = 0$, where each component function shares the same Hessian, along with an initialization point $\boldsymbol{x}_0$, such that for any constant step size $\eta$, the final iterate obtained by running Algorithm 2 satisfies*

$$F(\boldsymbol{x}_n^K) - F(\boldsymbol{x}^*) \gtrsim \frac{LG^2}{\mu^2 K^2}.$$

*Proof.* Similar to the approach in Theorem 3.1, we divide the range of step size into three regimes. For each regime, we construct the overall function $F_1$, $F_2$, and $F_3$, respectively, along with their respective component functions and an initial point. Finally, we aggregate these functions across different dimensions to derive the stated lower bound.

Each overall function is 1-dimensional, and carefully designed to satisfy the following properties:

- (Small step size regime) There exists an initialization point $x_0 = \text{poly}(\mu, L, n, K, G)$ such that for any choice of $\eta \in \left(0, \frac{1}{\mu n K}\right)$, the final iterate $x_n^K$ obtained by running Algorithm 2 satisfies $F_1(x_n^K) - F_1(x^*) \gtrsim \frac{LG^2}{\mu^2 K^2}$.

- (Moderate step size regime) There exists an initialization point $y_0 = \text{poly}(\mu, L, n, K, G)$ such that for any choice of $\eta \in \left[\frac{1}{\mu n K}, \frac{2}{L}\right)$, the final iterate $y_n^K$ obtained by running Algorithm 2 satisfies $F_2(y_n^K) - F_2(y^*) \gtrsim \frac{LG^2}{\mu^2 K^2}$.

- (Large step size regime) There exists an initialization point $z_0 = \text{poly}(\mu, L, n, K, G)$ such that for any choice of $\eta \in \left[\frac{2}{L}, \infty\right)$, the final iterate $z_n^K$ obtained by running Algorithm 2 satisfies $F_3(z_n^K) - F_3(z^*) \gtrsim \frac{LG^2}{\mu^2 K^2}$.

Here, $x^*$, $y^*$, $z^*$ denote the minimizers of $F_1$, $F_2$, and $F_3$, respectively. All these functions are designed to satisfy Assumption 2.3. $F_1$ and $F_3$ satisfy Assumption 2.4 with $G = P = 0$, and $F_2$ satisfies with $P = 0$. Moreover, each component function within each overall function shares the same Hessian. Detailed constructions of $F_1$, $F_2$, and $F_3$, as well as the verification of the assumptions and the stated properties are presented in Appendices D.1.1 to D.1.3.

By following a similar approach to the proof of Theorems 3.1 and 3.3, we can conclude that the aggregated 3-dimensional function $F(\boldsymbol{x}) := F(x, y, z) = F_1(x) + F_2(y) + F_3(z)$ and its component functions satisfy the stated assumptions. Also, since each dimension is independent, it is obvious that $\boldsymbol{x}^* = (x^*, y^*, z^*)$ minimizes $F$. Finally, by choosing the initialization point as $\boldsymbol{x}_0 = (x_0, y_0, z_0)$, the final iterate $\boldsymbol{x}_n^K = (x_n^K, y_n^K, z_n^K)$ obtained by running Algorithm 2 on $F$ satisfies

$$F(\boldsymbol{x}_n^K) - F(\boldsymbol{x}^*) \gtrsim \frac{LG^2}{\mu^2 K^2},$$

regardless of the choice of $\eta > 0$.

This concludes the proof of Theorem 4.1. $\qquad\square$

In the following subsections, we present the specific construction of $F_1$, $F_2$, and $F_3$, and demonstrate that each satisfies the stated lower bound within its corresponding step size regime. For simplicity of notation, we omit the index of the overall function when referring to its component functions, e.g., we write $f_i(x)$ instead of $f_{1i}(x)$. Moreover, we use the common variable notation $x$ while constructing functions for each dimension, though we use different variables in the "dimension-aggregation" step.

### D.1.1. CONSTRUCTION OF $F_1$

Let $F_1(x) = \frac{\mu}{2} x^2$ with component functions $f_i(x) = F_1(x)$ for all $i \in [n]$. It is clear that $F_1$ satisfies Assumption 2.3 and Assumption 2.4 with $G = P = 0$, and its component functions share an identical Hessian. Also, we note that $x^* = 0$ and $F_1(x^*) = 0$.

Let the initialization be $x_0 = \sqrt{\kappa} \frac{G}{\mu K}$. For all $\eta \in \left(0, \frac{1}{\mu n K}\right)$, the final iterate is given by

$$x_n^K = (1 - \eta\mu)^{nK} x_0 \geq \left(1 - \frac{1}{nK}\right)^{nK} x_0 \geq \frac{\sqrt{\kappa}G}{4\mu K},$$

where the last inequality uses the fact that $(1 - \frac{1}{m})^m \geq \frac{1}{4}$ for all $m \geq 2$.

Thus, we have

$$F_1(x_n^K) - F_1(x^*) = \frac{\mu}{2}(x_n^K)^2 \gtrsim \frac{LG^2}{\mu^2 K^2}.$$

### D.1.2. CONSTRUCTION OF $F_2$

In this subsection, we let $L'$ denote $L/2$. We construct the function by dividing the cases by the parity of $n$. We first consider the case where $n$ is even, and address the case where $n$ is odd later in this subsection. Let $F_2(x) = \frac{L'}{2}x^2$ with component functions

$$f_i(x) = \begin{cases} \frac{L'}{2}x^2 + Gx & \text{if } i \leq n/2, \\ \frac{L'}{2}x^2 - Gx & \text{otherwise.} \end{cases}$$

Since $\kappa \geq 2$, we have $L' = \frac{L}{2} \geq \mu$. Thus, it is clear that $f_i$ satisfies Assumptions 2.3 and 2.4 with $P = 0$, and shares the same Hessian. By Lemma F.1, the final iterate obtained by running Algorithm 2 is given by

$$x_n^K = (1 - \eta L')^{nK} x_0 + \frac{G}{L'} \cdot \frac{1 - (1 - \eta L')^{\frac{n}{2}}}{1 + (1 - \eta L')^{\frac{n}{2}}} \left(1 - (1 - \eta L')^{nK}\right).$$

By applying $\eta \geq \frac{1}{\mu n K}$ and setting $x_0 = 0$, we derive

$$\begin{aligned} x_n^K &= \frac{G}{L'} \cdot \frac{1 - (1 - \eta L')^{\frac{n}{2}}}{1 + (1 - \eta L')^{\frac{n}{2}}} \left(1 - (1 - \eta L')^{nK}\right) \\ &\geq \frac{G}{2L'} \left(1 - (1 - \eta L')^{\frac{n}{2}}\right) \left(1 - (1 - \eta \mu)^{nK}\right) \\ &\geq \frac{G}{2L'} \left(1 - (1 - \eta L')^{\frac{n}{2}}\right) \left(1 - \left(1 - \frac{1}{nK}\right)^{nK}\right) \\ &\geq \frac{G}{2L'} \left(1 - (1 - \eta L')^{\frac{n}{2}}\right) \left(1 - e^{-1}\right). \end{aligned} \tag{26}$$

We analyze equation (26) by dividing the range of $\eta$ into two regimes.

**Regime 1.** $\eta \in \left[\frac{1}{\mu n K}, \frac{1}{nL'}\right)$.

In this regime, we can bound $1 - (1 - \eta L')^{\frac{n}{2}}$ as:

$$1 - (1 - \eta L')^{\frac{n}{2}} \geq 1 - e^{-\frac{\eta n L'}{2}} \geq 1 - \left(1 - \frac{\eta n L'}{4}\right) = \frac{\eta n L'}{4} \geq \frac{L'}{4\mu K},$$

where the second inequality uses $e^{-u} \leq 1 - \frac{u}{2}$ for all $u \in [0, 1]$. Substituting this inequality into equation (26) gives

$$x_n^K \geq \frac{(1 - e^{-1})G}{8\mu K}.$$

Consequently, the function optimality gap satisfies

$$F_2(x_n^K) - F_2(x^*) = \frac{L'}{2}(x_n^K)^2 \gtrsim \frac{LG^2}{\mu^2 K^2}.$$

**Regime 2.** $\eta \in \left[\frac{1}{nL'}, \frac{1}{L'}\right)$.

In this regime, we can bound $1 - (1 - \eta L')^{\frac{n}{2}}$ as:

$$1 - (1 - \eta L')^{\frac{n}{2}} \geq 1 - \left(1 - \frac{1}{n}\right)^{\frac{n}{2}} \geq 1 - e^{-\frac{1}{2}}.$$

Substituting this inequality into equation (26) gives

$$x_n^K \geq \frac{\left(1 - e^{-1}\right)\left(1 - e^{-\frac{1}{2}}\right) G}{2L'}.$$

Since $K \geq \kappa$, we have $\frac{1}{L'} = \frac{2}{L} \geq \frac{2}{\mu K}$. Therefore, the final iterate $x_n^K$ can be bounded as:

$$x_n^K \geq \frac{\left(1 - e^{-1}\right)\left(1 - e^{-\frac{1}{2}}\right) G}{\mu K}.$$

Consequently, the function optimality gap satisfies

$$F_2(x_n^K) - F_2(x^*) = \frac{L'}{2}(x_n^K)^2 \gtrsim \frac{LG^2}{\mu^2 K^2}.$$

We now focus on the case where $n$ is odd. Let $F_2(x) = \frac{L'}{2} x^2$ with component functions

$$f_i(x) = \begin{cases} \frac{L'}{2} x^2 & \text{if } i = 1, \\ \frac{L'}{2} x^2 + Gx & \text{if } 2 \leq i \leq (n+1)/2, \\ \frac{L'}{2} x^2 - Gx & \text{if } (n+3)/2 \leq i \leq n. \end{cases}$$

Compared to the case of even $n$, $f_1(x) = \frac{L'}{2} x^2$ is introduced newly. By Lemma F.2, the final iterate $x_n^K$ obtained by running Algorithm 2 satisfies the following equation:

$$x_n^K = (1 - \eta L')^{nK} x_0 + \frac{G}{L'} \cdot \frac{1 - (1 - \eta L')^{nK}}{1 - (1 - \eta L')^n} \left(1 - (1 - \eta L')^{\frac{n-1}{2}}\right)^2.$$

By applying $\eta \geq \frac{1}{\mu n K}$ and setting $x_0 = 0$, we have

$$\begin{aligned} x_n^K &= \frac{G}{L'} \cdot \frac{1 - (1 - \eta L')^{nK}}{1 - (1 - \eta L')^n} \left(1 - (1 - \eta L')^{\frac{n-1}{2}}\right)^2 \\ &= \frac{G}{L'} \left(1 - (1 - \eta L')^{nK}\right) \frac{1 - (1 - \eta L')^{n-1}}{1 - (1 - \eta L')^n} \frac{\left(1 - (1 - \eta L')^{\frac{n-1}{2}}\right)^2}{1 - (1 - \eta L')^{n-1}} \\ &\geq \frac{G}{L'} \left(1 - (1 - \eta \mu)^{nK}\right) \frac{1 - (1 - \eta L')^{n-1}}{1 - (1 - \eta L')^n} \frac{1 - (1 - \eta L')^{\frac{n-1}{2}}}{1 + (1 - \eta L')^{\frac{n-1}{2}}} \\ &\geq \frac{G}{2L'} \left(1 - e^{-\eta \mu n K}\right) \frac{1 - (1 - \eta L')^{n-1}}{1 - (1 - \eta L')^n} \left(1 - (1 - \eta L')^{\frac{n-1}{2}}\right) \\ &\geq \frac{G}{2L'} \left(1 - e^{-1}\right) \frac{1 - (1 - \eta L')^{n-1}}{1 - (1 - \eta L')^n} \left(1 - (1 - \eta L')^{\frac{n-1}{2}}\right). \end{aligned}$$

Note that the inequality

$$\frac{1 - (1 - \eta L')^{n-1}}{1 - (1 - \eta L')^n} \geq \frac{1}{2}$$

holds for $n \geq 2$ since

$$
\begin{aligned}
2 - 2(1 - \eta L')^{n-1} \geq 1 - (1 - \eta L')^n &\Leftrightarrow 1 \geq 2(1 - \eta L')^{n-1} - (1 - \eta L')^n \\
&\Leftrightarrow 1 \geq (1 - \eta L')^{n-1}(2 - (1 - \eta L')) \\
&\Leftrightarrow 1 \geq (1 - \eta L')^{n-2}(1 - \eta^2 L'^2).
\end{aligned}
$$

Hence, we deduce that

$$
x_n^K \geq \frac{G}{4L'} \left(1 - e^{-1}\right) \left(1 - (1 - \eta L')^{\frac{n-1}{2}}\right). \tag{27}
$$

We again analyze equation (27) by dividing the range of $\eta$ into two regimes.

**Regime 1.** $\eta \in \left[\frac{1}{\mu n K}, \frac{1}{n L'}\right)$.

In this regime, we can bound $1 - (1 - \eta L')^{\frac{n-1}{2}}$ as:

$$
1 - (1 - \eta L')^{\frac{n-1}{2}} \geq 1 - e^{-\frac{\eta(n-1)L'}{2}} \geq 1 - \left(1 - \frac{\eta(n-1)L'}{4}\right) = \frac{\eta(n-1)L'}{4} \geq \frac{\eta n L'}{8} \geq \frac{L'}{8\mu K},
$$

where the second inequality uses $e^{-u} \leq 1 - \frac{u}{2}$ for all $u \in [0, 1]$. Substituting this inequality into equation (27) gives

$$
x_n^K \geq \frac{\left(1 - e^{-1}\right) G}{32 \mu K}.
$$

Consequently, the function optimality gap satisfies

$$
F_2(x_n^K) - F_2(x^*) = \frac{L'}{2}(x_n^K)^2 \gtrsim \frac{LG^2}{\mu^2 K^2}.
$$

**Regime 2.** $\eta \in \left[\frac{1}{n L'}, \frac{1}{L'}\right)$.

In this regime, we can bound $1 - (1 - \eta L')^{\frac{n-1}{2}}$ as:

$$
1 - (1 - \eta L')^{\frac{n-1}{2}} \geq 1 - \left(1 - \frac{1}{n}\right)^{\frac{n-1}{2}} \geq 1 - e^{-\frac{n-1}{2n}} \geq 1 - e^{-\frac{1}{4}}.
$$

Substituting this inequality into equation (27) gives

$$
x_n^K \geq \frac{\left(1 - e^{-1}\right)\left(1 - e^{-\frac{1}{4}}\right) G}{4L'}.
$$

Since $K \geq \kappa$, we have $\frac{1}{L'} = \frac{2}{L} \geq \frac{2}{\mu K}$. Therefore, the final iterate $x_n^K$ can be bounded as:

$$
x_n^K \geq \frac{\left(1 - e^{-1}\right)\left(1 - e^{-\frac{1}{4}}\right) G}{2\mu K}.
$$

Consequently, the function optimality gap satisfies

$$
F_2(x_n^K) - F_2(x^*) = \frac{L'}{2}(x_n^K)^2 \gtrsim \frac{LG^2}{\mu^2 K^2}.
$$

### D.1.3. CONSTRUCTION OF $F_3$

Let $F_3(x) = \frac{L}{2}x^2$ with component functions $f_i(x) = F_3(x)$ for all $i \in [n]$. It is clear that $F_1$ satisfies Assumption 2.3, Assumption 2.4 with $G = P = 0$ and its component functions share an identical Hessian. Also, we note that $x^* = 0$ and $F_3(x^*) = 0$.

For all $\eta \in \left[\frac{2}{L}, \infty\right)$, the final iterate is given by

$$x_n^K = (1 - \eta L)^{nK} x_0.$$

In this regime, the step size is excessively large, resulting in

$$1 - \eta L \leq 1 - \frac{2}{L} \cdot L \leq -1,$$

which implies $\left|(1 - \eta L)^{nK}\right| \geq 1$. Thus, the iterate does not converge and satisfies $\left|x_n^K\right| \geq |x_0|$.

By setting the initialization $x_0 = \frac{G}{\mu K}$, we have

$$F_3(x_n^K) - F_3(x^*) = \frac{L}{2}(x_n^K)^2 \geq \frac{L}{2}(x_0)^2 \gtrsim \frac{LG^2}{\mu^2 K^2}.$$

## D.2. Proof of Theorem 4.3

**Theorem 4.3.** *For any $n \geq 4$, $\kappa \geq n$, and $K \geq \max\left\{\kappa^3/n^2, \kappa^{3/2}\right\}$, there exists a 4-dimensional function $F$ satisfying Assumptions 2.3 and 2.4 with $P = \kappa$, along with an initialization point $\boldsymbol{x}_0$, such that for any constant step size $\eta$, the final iterate obtained by running Algorithm 2 satisfies*

$$F(\boldsymbol{x}_n^K) - F(\boldsymbol{x}^*) \gtrsim \frac{L^2 G^2}{\mu^3 K^2}.$$

*Proof.* Similar to the approach in Theorem 3.1, we divide the range of step sizes. However, unlike the previous theorems where the range is divided into three regimes, we divide the range into four regimes in this case. For each regime, we construct the overall functions $F_1, F_2, F_3$, and $F_4$, along with their respective component functions and an initial point. Finally, we aggregate these functions across different dimensions to derive the stated lower bound.

Each function is 1-dimensional, and carefully designed to satisfy the following properties:

- (Small step size regime) There exists an initial point $x_0 = \text{poly}(\mu, L, n, K, G)$ such that for any choice of $\eta \in \left(0, \frac{1}{\mu n K}\right)$, the final iterate $x_n^K$ obtained by running Algorithm 2 satisfies $F_1(x_n^K) - F_1(x^*) \gtrsim \frac{L^2 G^2}{\mu^3 K^2}$,

- (Moderate step size regime 1) There exists an initial point $y_0 = \text{poly}(\mu, L, n, K, G)$ such that for any choice of $\eta \in \left[\frac{1}{\mu n K}, \frac{1}{n L}\right)$, the final iterate $y_n^K$ obtained by running Algorithm 2 satisfies $F_2(y_n^K) - F_2(y^*) \gtrsim \frac{L^2 G^2}{\mu^3 K^2}$.

- (Moderate step size regime 2) There exists an initial point $z_0 = \text{poly}(\mu, L, n, K, G)$ such that for any choice of $\eta \in \left[\frac{1}{n L}, \frac{2}{L}\right)$, the final iterate $z_n^K$ obtained by running Algorithm 2 satisfies $F_3(z_n^K) - F_3(z^*) \gtrsim \frac{L^2 G^2}{\mu^3 K^2}$.

- (Large step size regime) There exists an initial point $w_0 = \text{poly}(\mu, L, n, K, G)$ such that for any choice of $\eta \in \left[\frac{1}{L}, \infty\right)$, the final iterate $w_n^K$ obtained by running Algorithm 2 satisfies $F_4(w_n^K) - F_4(w^*) \gtrsim \frac{L^2 G^2}{\mu^3 K^2}$.

Here, $x^*$, $y^*$, $z^*$, and $w^*$ denote the minimizers of $F_1, F_2, F_3$, and $F_4$, respectively. All these functions are designed to satisfy Assumption 2.3. $F_1$ and $F_4$ satisfy Assumption 2.4 with $G = P = 0$, $F_3$ satisfies with $P = 0$, and $F_2$ satisfies with $P = \kappa$. Detailed constructions for $F_1$ through $F_4$, as well as the verification of the assumptions and the stated properties are presented in Appendices D.2.1 to D.2.4.

By following a similar approach to the proof of Theorems 3.1 and 3.3, we can conclude that the aggregated 4-dimensional function $F(\boldsymbol{x}) := F_1(x) + F_2(y) + F_3(z) + F_4(w)$ satisfy the stated assumptions (additional scalar in $G$ can be absorbed

by rescaling $G$ in each overall function). Also, since each dimension is independent, it is obvious that $\boldsymbol{x}^* = (x^*, y^*, z^*, w^*)$ minimizes $F$. Finally, by choosing the initial point as $\boldsymbol{x}_0 = (x_0, y_0, z_0, w_0)$, the final iterate $\boldsymbol{x}_n^K = (x_n^K, y_n^K, z_n^K, w_n^K)$ obtained by running Algorithm 2 on $F$ satisfies

$$F(\boldsymbol{x}_n^K) - F(\boldsymbol{x}^*) \gtrsim \frac{L^2 G^2}{\mu^3 K^2},$$

regardless of the choice of $\eta > 0$.

This concludes the proof of Theorem 4.3. $\hfill\square$

In the following subsections, we present the specific construction of $F_1$, $F_2$, $F_3$, and $F_4$, and demonstrate that each satisfies the stated lower bound within its corresponding step size regime. For simplicity of notation, we omit the index of the overall function when referring to its component functions, e.g., we write $f_i(x)$ instead of $f_{1i}(x)$. Moreover, we use the common variable notation $x$ while constructing functions for each dimension, though we use different variables in the "dimension-aggregation" step.

### D.2.1. CONSTRUCTION OF $F_1$

Let $F_1(x) = \frac{\mu}{2} x^2$ with component functions $f_i(x) = F_1(x)$ for all $i \in [n]$. It is clear that $F_1$ satisfies Assumption 2.3 and Assumption 2.4 with $G = P = 0$. Also, we note that $x^* = 0$ and $F_1(x^*) = 0$.

Let the initialization be $x_0 = \frac{LG}{\mu^2 K}$. For all $\eta \in \left(0, \frac{1}{\mu n K}\right)$, the final iterate is given by

$$x_n^K = (1 - \eta\mu)^{nK} x_0 \geq \left(1 - \frac{1}{nK}\right)^{nK} x_0 \geq \frac{LG}{4\mu^2 K},$$

where the last inequality uses the fact that $(1 - \frac{1}{m})^m \geq \frac{1}{4}$ for all $m \geq 2$.

Thus, we have

$$F_1(x_n^K) - F_1(x^*) = \frac{\mu}{2}(x_n^K)^2 \gtrsim \frac{L^2 G^2}{\mu^3 K^2}.$$

### D.2.2. CONSTRUCTION OF $F_2$

In this section, we focus on the case when $n$ is a multiple of 4. Otherwise, we set $4\lfloor \frac{n}{4} \rfloor$ components satisfying the argument, and introduce at most three zero component functions. This adjustment does not affect the final result, but only modifies the parameters $\mu$ and $L$ by at most a constant factor.

Let $F_2(x) = \frac{\mu}{2} x^2$ with component functions

$$f_i(x) = \begin{cases} Gx & \text{if } 1 \leq i \leq n/4, \\ \frac{L}{2}x^2 & \text{if } n/4 + 1 \leq i \leq n/2, \\ -Gx & \text{if } n/2 + 1 \leq i \leq 3n/4, \\ -\frac{L-4\mu}{2}x^2 & \text{if } 3n/4 + 1 \leq i \leq n. \end{cases}$$

For simplicity of the notation, let $a$ denote $L - 4\mu$. Since $\kappa \geq 4$, we have $0 \leq a < L$. Thus, each $f_i$ is $L$-smooth, ensuring that the construction satisfies Assumption 2.3. The gradient difference between the component function $f_i$ and the overall function $F_2$ is bounded as

$$\|\nabla f_i(x) - \nabla F_2(x)\| \leq \begin{cases} \|\mu x\| + G & \text{if } 1 \leq i \leq n/4 \text{ or } n/2 + 1 \leq i \leq 3n/4, \\ \|(L - \mu)x\| & \text{if } n/4 + 1 \leq i \leq n/2 \text{ or } 3n/4 + 1 \leq i \leq n. \end{cases}$$

Since $\nabla F_2(x) = \mu x$, it follows that $\|(L - \mu)x\| < \kappa \|\nabla F_2(x)\|$. Therefore, the construction satisfies Assumption 2.4 with $P = \kappa$. Additionally, we note that $x^* = 0$ and $F_2(x^*) = 0$. Using these component functions, we first derive the closed-form expression for the iterates obtained by running Algorithm 2:

$$x_{n/4}^k = x_0^k - \frac{\eta n G}{4},$$

$$x_{n/2}^k = (1 - \eta L)^{\frac{n}{4}} x_{n/4}^k = (1 - \eta L)^{\frac{n}{4}} x_0^k - (1 - \eta L)^{\frac{n}{4}} \frac{\eta n G}{4},$$

$$x_{3n/4}^k = x_{n/2}^k + \frac{\eta n G}{4} = (1 - \eta L)^{\frac{n}{4}} x_0^k + \left(1 - (1 - \eta L)^{\frac{n}{4}}\right) \frac{\eta n G}{4},$$

$$x_n^k = (1 + \eta a)^{\frac{n}{4}} x_{3n/4}^k = (1 + \eta a)^{\frac{n}{4}} (1 - \eta L)^{\frac{n}{4}} x_0^k + (1 + \eta a)^{\frac{n}{4}} \left(1 - (1 - \eta L)^{\frac{n}{4}}\right) \frac{\eta n G}{4}.$$

Let $p := (1 - \eta L)^{\frac{n}{4}}$ and $q := (1 + \eta a)^{\frac{n}{4}}$. Using these definitions, the epoch-wise recursion equation can be expressed as:

$$x_0^{k+1} = pq x_0^k + q(1 - p)\frac{\eta n G}{4}.$$

By unrolling the above equation over $k \in [K]$, we obtain the final iterate $x_n^K$:

$$x_n^K = (pq)^K x_0 + \frac{1 - (pq)^K}{1 - pq} \cdot q(1 - p)\frac{\eta n G}{4}. \tag{28}$$

We now state key inequalities regarding $p$ and $q$:

**Lemma D.1.** *Under the conditions $K \geq \kappa \geq n \geq 3$, the following inequalities hold for $\eta \in \left[\frac{1}{\mu n K}, \frac{1}{nL}\right)$:*

1. $1 - p \geq \begin{cases} \frac{L}{8\mu K} & \text{if } \eta \in \left[\frac{1}{\mu n K}, \frac{\mu}{L^2}\right), \\ \frac{n\mu}{8L} & \text{if } \eta \in \left[\frac{\mu}{L^2}, \frac{1}{nL}\right). \end{cases}$

2. $1 - (pq)^K \geq 1 - e^{-1}.$

3. $\frac{1}{1-pq} \geq \begin{cases} \frac{4}{5\eta n \mu} & \text{if } \eta \in \left[\frac{1}{\mu n K}, \frac{\mu}{L^2}\right), \\ \frac{4}{5\eta^2 n L^2} & \text{if } \eta \in \left[\frac{\mu}{L^2}, \frac{1}{nL}\right). \end{cases}$

The proof of Lemma D.1 is presented in Appendix D.3. Setting the initialization point $x_0 = 0$, equation (28) simplifies to

$$x_n^K = \frac{1 - (pq)^K}{1 - pq} \cdot q(1 - p)\frac{\eta n G}{4} \geq \frac{1 - e^{-1}}{1 - pq} \cdot 1 \cdot (1 - p) \cdot \frac{\eta n G}{4} = \frac{1 - e^{-1}}{4} \cdot \frac{1 - p}{1 - pq} \cdot \eta n G. \tag{29}$$

Now, we divide the range of step size into two regimes: $\left[\frac{1}{\mu n K}, \frac{\mu}{L^2}\right)$ and $\left[\frac{\mu}{L^2}, \frac{1}{nL}\right)$.

**Regime 1.** $\eta \in \left[\frac{1}{\mu n K}, \frac{\mu}{L^2}\right)$.

In this regime, we have $1 - p \geq \frac{L}{8\mu K}$ and $\frac{1}{1-pq} \geq \frac{4}{5\eta n \mu}$. Substituting these inequalities to equation (29) results

$$x_n^K \geq \frac{\left(1 - e^{-1}\right) LG}{40\mu^2 K}.$$

**Regime 2.** $\eta \in \left[\frac{\mu}{L^2}, \frac{1}{nL}\right)$.

In this regime, we have $1 - p \geq \frac{n\mu}{8L}$ and $\frac{1}{1-pq} \geq \frac{4}{5\eta^2 n L^2}$. Substituting these inequalities to equation (29) results

$$x_n^K \geq \frac{1 - e^{-1}}{40} \cdot \frac{n\mu G}{\eta L^3} \geq \frac{1 - e^{-1}}{40} \cdot \frac{n^2 \mu G}{L^2}.$$

Using the assumption $K \geq \frac{\kappa^3}{n^2}$, it follows that $n^2 \geq \frac{\kappa^3}{K}$, resulting

$$x_n^K \geq \frac{\left(1 - e^{-1}\right) LG}{40\mu^2 K}.$$

Combining the results for the two subdivided step size regimes, we have

$$x_n^K \geq \frac{\left(1 - e^{-1}\right) LG}{40\mu^2 K}.$$

for all $\eta \in \left[\frac{1}{\mu n K}, \frac{1}{nL}\right)$.

Finally, the function optimality gap is

$$F_2(x_n^K) - F_2(x^*) = \frac{\mu}{2} \left(x_n^K\right)^2 \gtrsim \frac{L^2 G^2}{\mu^3 K^2}.$$

### D.2.3. CONSTRUCTION OF $F_3$

We focus on the case where $n$ is even. If $n$ is odd, we introduce an additional zero component function. This does not affect the final result but only modifies each parameter at most by a constant factor.

In this subsection, we let $L'$ denote $L/2$. Let $F_3 = \frac{L'}{2}x^2$ with component functions

$$f_i(x) = \begin{cases} \frac{L'}{2}x^2 + Gx & \text{if } i \leq n/2, \\ \frac{L'}{2}x^2 - Gx & \text{otherwise.} \end{cases}$$

It is clear that each $f_i$ is $L$-smooth. Since $\kappa \geq 2$, we have $L' = \frac{L}{2} \geq \mu$. Thus, $F_3$ is $\mu$-strongly convex, satisfying Assumption 2.3. Also, we can easily verify that the construction satisfies Assumption 2.4 with $P = 0$. We note that $x^* = 0$ and $F_3(x^*) = 0$.

By Lemma F.1, the final iterate obtained by running Algorithm 2 is given by

$$x_n^K = (1 - \eta L')^{nK} x_0 + \frac{G}{L'} \cdot \frac{1 - (1 - \eta L')^{\frac{n}{2}}}{1 + (1 - \eta L')^{\frac{n}{2}}} \left(1 - (1 - \eta L')^{nK}\right).$$

Recall that $\frac{1}{nL} = \frac{1}{2nL'}$ and $\frac{2}{L} = \frac{1}{L'}$. Since $\eta \in \left[\frac{1}{2nL'}, \frac{1}{L'}\right)$, it follows that

$$(1 - \eta L')^{\frac{n}{2}} \leq \left(1 - \frac{1}{2n}\right)^{\frac{n}{2}} \leq e^{-\frac{1}{4}}, \text{ and } (1 - \eta L')^{nK} \leq e^{-\frac{K}{4}}.$$

Using these inequalities and setting the initialization as $x_0 = 0$, the final iterate $x_n^K$ is expressed as:

$$x_n^K = \frac{G}{L'} \cdot \frac{1 - (1 - \eta L')^{\frac{n}{2}}}{1 + (1 - \eta L')^{\frac{n}{2}}} \left(1 - (1 - \eta L')^{nK}\right) \geq \frac{G}{L'} \frac{1 - e^{-\frac{1}{4}}}{2} \left(1 - e^{-\frac{K}{4}}\right) \geq \frac{G}{L'} \frac{\left(1 - e^{-\frac{1}{4}}\right)^2}{2}.$$

Finally, the function optimality gap becomes

$$F_3(x_n^K) - F_3(x^*) = \frac{L'}{2} \left(x_n^K\right)^2 \gtrsim \frac{G^2}{L} \geq \frac{L^2 G^2}{\mu^3 K^2},$$

where the last inequality holds since $K \geq \kappa^{3/2}$.

### D.2.4. CONSTRUCTION OF $F_4$

Let $F_4(x) = \frac{L}{2}x^2$ with component functions $f_i(x) = F_4(x)$ for all $i \in [n]$. It is clear that $F_4$ satisfies Assumption 2.3, Assumption 2.4 with $G = P = 0$. Also, we note that $x^* = 0$ and $F_4(x^*) = 0$.

For all $\eta \in \left[\frac{2}{L}, \infty\right)$, the final iterate is given by

$$x_n^K = (1 - \eta L)^{nK} x_0.$$

In this regime, the step size is excessively large, resulting in

$$1 - \eta L \leq 1 - \frac{2}{L} \cdot L \leq -1,$$

which implies $\left|(1 - \eta L)^{nK}\right| \geq 1$. Thus, the iterate does not converge and satisfies $\left|x_n^K\right| \geq |x_0|$.

By setting the initialization $x_0 = \sqrt{\kappa} \frac{G}{\mu K}$, we have

$$F_4(x_n^K) - F_4(x^*) = \frac{L}{2}(x_n^K)^2 \geq \frac{L}{2}(x_0)^2 \gtrsim \frac{L^2 G^2}{\mu^3 K^2}.$$

### D.3. Technical Lemmas

**Lemma D.1.** *Under the conditions $K \geq \kappa \geq n \geq 3$, the following inequalities hold for $\eta \in \left[\frac{1}{\mu n K}, \frac{1}{nL}\right)$:*

1. $1 - p \geq \begin{cases} \frac{L}{8\mu K} & \text{if } \eta \in \left[\frac{1}{\mu n K}, \frac{\mu}{L^2}\right), \\ \frac{n\mu}{8L} & \text{if } \eta \in \left[\frac{\mu}{L^2}, \frac{1}{nL}\right). \end{cases}$

2. $1 - (pq)^K \geq 1 - e^{-1}$.

3. $\frac{1}{1-pq} \geq \begin{cases} \frac{4}{5\eta n\mu} & \text{if } \eta \in \left[\frac{1}{\mu n K}, \frac{\mu}{L^2}\right), \\ \frac{4}{5\eta^2 n L^2} & \text{if } \eta \in \left[\frac{\mu}{L^2}, \frac{1}{nL}\right). \end{cases}$

*Proof.* Recall the definitions of $p$ and $q$:

$$p = (1 - \eta L)^{\frac{n}{4}},$$
$$q = (1 + \eta a)^{\frac{n}{4}} = (1 + \eta(L - 4\mu))^{\frac{n}{4}}.$$

To prove the first inequality, we divide the range of step size into two regimes: $\left[\frac{1}{\mu n K}, \frac{\mu}{L^2}\right)$ and $\left[\frac{\mu}{L^2}, \frac{1}{nL}\right)$. Note that the first regime may be empty depending on the condition on $K$, but remains valid (i.e. $\frac{1}{\mu n K} \leq \frac{\mu}{L^2}$) under the condition $K \geq \kappa^2/n$ in the current theorem.

**Regime 1.** $\eta \in \left[\frac{1}{\mu n K}, \frac{\mu}{L^2}\right)$.

In this regime, we can bound $p$ as:

$$p = (1 - \eta L)^{\frac{n}{4}} \leq \left(1 - \frac{L}{\mu n K}\right)^{\frac{n}{4}} \leq e^{-\frac{L}{4\mu K}} \leq 1 - \frac{L}{8\mu K}.$$

Here, the first step holds because $\eta \geq \frac{1}{\mu n K}$. In the final step, we utilize the inequalities $\frac{L}{4\mu K} < 1$ and $e^{-u} \leq 1 - \frac{1}{2}u$ for all $u \in [0, 1]$. Hence, we can obtain $1 - p \geq \frac{L}{8\mu K}$.

**Regime 2.** $\eta \in \left[\frac{\mu}{L^2}, \frac{1}{nL}\right)$.

In this regime, we can bound $p$ as:

$$p = (1 - \eta L)^{\frac{n}{4}} \leq \left(1 - \frac{\mu}{L}\right)^{\frac{n}{4}} \leq e^{-\frac{n\mu}{4L}} \leq 1 - \frac{n\mu}{8L}.$$

Here, the first step holds because $\eta \geq \frac{\mu}{L^2}$. At the final step, we utilize the inequalities $\frac{n\mu}{4L} < 1$ and $e^{-u} \leq 1 - \frac{1}{2}u$ for all $u \in [0, 1]$. Hence, we can obtain $1 - p \geq \frac{n\mu}{8L}$.

To bound $1 - (pq)^K$, we first establish bounds for $pq$:

$$pq = (1 - \eta L)^{\frac{n}{4}}(1 + \eta a)^{\frac{n}{4}} \leq e^{-\frac{\eta n L}{4}} \cdot e^{\frac{\eta n a}{4}} = e^{-\frac{\eta n(L-a)}{4}} = e^{-\eta n\mu} \leq e^{-\frac{1}{K}},$$

where we apply $\eta \geq \frac{1}{\mu n K}$ at the last step. Therefore, we can obtain

$$1 - (pq)^K \geq 1 - \left(e^{-\frac{1}{K}}\right)^K = 1 - e^{-1}.$$

The last inequality requires more careful analysis. We further refine the bounds for $pq$. Using $a = L - 4\mu < L$, it follows that

$$1 - \eta(L - a) - \eta^2 aL \geq 1 - 4\eta\mu - \eta^2 L^2 \geq 1 - \frac{4}{n\kappa} - \frac{1}{n^2} \geq 0,$$

where the second step is due to $\eta \leq \frac{1}{nL}$ and last step holds by the condition $\kappa \geq n \geq 3$. Hence,

$$pq = (1 - \eta L)^{\frac{n}{4}}(1 + \eta a)^{\frac{n}{4}} = (1 - \eta(L - a) - \eta^2 aL)^{\frac{n}{4}} \geq (1 - 4\eta\mu - \eta^2 L^2)^{\frac{n}{4}}. \tag{30}$$

We again divide the range of step size into two regimes: $\left[\frac{1}{\mu n K}, \frac{\mu}{L^2}\right)$ and $\left[\frac{\mu}{L^2}, \frac{1}{nL}\right)$.

**Regime 1.** $\eta \in \left[\frac{1}{\mu n K}, \frac{\mu}{L^2}\right)$.

In this regime, we have $\eta^2 L^2 \leq \eta\mu$. Hence, equation (30) becomes

$$pq \geq (1 - 4\eta\mu - \eta^2 L^2)^{\frac{n}{4}} \geq (1 - 5\eta\mu)^{\frac{n}{4}} \geq 1 - \frac{5}{4}\eta n\mu,$$

since $5\eta\mu \leq \frac{5\mu^2}{L^2} < 1$ (assuming $\kappa \geq n \geq 3$). Therefore, we obtain the following inequality:

$$\frac{1}{1 - pq} \geq \frac{4}{5\eta n\mu}.$$

**Regime 2.** $\eta \in \left[\frac{\mu}{L^2}, \frac{1}{nL}\right)$.

In this regime, we have $\eta^2 L^2 \geq \eta\mu$. Hence, equation (30) becomes

$$pq \geq (1 - 4\eta\mu - \eta^2 L^2)^{\frac{n}{4}} \geq (1 - 5\eta^2 L^2)^{\frac{n}{4}} \geq 1 - \frac{5}{4}\eta^2 nL^2,$$

since $5\eta^2 L^2 < \frac{5}{n^2} < 1$ (assuming $n \geq 3$). Therefore, we obtain the following inequality:

$$\frac{1}{1 - pq} \geq \frac{4}{5\eta^2 nL^2}.$$

This concludes the proof of the lemma. $\qquad\square$

# E. Proofs for Large Epoch Upper Bounds

In this section, we provide detailed proof for Theorem 4.4.

## E.1. Proof of Theorem 4.4

**Theorem 4.4.** *Let $n \geq 1$, $K \gtrsim (1 + P)\kappa$, and $\boldsymbol{x}_0$ be the initialization point. Suppose $F$ is a function satisfying Assumptions 2.3 and 2.4. Then, for any choice of permutation $\sigma_k$ in each epoch, the final iterate $\boldsymbol{x}_n^K$ obtained by Algorithm 1 with a step size $\eta = \frac{2}{\mu n K} \max\left\{\log\left(\frac{(F(\boldsymbol{x}_0) - F(\boldsymbol{x}^*))\mu^3 K^2}{L^2 G^2}\right), 1\right\}$ satisfies*

$$F(\boldsymbol{x}_n^K) - F(\boldsymbol{x}^*) \lesssim \frac{L^2 G^2}{\mu^3 K^2}.$$

*Proof.* We begin by noting the specific epoch condition used to prove the statement:

$$K \geq 8\kappa \max\{1, P\} \max\left\{\log\left(\frac{(F(\boldsymbol{x}_0) - F(\boldsymbol{x}^*))\mu^3 K^2}{L^2 G^2}\right), 1\right\}.$$

Given this epoch condition and the choice of step size $\eta$ specified in the theorem statement, we have $\eta n L \leq \frac{1}{4} \min\left\{1, \frac{1}{P}\right\}$, which will be repeatedly utilized throughout the proof.

Consider the following epoch-wise recursive inequality for the objective function:

$$F\left(\boldsymbol{x}_0^{k+1}\right) \leq F\left(\boldsymbol{x}_0^k\right) + \left\langle\nabla F\left(\boldsymbol{x}_0^k\right), \boldsymbol{x}_0^{k+1} - \boldsymbol{x}_0^k\right\rangle + \frac{L}{2}\left\|\boldsymbol{x}_0^{k+1} - \boldsymbol{x}_0^k\right\|^2$$

$$= F\left(\boldsymbol{x}_0^k\right) - \eta n\left\langle\nabla F\left(\boldsymbol{x}_0^k\right), \frac{1}{n}\sum_{i=1}^n \nabla f_{\sigma_k(i)}\left(\boldsymbol{x}_{i-1}^k\right)\right\rangle + \frac{\eta^2 n^2 L}{2}\left\|\frac{1}{n}\sum_{i=1}^n \nabla f_{\sigma_k(i)}\left(\boldsymbol{x}_{i-1}^k\right)\right\|^2$$

$$= F\left(\boldsymbol{x}_0^k\right) - \frac{\eta n}{2}\left\|\nabla F\left(\boldsymbol{x}_0^k\right)\right\|^2 - \frac{\eta n}{2}\left\|\frac{1}{n}\sum_{i=1}^n \nabla f_{\sigma_k(i)}\left(\boldsymbol{x}_{i-1}^k\right)\right\|^2$$

$$+ \frac{\eta n}{2}\left\|\nabla F\left(\boldsymbol{x}_0^k\right) - \frac{1}{n}\sum_{i=1}^n \nabla f_{\sigma_k(i)}\left(\boldsymbol{x}_{i-1}^k\right)\right\|^2 + \frac{\eta^2 n^2 L}{2}\left\|\frac{1}{n}\sum_{i=1}^n \nabla f_{\sigma_k(i)}\left(\boldsymbol{x}_{i-1}^k\right)\right\|^2$$

$$\overset{(a)}{\leq} F\left(\boldsymbol{x}_0^k\right) - \frac{\eta n}{2}\left\|\nabla F\left(\boldsymbol{x}_0^k\right)\right\|^2 + \frac{\eta n}{2}\left\|\nabla F\left(\boldsymbol{x}_0^k\right) - \frac{1}{n}\sum_{i=1}^n \nabla f_{\sigma_k(i)}\left(\boldsymbol{x}_{i-1}^k\right)\right\|^2$$

$$\overset{(b)}{\leq} F\left(\boldsymbol{x}_0^k\right) - \frac{\eta n}{2}\left\|\nabla F\left(\boldsymbol{x}_0^k\right)\right\|^2 + \frac{\eta L^2}{2}\sum_{i=1}^n\left\|\boldsymbol{x}_0^k - \boldsymbol{x}_{i-1}^k\right\|^2, \tag{31}$$

where (a) holds due to $\eta n L \leq \frac{1}{4} < 1$ and (b) follows from the inequality:

$$\left\|\nabla F\left(\boldsymbol{x}_0^k\right) - \frac{1}{n}\sum_{i=1}^n \nabla f_{\sigma_k(i)}\left(\boldsymbol{x}_{i-1}^k\right)\right\|^2 = \left\|\frac{1}{n}\sum_{i=1}^n\left(\nabla f_{\sigma_k(i)}\left(\boldsymbol{x}_0^k\right) - \nabla f_{\sigma_k(i)}\left(\boldsymbol{x}_{i-1}^k\right)\right)\right\|^2$$

$$\leq \frac{1}{n}\sum_{i=1}^n\left\|\nabla f_{\sigma_k(i)}\left(\boldsymbol{x}_0^k\right) - \nabla f_{\sigma_k(i)}\left(\boldsymbol{x}_{i-1}^k\right)\right\|^2$$

$$\leq \frac{L^2}{n}\sum_{i=1}^n\left\|\boldsymbol{x}_0^k - \boldsymbol{x}_{i-1}^k\right\|^2.$$

Next, we need to derive an upper bound for $\left\|\boldsymbol{x}_0^k - \boldsymbol{x}_{i-1}^k\right\|^2$. For $t \in [n]$, we have

$$\left\|\boldsymbol{x}_0^k - \boldsymbol{x}_t^k\right\|^2 = \eta^2\left\|\sum_{i=1}^t \nabla f_{\sigma_k(i)}\left(\boldsymbol{x}_{i-1}^k\right)\right\|^2$$

$$\leq 3\eta^2 \left\| \sum_{i=1}^t \left( \nabla f_{\sigma_k(i)} \left( \boldsymbol{x}_{i-1}^k \right) - \nabla f_{\sigma_k(i)} \left( \boldsymbol{x}_0^k \right) \right) \right\|^2 + 3\eta^2 \left\| \sum_{i=1}^t \left( \nabla f_{\sigma_k(i)} \left( \boldsymbol{x}_0^k \right) - \nabla F \left( \boldsymbol{x}_0^k \right) \right) \right\|^2$$

$$+ 3\eta^2 \left\| \sum_{i=1}^t \nabla F \left( \boldsymbol{x}_0^k \right) \right\|^2$$

$$\stackrel{(a)}{\leq} 3\eta^2 t \sum_{i=1}^t L^2 \left\| \boldsymbol{x}_0^k - \boldsymbol{x}_{i-1}^k \right\|^2 + 6\eta^2 t^2 \left( G^2 + P^2 \left\| \nabla F(\boldsymbol{x}_0^k) \right\|^2 \right) + 3\eta^2 t^2 \left\| \nabla F \left( \boldsymbol{x}_0^k \right) \right\|^2$$

$$= 3\eta^2 t L^2 \sum_{i=1}^t \left\| \boldsymbol{x}_0^k - \boldsymbol{x}_{i-1}^k \right\|^2 + 6\eta^2 t^2 G^2 + 3\eta^2 t^2 (1 + 2P^2) \left\| \nabla F \left( \boldsymbol{x}_0^k \right) \right\|^2. \tag{32}$$

Here, (a) is derived by applying Assumption 2.4 through the following sequence of inequalities:

$$\left\| \sum_{i=1}^t \left( \nabla f_{\sigma_k(i)} \left( \boldsymbol{x} \right) - \nabla F \left( \boldsymbol{x} \right) \right) \right\|^2 \leq t \sum_{i=1}^t \left\| \nabla f_{\sigma_k(i)} \left( \boldsymbol{x} \right) - \nabla F \left( \boldsymbol{x} \right) \right\|^2$$

$$\leq t \sum_{i=1}^t \left( G + P \left\| \nabla F(\boldsymbol{x}) \right\| \right)^2$$

$$\leq t \sum_{i=1}^t \left( 2G^2 + 2P^2 \left\| \nabla F(\boldsymbol{x}) \right\|^2 \right)$$

$$\leq 2t^2 \left( G^2 + P^2 \left\| \nabla F(\boldsymbol{x}) \right\|^2 \right).$$

Summing equation (32) over $t = 1, \ldots, n-1$, we have

$$\sum_{i=1}^n \left\| \boldsymbol{x}_0^k - \boldsymbol{x}_{i-1}^k \right\|^2 \leq 3\eta^2 \frac{(n-1)n}{2} L^2 \sum_{i=1}^n \left\| \boldsymbol{x}_0^k - \boldsymbol{x}_{i-1}^k \right\|^2 + 6\eta^2 \frac{(n-1)n(2n-1)}{6} G^2$$

$$+ 3\eta^2 \frac{(n-1)n(2n-1)}{6} (1 + 2P^2) \left\| \nabla F \left( \boldsymbol{x}_0^k \right) \right\|^2$$

$$\leq 3\eta^2 n^2 L^2 \sum_{i=1}^n \left\| \boldsymbol{x}_0^k - \boldsymbol{x}_{i-1}^k \right\|^2 + 2\eta^2 n^3 G^2 + \eta^2 n^3 (1 + 2P^2) \left\| \nabla F \left( \boldsymbol{x}_0^k \right) \right\|^2.$$

Given $\eta n L \leq \frac{1}{4}$, it follows that $3\eta^2 n^2 L^2 \leq \frac{1}{2}$ and the above inequality simplifies to

$$\sum_{i=1}^n \left\| \boldsymbol{x}_0^k - \boldsymbol{x}_{i-1}^k \right\|^2 \leq 4\eta^2 n^3 G^2 + 2\eta^2 n^3 (1 + 2P^2) \left\| \nabla F \left( \boldsymbol{x}_0^k \right) \right\|^2. \tag{33}$$

Substituting equation (33) to equation (31) results in

$$F \left( \boldsymbol{x}_0^{k+1} \right) \leq F \left( \boldsymbol{x}_0^k \right) - \frac{\eta n}{2} \left\| \nabla F \left( \boldsymbol{x}_0^k \right) \right\|^2 + \frac{\eta L^2}{2} \sum_{i=1}^n \left\| \boldsymbol{x}_0^k - \boldsymbol{x}_{i-1}^k \right\|^2$$

$$\leq F(\boldsymbol{x}_0^k) - \frac{\eta n}{2} \left\| \nabla F(\boldsymbol{x}_0^k) \right\|^2 + \frac{\eta L^2}{2} \left( 4\eta^2 n^3 G^2 + 2\eta^2 n^3 (1 + 2P^2) \left\| \nabla F(\boldsymbol{x}_0^k) \right\|^2 \right)$$

$$\leq F(\boldsymbol{x}_0^k) - \frac{\eta n}{2} \left( 1 - 2\eta^2 n^2 L^2 \left( 1 + 2P^2 \right) \right) \left\| \nabla F(\boldsymbol{x}_0^k) \right\|^2 + 2\eta^3 n^3 L^2 G^2$$

$$\stackrel{(a)}{\leq} F(\boldsymbol{x}_0^k) - \frac{\eta n}{4} \left\| \nabla F(\boldsymbol{x}_0^k) \right\|^2 + 2\eta^3 n^3 L^2 G^2$$

$$\stackrel{(b)}{\leq} F(\boldsymbol{x}_0^k) - \frac{\eta n \mu}{2} \left( F(\boldsymbol{x}_0^k) - F(\boldsymbol{x}^*) \right) + 2\eta^3 n^3 L^2 G^2,$$

where at $(a)$, we use $\eta n L \leq \frac{1}{4}$ and $\eta n L \leq \frac{1}{4P}$, ensuring $\eta^2 n^2 L^2 \left(1 + 2P^2\right) \leq \frac{1}{16} + \frac{1}{8} \leq \frac{1}{4}$, and at $(b)$, we utilize the assumption that $F$ satisfies $\mu$-strongly convexity. We note that $(b)$ is the only step where $\mu$-strong convexity of $F$ is utilized, and it also holds under the weaker assumption that $F$ satisfies the Polyak-Łojasiewicz condition. Thus, Theorem 4.4 remains valid when $F$ satisfies the PŁ condition.

Rearranging this inequality leads to

$$F\left(\boldsymbol{x}_0^{k+1}\right) - F(\boldsymbol{x}^*) \leq \left(1 - \frac{\eta n \mu}{2}\right) \left(F\left(\boldsymbol{x}_0^k\right) - F^*\right) + 2\eta^3 n^3 L^2 G^2,$$

and we can obtain

$$F\left(\boldsymbol{x}_n^K\right) - F(\boldsymbol{x}^*) \leq \left(1 - \frac{\eta n \mu}{2}\right)^K \left(F\left(\boldsymbol{x}_0\right) - F(\boldsymbol{x}^*)\right) + 2\eta^3 n^3 L^2 G^2 \cdot \sum_{k=1}^{K} \left(1 - \frac{\eta n \mu}{2}\right)^{k-1}$$

$$\leq \left(1 - \frac{\eta n \mu}{2}\right)^K \left(F\left(\boldsymbol{x}_0\right) - F(\boldsymbol{x}^*)\right) + 2\eta^3 n^3 L^2 G^2 \cdot \frac{2}{\eta n \mu}$$

$$\leq e^{-\frac{\eta \mu n K}{2}} \left(F\left(\boldsymbol{x}_0\right) - F^*\right) + \frac{4\eta^2 n^2 L^2 G^2}{\mu}.$$

We now substitute $\eta = \frac{2}{\mu n K} \max\left\{\log\left(\frac{(F(\boldsymbol{x}_0) - F(\boldsymbol{x}^*))\mu^3 K^2}{L^2 G^2}\right), 1\right\}$. For the case when $F(\boldsymbol{x}_0) - F(\boldsymbol{x}^*)$ is sufficiently large, the above inequality becomes

$$F\left(\boldsymbol{x}_n^K\right) - F(\boldsymbol{x}^*) \leq \frac{L^2 G^2}{\mu^3 K^2} + \frac{16 L^2 G^2}{\mu^3 K^2} \cdot \log^2\left(\frac{(F(\boldsymbol{x}_0) - F(\boldsymbol{x}^*))\mu^3 K^2}{L^2 G^2}\right) \lesssim \frac{L^2 G^2}{\mu^3 K^2}.$$

For the case when $F(\boldsymbol{x}_0) - F(\boldsymbol{x}^*)$ is small so that 1 is chosen after the max operation, the above inequality then becomes

$$F\left(\boldsymbol{x}_n^K\right) - F(\boldsymbol{x}^*) \leq \frac{1}{e} \cdot e \cdot \frac{L^2 G^2}{\mu^3 K^2} + \frac{16 L^2 G^2}{\mu^3 K^2} \lesssim \frac{L^2 G^2}{\mu^3 K^2},$$

where we utilize $F(\boldsymbol{x}_0) - F(\boldsymbol{x}^*) \leq e \cdot \frac{L^2 G^2}{\mu^3 K^2}$. This ends the proof of Theorem 4.4. $\qquad\square$

## F. Lemmas

**Lemma F.1.** *Let $n$ be an even number. Define $F(x) = \frac{a}{2}x^2$ with component functions*

$$f_i(x) = \begin{cases} \frac{a}{2}x^2 + Gx & \text{if } i \leq n/2, \\ \frac{a}{2}x^2 - Gx & \text{otherwise.} \end{cases}$$

*Then, the final iterate $x_n^K$ obtained by running Algorithm 2 for $K$ epochs with a step size $\eta$ starting from the initialization point $x_0$, satisfies:*

$$x_n^K = (1 - \eta a)^{nK} x_0 + \frac{G}{a} \cdot \frac{1 - (1 - \eta a)^{\frac{n}{2}}}{1 + (1 - \eta a)^{\frac{n}{2}}} \left(1 - (1 - \eta a)^{nK}\right).$$

*Proof.* For $i \leq \frac{n}{2}$, the update rule is given as:

$$x_i^k = x_{i-1}^k - \eta(ax_{i-1}^k + G) = (1 - \eta a)x_{i-1}^k - \eta G.$$

For $i \geq \frac{n}{2} + 1$, the update rule is given as:

$$x_i^k = x_{i-1}^k - \eta\left(ax_{i-1}^k - G\right) = (1 - \eta a)x_{i-1}^k + \eta G.$$

By sequentially applying the component functions, we derive the following epoch-wise recursion equation:

$$x_0^{k+1} = (1 - \eta a)^n x_0^k - \eta G \sum_{i=1}^{\frac{n}{2}} (1 - \eta a)^{n-i} + \eta G \sum_{i=\frac{n}{2}+1}^{n} (1 - \eta a)^{n-i}$$

$$= (1 - \eta a)^n x_0^k + \frac{G}{a}\left(1 - (1 - \eta a)^{\frac{n}{2}}\right)^2, \tag{34}$$

where the last equality follows from the following observation:

$$-\eta G \sum_{i=1}^{\frac{n}{2}} (1 - \eta a)^{n-i} + \eta G \sum_{i=\frac{n}{2}+1}^{n} (1 - \eta a)^{n-i} = \eta G \left(1 - (1 - \eta a)^{\frac{n}{2}}\right) \sum_{i=\frac{n}{2}+1}^{n} (1 - \eta a)^{n-i}$$

$$= \eta G \left(1 - (1 - \eta a)^{\frac{n}{2}}\right) \frac{1 - (1 - \eta a)^{\frac{n}{2}}}{\eta a}$$

$$= \frac{G}{a}\left(1 - (1 - \eta a)^{\frac{n}{2}}\right)^2.$$

By unrolling equation (34) over $k \in [K]$, we obtain the equation for the final iterate $x_n^K$:

$$x_n^K = (1 - \eta a)^{nK} x_0 + \frac{G}{a} \cdot \frac{1 - (1 - \eta a)^{nK}}{1 - (1 - \eta a)^n} \left(1 - (1 - \eta a)^{\frac{n}{2}}\right)^2$$

$$= (1 - \eta a)^{nK} x_0 + \frac{G}{a} \cdot \frac{1 - (1 - \eta a)^{\frac{n}{2}}}{1 + (1 - \eta a)^{\frac{n}{2}}} \left(1 - (1 - \eta a)^{nK}\right).$$

$\square$

**Lemma F.2.** *Let $n$ be an odd number. Define $F(x) = \frac{a}{2}x^2$ with component functions*

$$f_i(x) = \begin{cases} \frac{a}{2}x^2 & \text{if } i = 1, \\ \frac{a}{2}x^2 + Gx & \text{if } 2 \leq i \leq (n+1)/2, \\ \frac{a}{2}x^2 - Gx & \text{if } (n+3)/2 \leq i \leq n. \end{cases}$$

*Then, the final iterate $x_n^K$ obtained by running Algorithm 2 for $K$ epochs with a step size $\eta$ starting from the initialization point $x_0$, satisfies:*

$$x_n^K = (1 - \eta a)^{nK} x_0 + \frac{G}{a} \cdot \frac{1 - (1 - \eta a)^{nK}}{1 - (1 - \eta a)^n} \left(1 - (1 - \eta a)^{\frac{n-1}{2}}\right)^2.$$

*Proof.* Compared to Lemma F.1, we have an additional component function $f_1(x) = \frac{a}{2}x^2$ at the beginning of each epoch. For this function, the update for $x_1^k$ is given as:

$$x_1^k = x_0^k - \eta a x_0^k = (1 - \eta a)x_0^k.$$

Thus, the epoch-wise equation in equation (34) of Lemma F.1 is modified as follows:

$$x_0^{k+1} = (1-\eta a)^{n-1}x_1^k + \frac{G}{a}\left(1 - (1-\eta a)^{\frac{n-1}{2}}\right)^2 = (1-\eta a)^n x_0^k + \frac{G}{a}\left(1 - (1-\eta a)^{\frac{n-1}{2}}\right)^2.$$

By unrolling the above inequality over $k \in [K]$, we obtain the equation for the final iterate $x_n^k$:

$$x_n^K = (1-\eta a)^{nK}x_0 + \frac{G}{a} \cdot \frac{1 - (1-\eta a)^{nK}}{1 - (1-\eta a)^n}\left(1 - (1-\eta a)^{\frac{n-1}{2}}\right)^2.$$

$\square$

**Lemma F.3.** *Let $n$ be an even number. Define $F(x) = \frac{a}{8}x^2$ with component functions*

$$f_i(x) = \begin{cases} \frac{a}{2}x^2 + Gx & \text{if } i \leq n/2, \\ -\frac{a}{4}x^2 - Gx & \text{otherwise.} \end{cases}$$

*Consider applying Algorithm 2 for a single epoch, starting from $x_0^k$. The updated iterate $x_0^{k+1}$ satisfies the following equation:*

$$x_0^{k+1} = \left(1 + \frac{\eta a}{2}\right)^{\frac{n}{2}}(1-\eta a)^{\frac{n}{2}}x_0^k + \frac{G}{a}\left(\left(1 + \frac{\eta a}{2}\right)^{\frac{n}{2}}\left(1 + (1-\eta a)^{\frac{n}{2}}\right) - 2\right).$$

*Proof.* For $i \leq \frac{n}{2}$, the update rule is given as:

$$x_i^k = x_{i-1}^k - \eta(ax_{i-1}^k + G) = (1-\eta a)x_{i-1}^k - \eta G.$$

By sequentially applying the first half of the component functions, we obtain

$$x_{\frac{n}{2}}^k = (1-\eta a)^{\frac{n}{2}}x_0^k - \eta G \sum_{i=0}^{\frac{n}{2}-1}(1-\eta a)^i = (1-\eta a)^{\frac{n}{2}}x_0^k - \eta G \cdot \frac{1 - (1-\eta a)^{\frac{n}{2}}}{\eta a}$$

$$= (1-\eta a)^{\frac{n}{2}}x_0^k - \frac{G}{a}\left(1 - (1-\eta a)^{\frac{n}{2}}\right). \tag{35}$$

For $i \geq \frac{n}{2} + 1$, the update rule is given as:

$$x_i^k = x_{i-1}^k - \eta\left(-\frac{a}{2}x_{i-1}^k - G\right) = \left(1 + \frac{\eta a}{2}\right)x_{i-1}^k + \eta G.$$

Substituting the result from equation (35) into the update rule for the second half of the component functions, we obtain $x_0^{k+1}$ (equivalently, $x_n^k$) as follows:

$$x_0^{k+1} = \left(1 + \frac{\eta a}{2}\right)^{\frac{n}{2}}x_{\frac{n}{2}}^k + \eta G \sum_{i=0}^{\frac{n}{2}-1}\left(1 + \frac{\eta a}{2}\right)^i = \left(1 + \frac{\eta a}{2}\right)^{\frac{n}{2}}x_{\frac{n}{2}}^k + \eta G \cdot \frac{2\left(\left(1 + \frac{\eta a}{2}\right)^{\frac{n}{2}} - 1\right)}{\eta a}$$

$$= \left(1 + \frac{\eta a}{2}\right)^{\frac{n}{2}}x_{\frac{n}{2}}^k + \frac{2G}{a}\left(\left(1 + \frac{\eta a}{2}\right)^{\frac{n}{2}} - 1\right)$$

$$= \left(1 + \frac{\eta a}{2}\right)^{\frac{n}{2}}\left((1-\eta a)^{\frac{n}{2}}x_0^k - \frac{G}{a}\left(1 - (1-\eta a)^{\frac{n}{2}}\right)\right) + \frac{2G}{a}\left(\left(1 + \frac{\eta a}{2}\right)^{\frac{n}{2}} - 1\right)$$

$$= \left(1 + \frac{\eta a}{2}\right)^{\frac{n}{2}}(1-\eta a)^{\frac{n}{2}}x_0^k + \frac{G}{a}\left(\left(1 + \frac{\eta a}{2}\right)^{\frac{n}{2}}\left(1 + (1-\eta a)^{\frac{n}{2}}\right) - 2\right).$$

$\square$

# G. Experiments

In this section, we validate the lower bound convergence rates for the functions used in the lower bound construction of Theorems 3.3 and 3.5. We compare the performance of four permutation-based SGD methods: IGD, RR, Herding at Optimum, and with-replacement SGD. Here, Herding at Optimum refers to the instance of Algorithm 1 using the permutation suggested from Theorem 3.7 satisfying equation (22). As mentioned in Section 3.2, this permutation is generally unknown without prior knowledge of $x^*$. However, for the specific functions used in the lower bound construction, we can explicitly determine a permutation $\sigma$ that satisfies equation (22). Thus, the plot for Herding represents the convergence rate achieved by a well-chosen permutation in permutation-based SGD.

Additionally, we conduct experiments using the MNIST (appendix G.3) and CIFAR-10 (appendix G.4) datasets. For the real-world dataset, we compare the performance of IGD, RR, and with-replacement SGD. Training loss and the test accuracy of both MNIST and CIFAR-10 reveal the significant slowdown of IGDat the early stages of the training. For details of the experiments, we refer readers to the corresponding subsections.

## G.1. Results for the Function in Theorem 3.3

Recall that the proof of Theorem 3.3 uses 4-dimensional functions, formulated through the "dimension aggregation" step. For a clear observation, we conduct experiments using the construction for the "Moderate" step size regime, and remove the first and the last dimension.

We use the parameters $\mu = 1.0 \times 10^0$, $L = 1.0 \times 10^4$, $G = 1.0 \times 10^0$, $n = 1.0 \times 10^3$, and choose the step size as $\eta = \frac{1}{\mu n K}$ which corresponds to the moderate step size regime. First, we examine the trajectory of IGD when initialized at $x^*$ in Figure 2. Recall that our construction is carefully designed so that the trajectory forms a regular $n$-polygon when starting from $(u_0(\eta), v_0(\eta))$ (see Appendix B.2 for definitions). As illustrated in Figure 2, even when the iterate starts at $x^*$, it gradually drifts outward and rotates along a circular path.

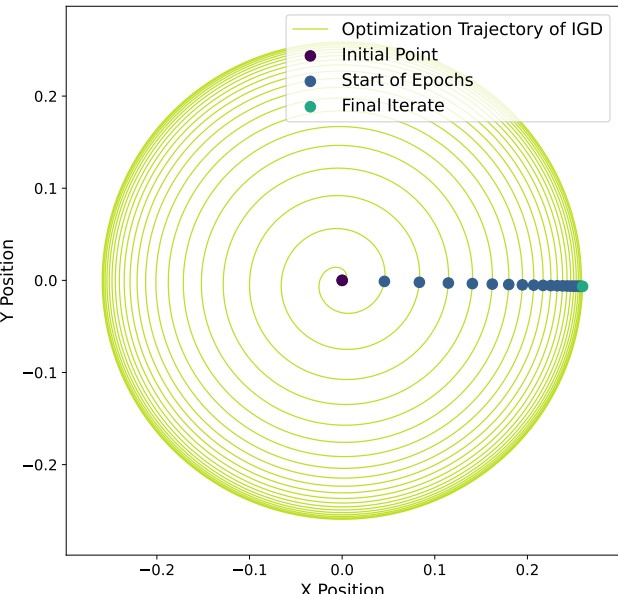

*Figure 2.* Trajectory of IGD with the function for Theorem 3.3, starting from the $x^*$ (the origin, purple dot), when $K = 20$. Blue dots starting point of each epoch, $x_0^k$, while the cyan dot indicates the final iterate $x_0^K$.

Figure 3 reports the function optimality gap for different permutation-based SGD methods, when initialized at $(u_0(\frac{1}{\mu n K}), v_0(\frac{1}{\mu n K})))$. Results for RR and with-replacement SGD, which involves randomness, are reported after averaging over 20 trials for each number of epochs $k$. The shaded region represents the first and the third quartiles across the 20 trials.

One might wonder why the trend of IGD does not match the rate derived in Theorem 3.3, given by $\frac{LG^2}{\mu^2} \min\left\{1, \frac{\kappa^2}{K^4}\right\}$. We believe this occurs because the theoretical rate serves as a lower bound on the true convergence rate, and the empirical performance of IGD in this experiment can be influenced by additional factors not captured in the theoretical bound.

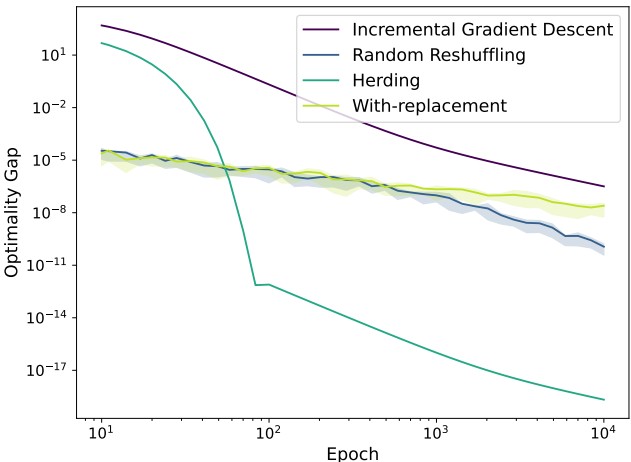

*Figure 3.* Experiments on Theorem 3.3 for IGD, RR, Herding at Optimum, and with-replacement SGD. Both axes are log-scaled.

### G.2. Results for the Function in Theorem 3.5

Recall that the proof of Theorem 3.5 uses 4-dimensional functions, formulated through the "dimension aggregation" step. For a clear observation, we conduct experiments using the construction for the "Moderate & Large" step size regime, and remove the first dimension.

We use the parameters $\mu = 1.0 \times 10^0$, $L = 1.0 \times 10^4$, $G = 1.0 \times 10^0$, $n = 1.0 \times 10^2$, and choose the step size as $\eta = \frac{1}{\mu n K}$.

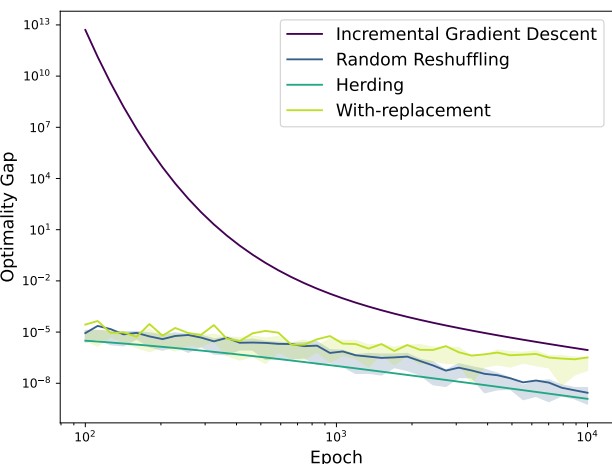

*Figure 4.* Experiments on Theorem 3.5 for IGD, RR, Herding at Optimum, and with-replacement SGD. Both axes are log-scaled.

Figure 4 reports the function optimality gap for different permutation-based SGD methods, when initialized at $(0, 0)$. Results for RR and with-replacement SGD, which involves randomness, are reported after averaging over 20 trials for each number of epochs, $k$. The shaded region represents the first and the third quartiles over the 20 trials. As suggested by Theorem 3.5, the function optimality gap increases sharply as $K$ decreases. In contrast, RR remains robust even for small $K$.

### G.3. Experiments on MNIST Dataset

For the MNIST dataset, we consider the binary classification using only the data corresponding to the labels *0* and *1*. We consider the natural data ordering where all *0* images are followed by all *1* images. In this configuration, we have a total of $5{,}923 + 6{,}742 = 12{,}665$ training data. We use a step size $\eta = 0.01$ throughout every part of the training.

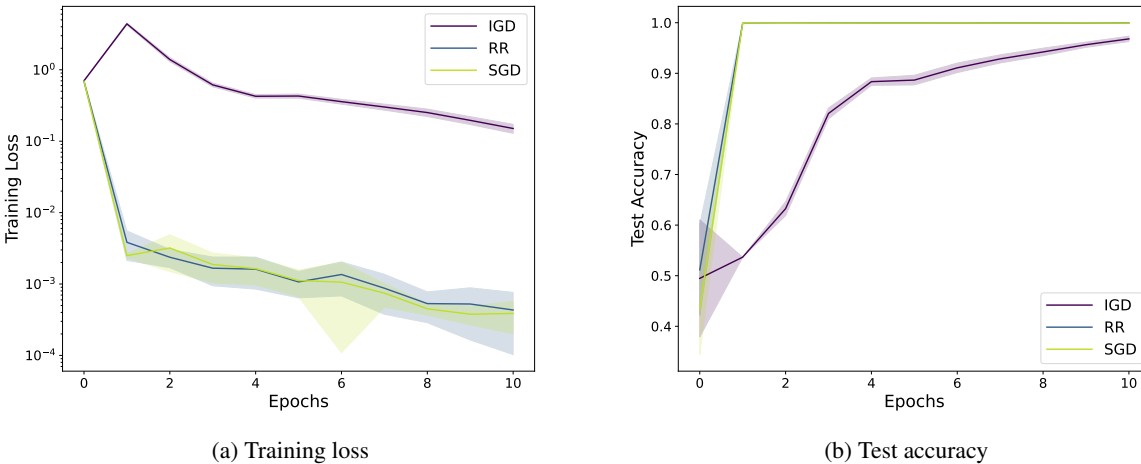

(a) Training loss                    (b) Test accuracy

*Figure 5.* Experiments on MNIST dataset for IGD, RR, and with-replacement SGD. $y$-axis for the training loss is log-scaled.

Figure 5 reports the training loss and the test accuracy for different permutation-based SGD methods, with a random initialization. Results are reported after averaging over 10 trials for each number of epochs, $k$. The shaded region represents the 95% confidence interval over 10 trials. Unlike the experiments on the functions corresponding to the theorems using a fixed initialization, the randomness in the initialization for this experiment introduces a confidence interval even to IGD. Both the loss and the accuracy show no significant difference between RR and with-replacement SGD, while IGD shows a significantly slower convergence compared to the other two methods.

### G.4. Experiments on CIFAR-10 Dataset

For the CIFAR-10 dataset, we also consider the binary classification using only the data corresponding to the labels *airplane* and *automobile*. We consider the natural data ordering where all *airplane* images are followed by all *automobile* images. In this configuration, we have a total of $5{,}000 + 5{,}000 = 10{,}000$ training data. We use a step size $\eta = 0.001$ throughout every part of the training.

One slight difference from the experiment on the MNIST dataset is that we use a mini-batch of size 16 for the training. This is due to the instability of IGD training. To ensure convergence of IGD with a reasonable step size—such that the loss function decreases even with a small number of training epochs—we employ its mini-batch variant. For a fair comparison, we also adopt the corresponding mini-batch versions of RR and with-replacement SGD.

Figure 6 reports the training loss and the test accuracy for different permutation-based SGD methods, with a random initialization. Results are reported after averaging over 10 trials for each number of epochs, $k$. The shaded region represents the 95% confidence interval over 10 trials. Both the loss and the accuracy show no significant difference between RR and with-replacement SGD, while IGD shows a significantly slower convergence compared to the other two methods.

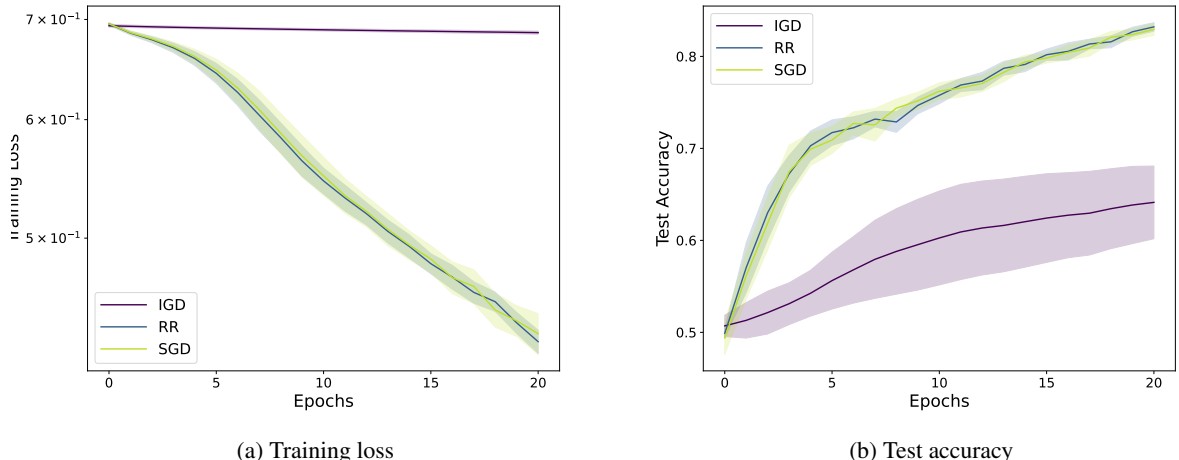

(a) Training loss

(b) Test accuracy

*Figure 6.* Experiments on CIFAR-10 dataset for IGD, RR, and with-replacement SGD. $y$-axis for the training loss is log-scaled.

