# OpenReview forum: "Incremental Gradient Descent with Small Epoch Counts is Surprisingly Slow on Ill-Conditioned Problems"
_ICML.cc/2025/Conference — ICML 2025 poster_

### Official Review · Reviewer_1iDN · 2025-03-13

**Overall Recommendation:** 4

**Summary:**

This work investigates the convergence of shuffling gradient methods, especially focusing on the small epoch regime. The authors establish several new upper/lower bounds that are matched to each other, providing new insights into the finite-sum optimization problem.

## update after rebuttal

I keep my positive score as this is a good paper in my opinion.

**Claims And Evidence:**

All claims are proved.

**Essential References Not Discussed:**

N/A.

**Experimental Designs Or Analyses:**

The experiments are enough to demonstrate the correctness of theorems.

**Methods And Evaluation Criteria:**

N/A.

**Other Comments Or Suggestions:**

I found one inaccurate statement.

Line 216 (right column), [1] also refines the exponential term. Concretely, the exponential term in Theorem 4.6 of [1] is in the order of $LD^2\exp(-K/\kappa)K^{-1}$, which improves a factor of $K$. Although this change doesn't imply [1] is optimal in the small epoch regime, it is better to be accurate.

**Reference**

[1] Liu, Zijian, and Zhengyuan Zhou. "On the last-iterate convergence of shuffling gradient methods." arXiv preprint arXiv:2403.07723 (2024).

**Other Strengths And Weaknesses:**

The paper is written clearly and is highly polished. There is no specific weakness I can find.

**Questions For Authors:**

I only have one question.

Theorem 5 in [1] only requires $\frac{1}{n}\sum_{i=1}^n\left\Vert\nabla f_i(\boldsymbol{x}^*)\right\Vert \leq G_*$ in contrast to the stronger assumption $\left\Vert\nabla f_i(\boldsymbol{x}^*)\right\Vert \leq G_*,\forall i\in\left[n\right]$ assumed in Proposition 3.4. Can this gap be fixed?

**Reference**

[1] Mishchenko, Konstantin, Ahmed Khaled, and Peter Richtárik. "Random reshuffling: Simple analysis with vast improvements." Advances in Neural Information Processing Systems 33 (2020): 17309-17320.

**Relation To Broader Scientific Literature:**

N/A.

**Theoretical Claims:**

As far as I can see, theorems are correct.

---

> ### Author Rebuttal · Authors · 2025-04-01
>
> We appreciate the reviewer for the positive feedback. Below, we address the reviewer’s comments.
>
> 1. **The exponential term in Theorem 4.6 of [Liu et al., 2024] is in the order of $LD^2 \exp(-K/\kappa)K^{-1}$, which improves a factor of $K$.**
>     - Thank you for pointing out this issue. We have revised our original statement to accurately reflect that Theorem 4.6 of [Liu et al., 2024] also refines the exponential term. Specifically, we now write: “Also, a more recent result by [Liu & Zhou, 2024] (Theorem 4.6) refines the polynomial term and also improves the exponential term to $\exp(-K/\kappa) \frac{LD^2}{K}$. However, since the term inside the exponential remains unchanged, this still fails to reveal a tight bound when $K$ is small.”
> 2. **Can the assumption in Proposition 3.4 be relaxed to $\frac{1}{n} \sum_{i=1}^n \|\| \nabla f_i (x^*) \|\| \le G_*$?**
>     - Great question. Yes, the reviewer is absolutely right that we can obtain exactly the same convergence result as in Proposition 3.4 under the weaker assumption $\frac{1}{n} \sum_{i=1}^n \|\| \nabla f_i (x^*) \|\| \le G_*$. The reason why we state the stronger assumption is to maintain consistency with the assumptions used throughout the rest of the paper.
>
> If you have any remaining questions or comments, we will happily answer them.

---

### Official Review · Reviewer_RSZD · 2025-03-14

**Overall Recommendation:** 3

**Summary:**

This paper studies the Incremental Gradient Descent, a permutation-based SGD method. The authors derive a lower bound on the algorithm's progress when the number of epochs is small. This paper provides results for various classes of problems, when 1. all he component functions are strongly convex, 2. all components are strongly convex with the same Hessian, or 3. some functions are non-convex. This is an essential result as modern machine learning applications are ill-conditioned; in most cases, we are in a small epoch regime.

**Claims And Evidence:**

The algorithm's proofs use a step size that depends on the optimal point $x_*$. This is not practical as $x_*$ is unknown.

**Essential References Not Discussed:**

I am happy with the papers cited in the paper to understand this work.

**Experimental Designs Or Analyses:**

The papers lack experiments. It fails to show in practice that Incremental Gradient Descent does not perform well in the small epoch regime.

**Methods And Evaluation Criteria:**

The paper doesn't provide extensive experiments, so it is hard to evaluate whether the IGD algorithm is worse than SGD with uniform sampling for the small epoch regime.

**Other Comments Or Suggestions:**

1. Authors should conduct more experiments to evaluate the performance of these algorithms in the small epoch regime.

**Other Strengths And Weaknesses:**

Strength:
1. The paper provides new lower bounds for Incremental Gradient Descent in small epoch regime.
2. This paper provides insight into the computationally constrained setting.

Weakness:
1. I believe similar lower bounds can be derived for other algorithms like SGD (with uniform sampling) in the small epoch regime. This is because you put a upper bound on the number of iterations (with condition number).
2. The paper requires experiments to evaluate if IGD performs worse than SGD with uniform sampling in the small epoch regime.
3. The step sizes in Theorem 3.2, Proposition 3.4 require the knowledge of $x_*$ which is not known in practice.

**Questions For Authors:**

1. The IGD algorithm is deterministic. Then why do you need assumptions 2.4, 2.5 to prove convergence?

**Relation To Broader Scientific Literature:**

This is an essential result as modern machine learning applications are ill-conditioned; and in most cases, we are in a small epoch regime. Therefore, this result helps us to understand what is a good algorithm for training modern neural networks.

**Theoretical Claims:**

The authors claim "Our lower bounds reveal that for the small epoch regime, IGD can exhibit surprisingly slow convergence even when all component functions are strongly convex."

---

> ### Author Rebuttal · Authors · 2025-04-01
>
> We appreciate the reviewer for the constructive feedback. Below, we address the reviewer’s concerns.
>
> 1. **Similar lower bounds can be derived for other algorithms like SGD (with uniform sampling) in the small epoch regime.**
>     - As the reviewer pointed out, it is true that any algorithm (including IGD, SGD with uniform sampling, etc) with a finite number of iterations will naturally have some lower bound. However, our focus is more specific: given a fixed number of epochs $K$, we aim to characterize how fast an algorithm converges to the minimum with respect to $K$.
>     - It is well known that for SGD with uniform sampling, both the upper and lower bounds are of order $O(1/\mu T)$, where $T$ is the total iterations. Recall that $T = nK$ in our setting. This rate holds uniformly in both the large and small epoch regimes. In contrast, our results for IGD reveal a clear separation between the convergence rates in the large and small epoch regimes. In particular, we show that in the small epoch regime, the LB for IGD is strictly worse than the UB for SGD with uniform sampling. This directly implies that IGD converges more slowly than SGD with uniform sampling in this setting.
>     - If we have misunderstood the reviewer's question, we would appreciate it if you could kindly clarify.
> 2. **The algorithm's proofs use a step size that depends on the optimal point $x_*$.**
>     - The reason why we introduced $x^*$ into the step size is to reduce the dependence of the final convergence rate on the initial optimality gap $f(x_0)-f(x^*)$ from linear to a logarithmic scale. Even if the step size is chosen without $x^*$, the overall dependence on major parameters (e.g., $\mu, L, n, K$) remains unchanged.
>     - We also would like to note that using step sizes depending on $x^*$ is a common practice in the literature [1, 2, 3].
> 3. **The paper requires experiments to evaluate if IGD performs worse than SGD with uniform sampling in the small epoch regime.**
>     - During the rebuttal period, we conducted experiments on a binary classification task using two selected labels from the MNIST dataset. We trained a CNN using IGD, RR, and with-replacement SGD. In our setup, IGD uses all samples from one class, followed by the remaining samples from the other class. We provide the figures for the training loss and test accuracy at the following link: https://anonymous.4open.science/r/ICML2025-IGD-134A. In the figures, confidence intervals are calculated over 10 different runs; for IGD, the variability arises from different random initializations. As expected, we observe that IGD converges much slower than vanilla SGD. We plan to conduct more extensive experiments on broader datasets and model architectures, and will include these results in the next revision. We hope this addresses your concerns.
> 4. **IGD is deterministic. Why do you need assumptions 2.4, 2.5?**
>     - We note that our upper bound results hold for arbitrary permutation-based SGD, not solely for IGD. As this covers all possible permutation selection schemes, including random choices, we believe that it is natural to have some kind of gradient error bound like Assumptions 2.4 and 2.5.
>     - However, even when focusing solely on upper bounds for IGD, we still believe that Assumptions 2.4 and 2.5 are essential. While IGD is a deterministic algorithm, the convergence behavior is still significantly influenced by the structure of component gradients. For instance, in the case where all components are identical, i.e., $f_i=f$, IGD can exhibit an exponential convergence rate. On the other hand, when the gradients of component function differ substantially from each other, the iterates can fluctuate significantly, leading to slower convergence.
>     - As discussed in Section 2.4, a key technique in analyzing the convergence of permutation-based SGD methods is controlling the cumulative error within each epoch. We note that this cumulative error and the fluctuation of iterates are closely related. Therefore, even for IGD—the simplest and fully deterministic permutation-based algorithm—some form of assumption to control gradient error is still necessary to establish proper convergence guarantees.
>     - Finally, we note that similar assumptions on component gradients are commonly made in prior IGD literature [4, 5].
>
> If you have any remaining questions or comments, we will happily answer them.
>
> ---
>
> [1] Lu, Yucheng, et al. "Grab: Finding provably better data permutations than random reshuffling." NeurIPS 2022
>
> [2] Cha, Jaeyoung, et al. "Tighter lower bounds for shuffling SGD: Random permutations and beyond." ICML 2023
>
> [3] Liu, Zijian, and Zhengyuan Zhou. "On the last-iterate convergence of shuffling gradient methods." ICML 2024
>
> [4] Mishchenko, Konstantin, et al. "Random reshuffling: Simple analysis with vast improvements." NeurIPS 2020
>
> [5] Koloskova, Anastasia, et al. "On convergence of incremental gradient for non-convex smooth functions." ICML 2024

---

### Official Review · Reviewer_oi4v · 2025-03-14

**Overall Recommendation:** 4

**Summary:**

The paper is inspired by the common use of shuffling-based methods such as Random Reshuffling in practice, and the authors provide new theoretical results for specific permutations. In particular, they give new lower bounds on the Incremental Gradient (IG) method in the low-epoch regime, which wasn't considered much in the prior literature. The results concern the strongly convex case, but with multiple options on the individual components: either all of them are strongly convex, or just convex, or even potentially nonconvex. Somewhat surprisingly, this affects the provided lower bounds, despite that distinction having smaller importance in the large-epoch regime.

The results can be of interest due to the popularity of shuffling methods in practice and the fact that there are still open questions and gaps. The result concerning Herding is interesting since it shows that potentially better permutations can be obtained to make shuffling methods faster.

## update after rebuttal

I thanks the authors for the interesting discussion. I remain positive this is a good paper and I hope it gets accepted.

**Claims And Evidence:**

All theoretical claims are supported by rigorous proofs. One of the results is also verified numerically.

**Essential References Not Discussed:**

I don't think there are any crucial papers missed. The paper doesn't discuss some of the papers on random reshuffling in other settings (proximal, distributed, etc.) but I think it's completely reasonable as the main focus on closing the gaps in the most basic setting of stochastic optimization.

**Experimental Designs Or Analyses:**

The numerical results are pretty simple and their soundness is immediately verifiable.

**Methods And Evaluation Criteria:**

The used criteria are standard in the optimization community.

**Other Comments Or Suggestions:**

From the proofs, it appears to me that the lower bounds can be immediately extended to also include claims on the distance to the solutions. Am I missing something? If not, I encourage the authors to state those as well.

Minor:
In the equation in Definition 2.1, $x$ and $y$ after the $\forall$ symbol should be bold as well.
"Cha et al.(2023) establishes" -> "Cha et al.(2023) establish".
It would be nice to capitalize names in the citations, for instance "SGD" instead of "Sgd".

**Other Strengths And Weaknesses:**

The main limitation of the work, in my opinion, is that it considers the least interesting of the shuffling methods. I still believe it provides sufficient new intuition to publish the paper, but I just wanted to point out that having new results on random reshuffling would have been more interesting.

**Questions For Authors:**

Please see my suggestion/question on the extension of lower bounds to distances.

**Relation To Broader Scientific Literature:**

The work follows up on the series of papers published in the last few years and adds a solid contribution closing some of the theoretical gaps.

**Theoretical Claims:**

I checked the correctness of Theorems 3.1 and Theorem 3.7. The former is a simple construction based on three one-dimensional functions, each of which plays a role in a certain stepsize regime: small, medium, or large. The small and the large regime functions are simple quadratics. The actually interesting case is the medium stepsize, that's when the dynamic is controlled by linear terms that make IG stray away from the optimal point.

The proof of Theorem 3.7 is mostly the same as the proof of Theorem 1 in (Mishchenko et al., 2020), which the authors explicitly state in the appendix. The main difference is the variance bound, which for Herding is improved since it's by definition better than a randomly sampled permutation.

---

> ### Author Rebuttal · Authors · 2025-04-01
>
> We thank the reviewer for the careful review and for the positive assessment of our paper. Below, we address the reviewer’s concerns and comments.
>
> 1. **The main limitation of the work, in my opinion, is that it considers the least interesting of the shuffling methods.**
>     - We agree with the reviewer’s view that IGD is the least interesting among the shuffling methods, and that deriving new results for RR would be more impactful. While we did attempt to extend our analysis to RR, we were unable to derive new convergence bounds. Nevertheless, given the scarcity of existing literature on permutation-based SGD in the small epoch regime, we believe that IGD offers a good starting point for understanding this regime better.
>     - Specifically, for IGD, constructing a lower bound requires designing a function specifically tailored to exhibit poor performance under a fixed permutation, and it is acceptable if this function converges quickly under other permutations. In contrast, for RR, we now have to design a function that consistently exhibits slow convergence on average over a wide range of permutations, making the analysis more challenging.
>     - We believe that obtaining new bounds (both upper and lower) for RR on general functions in the small epoch regime likely requires fundamentally new techniques. Developing such bounds for more complex shuffling schemes remains an important direction for future work.
> 2. **From the proofs, it appears to me that the lower bounds can be immediately extended to also include claims on the distance to the solutions.**
>     - Yes, the reviewer is absolutely correct that the proofs can be directly applied to derive lower bounds in terms of the distance to the solution, i.e., $\lVert x_n^K - x^*\rVert$. The reason why we state the lower bounds in terms of the function optimality gap is to match the form of the upper bounds. This consistency allows us to make a direct comparison and claim tightness between the lower and upper bounds. We will add a remark to clarify that our lower bound proofs can directly be used to obtain bounds in terms of the distance metric.
> 3. **Minor suggestion (vector, plural, capitalize)**
>     - Thank you for the careful reading. We have revised your suggested corrections.
>
> We are glad the reviewer took the time to read the proofs of Theorems 3.1 and 3.7. If the reviewer is interested, we would also like to recommend going over Theorem 3.3, which we believe contains the most novel idea in the paper. If there are any further comments or questions, we would be happy to address them.

---

> > ### Comment · Reviewer_oi4v · 2025-04-02
> >
> > > In contrast, for RR, we now have to design a function that consistently exhibits slow convergence on average over a wide range of permutations, making the analysis more challenging.
> >
> > I can see why this is more challenging, IGD seems easier to attack since we can control both the functions and the permutation. My intuition is that the counterexample for RR should be somewhat similar in nature. The cases of small and large stepsizes are equally trivial for RR, while for the mid range of stepsizes, the functions should probably still be roughly linear to make the method stray away from the solution. After all, the upper bound in the strongly convex case is derived using the extra sequence of points $x^*_i = x^* - \eta \sum\_{j=0}^{i-1} \nabla f\_{\pi\_{j}} (x^*)$, so we only care about the values of $\nabla f\_{\pi\_{j}} (x^*)$ and making the associated functions linear would only make it easier to study.
> >
> > > The reason why we state the lower bounds in terms of the function optimality gap is to match the form of the upper bounds.
> >
> > I encourage you to state the lower bounds for the distance terms as well, just in case others manage to derive upper bounds on the distances instead of the functional values, you will make their job of comparing the results easier. For instance, the guarantees in Theorem 1 of Mishchenko et al. (2020) are stated in terms of distances, so it's not a far fetched scenario.
> >
> > > We believe that obtaining new bounds (both upper and lower) for RR on general functions in the small epoch regime likely requires fundamentally new techniques. Developing such bounds for more complex shuffling schemes remains an important direction for future work.
> >
> > Hmmm, maybe, but I somehow feel that the lower bound construction shouldn't be that different. My intuition is that we can construct high-dimensional functions such that $\nabla f_i(x^*)$ is very hard to cancel unless a lot of other functions are sampled and added to it. In other words, we want to avoid the effect of the law of large numbers by using many dimensions and ensuring that $\Vert \sum\_{j=0}^{i-1} \nabla f\_{\pi\_{j}} (x^*)\Vert$ stays away from 0 with very high probability. Then, a construction similar to yours should do the thing I think.
> >
> > For instance, let us choose for each coordinate $s$ a pair of indices $i\_s, j\_s$ so that only $\nabla f\_{i\_s}(x^*)$ and $\nabla f\_{j\_s}(x^*)$ have non-zero entries at coordinate $s$ and they cancel each other out. In other words $[\nabla f\_{i\_s}(x^*)]\_s = -[\nabla f\_{j\_s}(x^*)]\_s$. Then, if $i\_s$ is sampled at the beginning of a permutation, with high probability the coordinate $s$ of $\sum\_{j=0}^{i-1} \nabla f\_{\pi\_{j}} (x^*)\$ is going to stay away from 0 for a long time just because it will take a lot of time to sample $j\_s$ to cancel that coordinate. And if we do this for a lot of coordinates, it basically means that whatever functions I sample at the beginning, they are likely to keep increasing the magnitude of $\Vert \sum\_{j=0}^{i-1} \nabla f\_{\pi\_{j}} (x^*)\Vert$ as long as $i\le n/2$. More formally, if we split numbers $1, 2, \dotsc, n$ into $n/2$ pairs, and then sample a permutation, it seems likely that among the first $n/2$ numbers there would be at least $\Omega(n)$ numbers without a pair among them. Maybe something like that would work?
> >
> > >  we would also like to recommend going over Theorem 3.3, which we believe contains the most novel idea in the paper.
> >
> > Thanks, that's indeed quite interesting. Reminded me of the old counterexample (in the sense of slow convergence) for the iterative projection method, where $n$ lines intersecting at 0 are constructed so that the iterates go in a slow spiral around the solutions. Maybe it's even equivalent since your function's are quadratic. I can't find the reference, but the visualization looks roughly like this:
> > ```
> > \   |   /
> >  \  |  /
> >   \ | /
> > ----+---
> >   / | \
> >  /  |  \
> > /   |   \
> >  ```

---

> > > ### Author Response · Authors · 2025-04-06
> > >
> > > Thank you very much for the insightful comments. We fully agree with the reviewer’s points, and would like to briefly share our own thoughts regarding upper and lower bounds for RR in the small epoch regime.
> > >
> > > We would like to begin by summarizing the current state of research on RR in the small epoch regime. To the best of our knowledge, there are **two noteworthy results** (under the assumption that the overall function is strongly convex and each component is smooth):
> > >
> > > 1. [Mishchenko et al., 2020]: When all component functions are also strongly convex, an upper bound of $O(\frac{L^2}{\mu^3 n K^2})$ is provided.
> > > 2. [Safran & Shamir, 2021]: When all component functions are quadratic and their Hessians commute, a tight convergence rate of $\Theta(\frac{1}{\mu n K})$ is established.
> > >
> > > Unlike scenario (2) where the authors provide matching UB and LB (up to polylogarithmic factor), the lower bound in scenario (1) is unknown, and it remains open whether the rate $O(\frac{L^2}{\mu^3 n K^2})$ can be improved or not.
> > >
> > > Given this context, there are **two clear directions for future exploration** in small epoch RR literature:
> > >
> > > - [**Upper Bound Direction**]: Improve the existing bound of $O(\frac{L^2}{\mu^3 n K^2})$ under the strongly convex component assumption, or derive new bounds under weaker assumptions (e.g., convexity, or even without convexity).
> > > - [**Lower Bound Direction**]:  Develop a matching lower bound (under the strongly convex component case) to close the gap with the existing upper bound $O(\frac{L^2}{\mu^3 n K^2})$.
> > >
> > > The primary challenge on the **upper bound** side is that deriving new upper bounds in the small epoch regime appears to require sophisticated analytical techniques (due to challenges discussed in Section 2.4). As can be found in [Safran & Shamir, 2021], even the proof for 1D quadratic is highly technical. One promising technique we explored is from [Koloskova et al., 2024]. In contrast to traditional analyses that group updates within a single epoch (i.e., chunks of size $n$), this method groups updates into chunks of size $\tau:=1/\eta L$. While this chunk-based approach can be successfully applied to derive upper bounds for IGD, it becomes problematic for RR. Specifically, when the chunk size $\tau$ does not align neatly within epochs, handling the dependencies between iterates becomes extremely difficult.
> > >
> > > Regarding the **lower bound** direction, we believe any progress beyond current results will likely require more “complicated” constructions that go beyond simple quadratic functions. This is because for simple quadratic functions where the Hessians commute with each other (e.g., $f_i(x_1, x_2) = \frac{L}{2}x_1^2 + a_i x_1 + \frac{\mu}{2}x_2^2 + b_i x_2$), the tight rate of $\Theta(\frac{1}{\mu n K})$ is already established by [Safran & Shamir, 2021]. Therefore, to surpass the existing LB barrier $\Omega(\frac{1}{\mu n K})$, future constructions must involve **quadratic functions with non-commuting Hessians or even non-quadratic functions**, necessitating more advanced analytical techniques. While our own lower bound construction in Theorem 3.3 is based on quadratic functions with non-commuting Hessians, it is tailored to IGD, and we do not see a clear way to extend this idea to RR.
> > >
> > > Regarding the reviewer’s intuition about potential lower bound constructions for RR involving high-dimensional pairing, such constructions would similarly require quadratic functions with non-commuting Hessians or non-quadratic functions, thus still posing analytical challenges. Moreover, introducing a pairing-based construction may not necessarily be beneficial. To explain why, consider a simple example: focusing on the first dimension (denoted as $x_1$), and suppose the first two component functions are $f_1(x_1) = a_1x_1^2 - Gx_1$ and $f_2(x_1) = a_2x_1^2 + Gx_1$, respectively. Now, if the remaining component functions are set to zero, i.e., $f_i(x_1) \equiv 0$, then the overall strong convexity parameter decreases by a factor of $1/n$, negatively affecting the function optimality gap. Conversely, if the component functions are set to $f_i(x_1) = a_ix_1^2$, then the iterates along the first dimension shrink with each step, preventing the function optimality gap from becoming sufficiently large. Due to these reasons, we do not find an immediate way to establish a new lower bound using the reviewer’s suggested construction. Nevertheless, we find the reviewer’s suggestion valuable, and believe it serves as a promising starting point.

---

### Official Review · Reviewer_oG8h · 2025-03-16

**Overall Recommendation:** 3

**Summary:**

This paper analyzes Incremental Gradient Descent (IGD) method in various convex settings, and establishes lower bounds of IGD in the small epoch regime and large epoch regime. This paper also provides upper bound results for arbitrary permutation-based SGD in several small epoch and large epoch regimes. This paper also design a new permutation-based SGD method that outperforms with-replacement SGD in small epoch regime with strongly convex component functions.

**Claims And Evidence:**

Yes

**Essential References Not Discussed:**

NA

**Experimental Designs Or Analyses:**

NA, purely theoretic work.

**Methods And Evaluation Criteria:**

Yes

**Other Comments Or Suggestions:**

Adding some simple numerical experiments to justify the theoretical results would greatly boost the confidence in the correctness of the proofs.

**Other Strengths And Weaknesses:**

Strengths:
- This paper is generally well-written.
- The upper bound and lower bound analysis of IGD is comprehensive, complementing existing work.
- The discovered slow performance of IGD in the small epoch regime is interesting.

Weaknesses:
- The storyline of paper is not so clear. The current version seems like a collection of upper bounds and lower bounds under various setups, using different permutation schemes, i.e., the results are not strongly connected.
- Several assumptions are very restrictive, e.g., 1-dimension assumption in Theorem 3.2
- This paper seems to lack a stricking result imho.

**Questions For Authors:**

NA

**Relation To Broader Scientific Literature:**

There are several permutation-based SGD published in past ICML events.

**Theoretical Claims:**

I do not have the time to go through all the details. Up to what I have verified, everything looks correct.

---

> ### Author Rebuttal · Authors · 2025-04-01
>
> We thank the reviewer for the thoughtful review. Below, we address the reviewer’s concerns.
>
> 1. **The storyline of paper is not so clear.**
>     - While our analysis does involve three different permutation schemes—Incremental Gradient Descent (IGD), arbitrary permutation-based SGD, and Herding at Optimum—we would like to clarify that these schemes form a well-connected framework rather than a disjointed collection of results.
>     - First, we emphasize that **(1) IGD** appears exclusively in **lower bounds**, and **(2) Arbitrary permutation-based SGD** appears only in **upper bounds**. Importantly, arbitrary permutation-based SGD contains the worst-case permutation scenario. Therefore, when the LB of IGD matches the UB of arbitrary permutation-based SGD, we obtain tight convergence guarantees for worst-case permutation-based SGD (as discussed in Line 171 (left) and Appendix A). Our results establish such bounds across both small and large epoch regimes, and under varying assumptions on the component functions.
>     - **Herding at Optimum** (Theorem 3.7) illustrates how fast permutation-based SGD can converge under (near-)optimal permutations in the small epoch regime. Together, the three schemes characterize the full spectrum of convergence behavior for permutation-based SGD, from worst-case to (nearly) best-case. One caveat is that as discussed in Line 377 (left), the current Herding at Optimum is not an implementable algorithm in general scenarios; making it practical is a promising direction for future work.
> 2. **Several assumptions are very restrictive.**
>     - We would like to clarify several points that may help the reviewer better understand the context behind these choices.
>     - First, unlike the upper bound analysis, strong assumptions in lower bound theorems strengthen the result. This is because the **narrower** the function class for which a lower bound holds, the **stronger** the applicability when compared with upper bounds.
>     - Also, we acknowledge that assumptions on the component functions’ Hessians (e.g., identical Hessians or strong convexity) may appear restrictive. However, our results show that the convergence behavior of IGD varies significantly under different settings. Thus, these assumptions serve to illustrate meaningful distinctions, rather than to artificially strengthen the conclusions.
>     - Lastly, in upper bound analyses, stronger assumptions indeed weaken the result. As pointed out by the reviewer, 1D setting in Theorem 3.2 is quite restrictive. However, deriving upper bounds for permutation-based SGD methods in the small epoch regime is known to be **extremely challenging** without strong assumptions. Specifically, [1] derives an upper bound for Random Reshuffling, but only under strong conditions—quadratic objectives with component Hessians that are mutually commutative, symmetric, and PSD—essentially reducing the problem to 1D.
> 3. **This paper seems to lack a striking result.**
>     - We would like to highlight that our main contribution lies in addressing a largely underexplored regime: **the small-epoch behavior of shuffling methods**. To our knowledge, except for the result by [1], no prior work establishes tight convergence bounds in this regime.
>     - Our paper provides tight lower and upper bounds for IGD in this regime. While IGD is the simplest among permutation-based methods, the theoretical analysis in this regime is highly nontrivial and challenging (see Section 2.4).
>     - In particular, when the component functions are allowed to be nonconvex, we show that the convergence of IGD can be *exponentially* slow. Our result is the first to rigorously establish this phenomenon, suggesting that permutation-based SGD methods can suffer severe slowdowns in the small epoch regime. This motivates further investigation into whether similar slowdowns occur in other permutation-based SGD methods, including Random Reshuffling (RR).
> 4. **Adding some simple numerical experiments would boost the confidence in the correctness of the proofs.**
>     - We would like to note that we included numerical experiments on our lower bound constructions in Appendix G (as mentioned in Line 303 (right)). These experiments confirm that IGD exhibits slow convergence on the constructed objectives from Theorems 3.3 and 3.5. In particular, we observe exponential-type slow convergence in the case of Theorem 3.5.
>     - In addition, another reviewer suggested evaluating performance in a more practical scenario. During the rebuttal period, we conducted experiments on a binary classification task using two selected labels from the MNIST dataset. We kindly refer the reviewer to our response to Reviewer RSZD for further details.
>
> We hope our responses have fully addressed your concerns. If you have any remaining concerns or questions, we would be happy to respond.
>
> ---
>
> [1] Safran, Itay, and Ohad Shamir. "Random shuffling beats SGD only after many epochs on ill-conditioned problems." NeurIPS 2021

---

### Decision · Program_Chairs · 2025-05-01

**Decision:**

Accept (poster)

**Comment:**

Summary: The paper inspired studies of shuffling-based methods and provides new theoretical results for specific permutations. In particular, it gives new lower bounds on the Incremental Gradient (IG) method in the low-epoch regime. The results concern the strongly convex case, but with multiple options on the individual components: either all of them are strongly convex, or just convex, or even potentially nonconvex.

On Reviews: All reviewers suggested acceptance. During the rebuttal phase, the authors respond clearly to all questions raised.

I advise the authors to incorporate the feedback they received into the updated version of their work, especially the comments regarding upper and lower bounds for RR in the small epoch regime, mentioned during discussion with Reviewer oi4v.

I recommend acceptance.